



# Correcting high-frequency losses of reactive nitrogen flux measurements

Pascal Wintjen[1], Christof Ammann[2], Frederik Schrader[1], and Christian Brümmer[1]

[1]Thünen Institute of Climate-Smart Agriculture, Bundesallee 65, 38116, Braunschweig, Germany
[2]Climate and Agriculture Group, Agroscope, Reckenholzstrasse 191, 8046, Zürich, Switzerland

**Correspondence:** Christian Brümmer (christian.bruemmer@thuenen.de)

**Abstract.** The eddy-covariance (EC) technique is nowadays widely used in experimental field studies to measure land surface-atmosphere exchange of a variety of trace gases. In recent years applying the EC technique to reactive nitrogen compounds has become more important since atmospheric nitrogen deposition influences the productivity and biodiversity of (semi-)natural ecosystems and its carbon dioxide ($CO_2$) exchange. Fluxes which are calculated by EC have to be corrected for setup-specific
effects like attenuation in the high-frequency range. However, common methods for correcting such flux losses are mainly optimized for inert greenhouse gases like $CO_2$ and methane or water vapor. In this study, we applied a selection of correction methods to measurements of total reactive nitrogen ($\Sigma N_r$) conducted in different ecosystems using the Total Reactive Atmospheric Nitrogen Converter (TRANC) coupled to a chemiluminescence dectector (CLD). Average flux losses calculated by methods using measured cospectra and ogives were about 26-38% for a semi-natural peatland and about 16-22% for a mixed
forest. The investigation of the different methods showed that damping factors calculated with measured heat and gas flux cospectra using an empirical spectral transfer function were most reliable. Flux losses of $\Sigma N_r$ with this method were on the upper end of the median damping range, i.e. 38% for the peatland site and 22% for the forest site. Using modified Kaimal cospectra for damping estimation worked well for the forest site, but underestimates damping for the peatland site by about 12%. Correction factors of methods based on power spectra or on site-specific and instrumental parameters were mostly less
than 10%. Power spectra of $\Sigma N_r$ were heavily affected likely by white noise and deviated substantially at lower frequencies from the temperature (power) spectrum. Our study suggests using an empirical method for estimating flux losses of $\Sigma N_r$ or any reactive nitrogen compound and locally measured cospectra.



## 1   Introduction

The eddy-covariance (EC) method is widely applied for determining turbulent exchange of trace gases and energy between
biosphere and atmosphere (Aubinet et al., 2012; Burba, 2013). EC is mainly used for long-lived, stable gases like carbon dioxide
($CO_2$), water vapour ($H_2O$) and methane ($CH_4$). Only a few studies concentrated on reactive, short-lived gases like reactive
nitrogen compounds ($N_r$). In our study $N_r$ covers species like nitrogen monoxide (NO), nitrogen dioxide ($NO_2$), nitric acid
($HNO_3$), nitrous acid (HONO), peroxyacetyl nitrate (PAN), ammonia ($NH_3$) and particulate ammonium nitrate ($NH_4NO_3$).
The sum of these species is called total reactive nitrogen ($\Sigma N_r$). Nitrous oxide ($N_2O$), sometimes also considered as reactive
N compound, is not detected with our system (*cf.* Sec. 2.1) and is excluded from $\Sigma N_r$ here and not taken into account.

Application of the EC technique to $N_r$ or $NH_3$ is challenging, because most $N_r$ compounds are highly reactive, water soluble
and background concentrations are typically low. In close proximity to sources like stables, managed fields (Sutton et al.,
2011; Flechard et al., 2013), traffic, or industry (Sutton et al., 2011; Fowler et al., 2013), compounds of $N_r$ like $NH_3$ or $NO_2$
can reach high concentrations. In the past, low-cost measurement devices like passive samplers (Tang et al., 2009), DELTA
denuder (DEnuder for Long-Term Atmospheric sampling) (Sutton et al., 2001) or wet chemistry analyzers (von Bobrutzki
et al., 2010) were mainly used in $N_r$ measurement studies. However, these instruments typically have a low time resolution
and require inferential modeling for estimating fluxes (e.g., Hurkuck et al., 2014). Recently, new measurement techniques for
$N_r$ compounds were developed, such as quantum cascade lasers (QCL) using Tunable Infrared Laser Differential Absorption
Spectroscopy (TILDAS) (Ferrara et al., 2012; Zöll et al., 2016; Moravek et al., in review, 2019) or the total reactive nitrogen
converter (TRANC) (Marx et al., 2012; Ammann et al., 2012; Brümmer et al., 2013; Zöll et al., 2019) coupled to a fast-
response chemiluminescene detector (CLD) which have a certain robustness, a high sampling frequency and are sensitive
enough to allow EC measurements of $NH_3$ or $\Sigma N_r$.

Evaluating fluxes with these closed-path EC systems leads to underestimation of fluxes due to damping in the high and low-
frequency range. An EC setup, like any measurement setup, is comparable with a filter which removes high and low-frequency
parts from measured signals. High-frequency losses are for example related to sensor separation (Lee and Black, 1994), air
transport through tubes in closed-path systems (Leuning and Moncrieff, 1990; Massman, 1991; Lenschow and Raupach, 1991;
Leuning and Judd, 1996), different response characteristics of the instruments and phase-shift mismatching (Ammann, 1999).
These processes inducing flux losses are usually described by spectral transfer functions (Moore, 1986; Zeller et al., 1988;
Aubinet et al., 1999).

The magnitude of the high-frequency flux loss depends on the trace gas of interest, the experimental setup, wind speed,
and atmospheric stability. In the recent literature, different estimates of flux losses due to high-frequency damping have been
reported. For example Zöll et al. (2016) found flux losses of 33% for $NH_3$ at an ombrotrophic, moderately drained peatland site.
Ferrara et al. (2012) used the same QCL instrument and estimated flux losses from 23% to 43% depending on the correction
method. Ammann et al. (2012) measured $\Sigma N_r$ with a TRANC-CLD system at an intensively managed grassland site and
estimated flux losses between 19% and 26%. Brümmer et al. (2013) operated a TRANC-CLD system at a managed agricultural
site and calculated flux losses of roughly 10%. Stella et al. (2013) calculated flux losses of 12–20% for NO and 16–25% for





$NO_2$. Evidently, the range and magnitude of flux losses of $\Sigma N_r$ and several compounds is quite large. Correction factors for $CO_2$ and $H_2O$ are usually lower. $CO_2$ shows for a closed-path EC setup attenuation factors from 2% up to 15% (Su et al., 2004; Ibrom et al., 2007; Mammarella et al., 2009; Burba et al., 2010; Butterworth and Else, 2018). $H_2O$ shows a stronger

damping than $CO_2$ that depends on humidity and age of intake tube due to interactions of sample air water vapor with the inner tube surfaces. The corresponding flux loss varies from 10% to 42% (Su et al., 2004; Ibrom et al., 2007; Mammarella et al., 2009; Burba et al., 2010). Mammarella et al. (2009) reported that strong damping (up to 40%) of $H_2O$ occurs in wintertime and during night due to high relative humidity and only 10% to 15% during summertime.

In the past decades several methods for calculating spectral correction factors have been proposed based on theoretical

cospectra (Kaimal et al., 1972; Moore, 1986; Moncrieff et al., 1997), measured power spectra (Ibrom et al., 2007; Fratini et al., 2012) and measured cospectra or ogives (Ammann et al., 2006). Some of these methods are implemented in ready-to-use eddy covariance post-processing packages like EddyPro (LI-COR Biosciences, Lincoln, USA). In principle, it is possible to calculate flux losses without measuring trace gas concentrations, if all physical parameters of the setup and process losses are known. Such a method does not consider gas-specific properties and may not be suitable for highly reactive gases. In general, all these

methods are optimized for inert greenhouse gases and not for $N_r$ species. It is therefore questionable if common methods for spectral correction are applicable for $N_r$ given the high reactivity and chemical characteristics of single compounds. Recently, Polonik et al. (2019) found that the applied correction method depends strongly on the gas of interest ($CO_2$ and $H_2O$) and the type of gas analyzer used. They suggest that high-frequency attenuation of closed and enclosed devices measuring $H_2O$ should be corrected empirically. Consequently, common methods are not perfectly suited for dealing with specific EC setups. In this

study, we test five different spectral damping correction methods for EC fluxes of $\Sigma N_r$ that were measured at two different sites using a TRANC-CLD system. We investigate (1) quantitative differences between the methods, (2) their sensitivity to the input data and (3) dependencies on meteorological conditions (wind speed, atmospheric stability, etc.) and measurement height.

## 2 Methods

### 2.1 Sites and experimental setup

We analyzed data from two measurement sites. At both sites we installed a custom-built $\Sigma N_r$ converter (total reactive atmospheric nitrogen converter, TRANC) after Marx et al. (2012), a 3-D ultrasonic anemometer (GILL-R3, Gill Instruments, Lymington, UK), a fast-response chemiluminescence detector (CLD 780 TR, ECO PHYSICS AG, Dürnten, Switzerland) and a dry vacuum scroll pump (BOC Edwards XDS10, Sussex, UK).

The first site (52°39'N 7°11'N, 19 m a.s.l) is a semi-natural peatland in Northwest Germany, called 'Bourtanger Moor'

(BOG). It is an ombrotrophic, moderately drained bog with high ambient $NH_3$ concentrations (Zöll et al., 2016) dominating the local deposition of $\Sigma N_r$ (Hurkuck et al., 2014). A detailed description of the site is given in Hurkuck et al. (2014, 2016). The EC system was operated from October 2012 to mid of July 2013.

TRANC and sonic anemometer were installed at 2.50 m above ground. While passing through the TRANC, air samples undergo two conversion steps. The first one is a thermal pathway inside an iron-nickel-chrome (FeNiCr) alloy tube at approx.





870 °C. In a passively heated gold tube (approx. 300 °C) a catalytic conversion follows. Before reaching the gold tube, carbon monoxide is applied as a reducing agent. Finally, all $\Sigma N_r$ (except for $N_2O$ and $N_2$) are converted to NO. At the end of the converter a critical orifice is mounted, which ensures a pressure reduction at a constant flow rate of $\sim 2.0 \, \mathrm{L\,min^{-1}}$. After passing through a 12 m opaque Polytetrafluoroethylene (PTFE) tube the sample air is analyzed in the CLD with a sampling frequency of 20 Hz. The GILL-R3 was installed next to the inlet of the TRANC. CLD and pump were located in an air-

conditioned box. For further details of converter and field applications, we refer to Marx et al. (2012), Ammann et al. (2012), and Brümmer et al. (2013). It was shown that concentrations measured by the CLD are affected by water vapour due to quantum mechanical quenching. To compensate for this effect, calculated fluxes were corrected after the approach by Ammann et al. (2012) and Brümmer et al. (2013). Next to the $\Sigma N_r$ setup, another EC system for $CO_2$ and $H_2O$ measurements was placed (Hurkuck et al., 2016) using a GILL-R3 and a fast-response, open-path infrared gas analyzer (IRGA, LI-7500, LI-COR

Biosciences, Lincoln, USA).

Our second site (48°56'N 13°56'N, 807 m a.s.l) was located in the Bavarian Forest (FOR) National Park, Germany. The same TRANC and sonic anemometer were mounted on different booms next to each other at a height of 30 m above ground and approximately 10 m above the forest canopy. Next to the sonic, an open-path LI-7500 infrared gas analyzer (IRGA) for measuring $CO_2$ and $H_2O$ concentrations was installed. CLD and pump were placed in an air-conditioned box at the bottom of

the tower. A 45 m long, opaque PTFE tube connected the TRANC with the CLD. A critical orifice at the end of the TRANC restricted the flow to $2.1 \, \mathrm{L\,min^{-1}}$ and assures low pressure along the tube. Air temperature and relative humidity sensors (HC2S3, Campbell Scientific, Logan, Utah, USA) were mounted at four different heights along a vertical gradient (10, 20, 40 and 50 m). The site was located in a remote area, next to the Czech border, with no local industrial and agricultural emission hotspots (Beudert et al., 2018). Therefore, concentrations of $N_r$ species such as $NH_3$ (1.3 ppb), NO (0.4-1.5 ppb), and $NO_2$

(1.9-4.4 ppb) were very low (Beudert and Breit, 2010). A detailed description of the forest site can be found in Zöll et al. (2019). For the attenuation analysis data from June 2016 to end of June 2018 were selected. Important site specific parameters of both measurement sites are listed in Table 1.





**Table 1.** Physical parameters of the EC-setups

| Parameter | Bourtanger Moor (BOG) | Bavarian Forest (FOR) |
|---|---|---|
| canopy height | $0.4\,\mathrm{m}$ | $20\,\mathrm{m}$ |
| measurement height (from ground) | $2.5\,\mathrm{m}$ | $31\,\mathrm{m}$ |
| displacement height | $0.268\,\mathrm{m}$ | $13.4\,\mathrm{m}$ |
| tube length | $12\,\mathrm{m}$ | $48\,\mathrm{m}$ |
| tube diameter (OD) | $6.4\,\mathrm{mm}$ | $6.4\,\mathrm{mm}$ |
| flow rate | $2.0\,\mathrm{Lmin^{-1}}$ | $2.1\,\mathrm{Lmin^{-1}}$ |
| horizontal sensor separation | $5\,\mathrm{cm}$ | $32\,\mathrm{cm}$ |
| vertical sensor separation (below the sonic) | $20\,\mathrm{cm}$ | $20\,\mathrm{cm}$ |
| sonic path length | $15\,\mathrm{cm}$ | $15\,\mathrm{cm}$ |
| CLD analyser response time | $0.3\,\mathrm{s}$ | $0.3\,\mathrm{s}$ |
| acquisition frequency | $20\,\mathrm{Hz}$ | $10\,\mathrm{Hz}$ |
| kinematic viscosity | $1.46\cdot10^{-5}\,\mathrm{m^2s^{-1}}$ | $1.46\cdot10^{-5}\,\mathrm{m^2s^{-1}}$ |
| Schmidt number for NO | $0.87$ | $0.87$ |
| time delay | $2.5\,\mathrm{s}$ | $20\,\mathrm{s}$ |

## 2.2 Calculation and quality selection of fluxes and spectra

Data were collected with the software EddyMeas, included in the software EddySoft (Kolle and Rebmann, 2007), with time
resolutions of $20\,\mathrm{Hz}$ at BOG and $10\,\mathrm{Hz}$ at FOR. Analog signals from CLD and LI-7500 were sampled by the interface of the
anemometer and combined with the ultrasonic wind components and temperature data to a common data stream. The software
Eddy Pro 6.2.1 (LI-COR Biosciences, 2017) was used for raw data processing and flux calculation. A 2-D coordinate rotation
of the wind vector was selected (Wilczak et al., 2001), spikes were detected and removed after Vickers and Mahrt (1997) and
block averaging was applied.

The recorded datasets show a time lag between the measurements of the sonic and the gas analysers due to sampling of air
through the inlet system (converter, tube, analyzer cell), the processing of signals within the analysers and the distance between
the two instruments. The time lag was estimated with the covariance maximization method (Aubinet et al., 2012; Burba, 2013),
which is based on shifting the time series of vertical wind and concentration against each other to determine the lag time at
which the covariance between the two is maximized. At BOG the time lag was around $2.5\,\mathrm{s}$ and at FOR the time lag was around
$20\,\mathrm{s}$. Accordingly, the time lag computation method in Eddy Pro was set to covariance maximization with default. Based on
theoretical considerations, we restricted the range for time lag computation from $15\,\mathrm{s}$ to $25\,\mathrm{s}$ for the FOR data and from $0\,\mathrm{s}$ to
$5\,\mathrm{s}$ for the BOG data. The default value was set to $20\,\mathrm{s}$ for FOR and to $2.5\,\mathrm{s}$ for BOG, respectively. The windows for the time
lag compensation were chosen in such a way, because estimated lags were broadly distributed around the physical (default)





lag. Time lags, estimated with a stand alone script, are used as filtering criteria for the damping analysis (see Sec. 3.1). For the
$CO_2$ and $H_2O$ measurements, time lags were mostly negligible.

For the high-frequency damping analysis, we selected time series of vertical wind, temperature and $\Sigma N_r$ concentrations.
These raw data were corrected for several effects in the following order: despiking (Vickers and Mahrt, 1997), cross wind
correction (Liu et al., 2001), angle of attack correction (Nakai et al., 2006), tilt correction (Wilczak et al., 2001), time lag
compensation and block averaging. Then the timeseries were subject to a fast fourier transformation (FFT) that yielded the
power spectra of individual quantities like the temperature (power) spectrum $Ps(T)$ and the cospectra of two quantities like the
heat flux cospectrum $Co(w, T)$ (Aubinet et al., 2012). The same was done for $CO_2$, $H_2O$ and $\Sigma N_r$ resulting in $Co(w, CO_2)$,
$Co(w, H_2O)$ and $Co(w, \Sigma N_r)$; $Ps(CO_2)$, $Ps(H_2O)$ and $Ps(\Sigma N_r)$, respectively. From the cospectra flux-normalized ogives (Og)
were calculated (Ammann et al., 2006) as cumulative cospectrum (Desjardins et al., 1989; Oncley et al., 1996). The ogives and
cospectra consisted of 40 log-spaced frequency bins.

For a quantitative evaluation of the high-frequency damping from the half-hourly flux (co)spectra a quality flagging has to be
applied. Flagging of (co)spectra is done automatically in EddyPro. However, the criteria are usually optimized for inert gases
like $CO_2$ and $H_2O$ that show characteristic daily flux cycles and magnitudes. They are much less specific and were not very
successful for filtering $\Sigma N_r$ fluxes and spectra. Therefore we performed a two-stage quality selection. First common criteria
were applied: discarding cases with (i) insufficient turbulence ($u_* < 0.1\,\mathrm{ms}^{-1}$), (ii) low flux quality (flag=2) after Mauder and
Foken (2006), (iii) variances of $T$ and $\Sigma N_r$ exceeding a threshold of $1.96\sigma$ and (iv) a time lag outside the predefined range (see
Sect. 2.2). In second stage with manual screening we checked whether the shape of ogives and cospectra is relatively smooth
and not influenced by considerable noise or outliers. A total of 821 cospectra passed the flagging criteria at BOG and 872
cospectra passed the flagging criteria at FOR. With common selection criteria, 3232 cases at BOG and 9889 at FOR would
have been retrieved.

Another possibility for the characterization of the quality or influence of noise on power spectra and cospectra is the deter-
mination of the decline in the inertial subrange following the power law. Therefore the slope of the decrease was evaluated on
a double logarithmic scale by a linear regression. The theoretical slope is -2/3 for $Ps(T)$ as well as for power spectra of inert
gases or even lower depending on trace gas and measurement setup.

## 2.3    High-frequency damping and determination of correction factor

We used four different cospectral approaches for the computation of high-frequency losses. The fifth approach was the method
of Ibrom et al. (2007) based on power spectral analysis and is implemented in EddyPro. The majority of the approaches
determine the damping factor of a trace gas flux as integral of the frequency-dependent attenuation of the corresponding
cospectrum. With $Co(f)$ being the true undamped cospectrum, the flux damping factor(s) $\alpha$ or its inverse, the correction factor
$\alpha^{-1}$, can be described in the following way, e.g. (Moore, 1986):

$$\alpha = \frac{\int_{f=0}^{\infty} TF(f) Co(f)\,\mathrm{d}f}{\int_{f=0}^{\infty} Co(f)\,\mathrm{d}f} \tag{1}$$


$TF$ is the overall spectral transfer function of the EC setup and is usually a product of several individual damping processes with specific transfer functions $TF_i$. In the following subsections we describe the methods in detail.

### 2.3.1 Theoretical damping calculation [THEO]

The theoretical damping calculation [THEO] is the most commonly applied method (Spank and Bernhofer, 2008). It is in-
dependent of any measured data and works for open-path as well as closed-path EC systems (Leuning and Moncrieff, 1990; Lenschow and Raupach, 1991; Massman, 1991; Leuning and Judd, 1996; Moncrieff et al., 1997). It is based on the assumption, that all relevant attenuation processes are known and can be quantitatively described by spectral transfer functions $TF_i$. Detailed descriptions of the $TF_i$ are given in Moore (1986); Moncrieff et al. (1997); Ammann (1999); Aubinet et al. (1999, 2012). The $TF_i$ and physical parameters for the EC setups used here, like the analyser response time $\tau_r$, flow rate, tube length,
sensor separation, are listed in Tables A1 and 1. All $TF_i$ were merged to a single total transfer function ($TF_{\text{theo}}$), which was applied to theoretical (modified) Kaimal cospectra (from the original (Kaimal et al., 1972)). Subsequently, $\alpha$ was calculated after Eq. (1) for every quality selected flux averaging interval. Kaimal cospectra exclusively depend on stability, wind speed and measurement height above canopy (Moore, 1986; Ammann, 1999). Further in-situ measurements were not used for this approach.

### 2.3.2 In-situ cospectral method [ICO]

Theoretical cospectra could deviate from site-specific characteristics of the turbulent transfer, while theoretical transfer functions could miss important chemical or microphysical processes, which are more important for $\Sigma N_r$ than for inert gases like $CO_2$, $H_2O$, $CH_4$ or $N_2O$. In the exemplary case of Fig. 1, the prescribed cospectrum of Kaimal overall corresponds well with $Co(w,T)$, but a systematic deviation may exist in the low-frequency range for BOG. For $Co(w,N_r)$ at both sites differences are
also visible in the high-frequency range right of the cospectral maximum which is around $0.2\,\text{Hz}$ for BOG and around $0.02\,\text{Hz}$ for FOR in the present example.





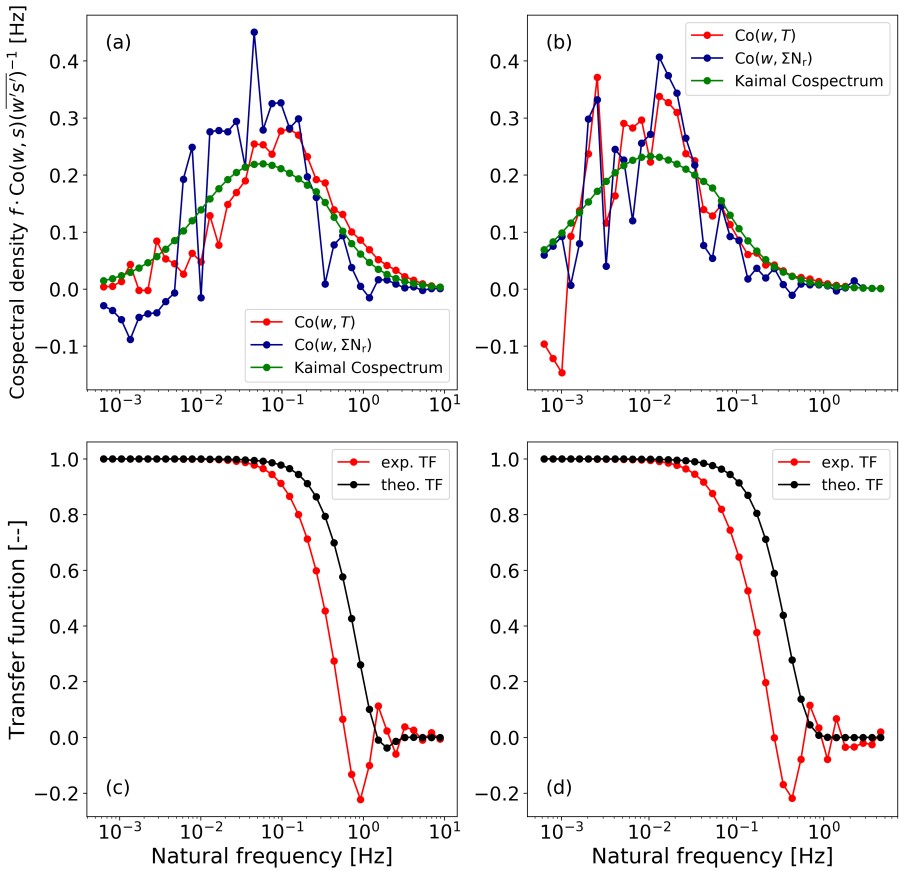

**Figure 1.** Comparison of observed normalized cospectra with modified Kaimal cospectra (green) for similar wind speed and stability and their theoretical and experimental transfer functions at ((a),(c)) Bourtanger Moor ($\zeta$= -0.23, $\bar{u}$= 1.38 ms$^{-1}$) and ((b),(d)) Bavarian Forest site ($\zeta$= 0.17, $\bar{u}$= 2.04 ms$^{-1}$). Panels (c) and (d) show the theoretical cospectral transfer function ($TF_{\text{theo}}$) (black) and the experimental transfer function ($TF_{\text{exp}}$) (red). The experimental transfer functions were determined with the cospectra in (a) and (b). The displayed cospectra of heat (red) and $\Sigma N_r$ mass flux (blue) are averaged over half-hourly measurements on 10.10.2012 between 09:30 and 14:00 and on 28.10.2016 between 10:00 and 15:30 for BOG and FOR, respectively.

Cospectra of FOR are shifted to the left due to the larger measurement height above canopy and the increased contribution of low-frequent, large-scale eddies with height (Burba, 2013). The in-situ cospectra method (ICO) utilizes $\text{Co}(w,T)$ instead of the Kaimal cospectrum in (Eq. 1). $\text{Co}(w,T)$ is used as reference cospectrum, because it is almost unaffected by damping

processes. Assuming spectral similarity between $\text{Co}(w,T)$ and $\text{Co}(w,\Sigma N_r)$ we can derive $TF_{\text{exp}}$ as follows (Aubinet et al., 1999; Su et al., 2004):

$$\alpha \cdot \frac{\text{Co}(w,\Sigma N_r)}{\overline{w'\Sigma N_r'}} = TF_{\text{exp}} \cdot \frac{\text{Co}(w,T)}{\overline{w'T'}} \tag{2}$$


$TF_{\mathrm{exp}}$ consists of a first-order filter combined with a mismatching phase shift for first order systems (Ammann, 1999) (Table A1).

$$TF_{\mathrm{exp}}(f) = TF_{\mathrm{R}}(f) \cdot TF_{\Delta\mathrm{R}}(f) \tag{3}$$

A least-square fit of Eq. (2) was performed with $\tau_{\mathrm{r}}$ as optimization parameter. Equation (2) was solved iteratively until $\alpha$ converged. Thereby $\alpha$ was determined with (1). The fit was done for frequencies larger than 0.055 Hz for the BOG campaign. This frequency range is assumed to be affected by damping effects. A frequency limit had been used in the damping analysis of Zöll et al. (2016) for the same site. For the FOR campaign the frequency limit was set to 0.025 Hz. $\tau_{\mathrm{r}}$ is linked to the cut-off frequency $f_c = 1/2\pi\tau_{\mathrm{r}}$ at which the cospectrum is damped to $1/\sqrt{2} \approx 0.71$. Knowing $\tau_{\mathrm{r}}$ we calculated $\alpha$ with Eq. (3) and (1). Panel (c) and (d) of Fig. 1 show examples of the theoretical and experimentally determined transfer functions for the two measurement sites. In both cases the experimental transfer function drops earlier than the theoretical and reveals a significant variation in the high-frequency range.

### 2.3.3 Semi in-situ cospectra method [sICO]

The semi in-situ cospectra approach is similar to the one described in Sec. 2.3.2. The determination of $\tau_{\mathrm{r}}$ follows the same procedure as for ICO, but instead of using $\mathrm{Co}(w,T)$ in Eq. (1) this approach uses Kaimal cospectra (Eqs. (A1) and (A2)) as reference. This method is useful, if the quality of $\mathrm{Co}(w,T)$ is not sufficient for estimating the damping factors, especially in the low-frequency range.

### 2.3.4 In-situ ogive method [IOG]

The in-situ ogive method (IOG) is based on Ammann et al. (2006) and Ferrara et al. (2012). This method is similar to ICO but does not rely on a specific form for the spectral transfer functions or cospectra and only requires $\mathrm{Og}(w,T)$ and $\mathrm{Og}(w,\Sigma\mathrm{N_r})$. Again spectral similarity between $\mathrm{Og}(w,T)$ and $\mathrm{Og}(w,\Sigma\mathrm{N_r})$ is assumed. For estimating the damping a linear regression between $\mathrm{Og}(w,T)$ and $\mathrm{Og}(w,\Sigma\mathrm{N_r})$ was performed in a specific frequency range. The range was constrained by frequencies for which $\mathrm{Og}(w,T) > 0.2$ and $\mathrm{Og}(w,\Sigma\mathrm{N_r}) < 0.85$ was fulfilled. Frequencies lower than 0.002 Hz were excluded. The difference between the regression line and $\mathrm{Og}(w,\Sigma\mathrm{N_r})$ was calculated and points exceeding a difference of 0.1 or frequencies above which the signal is totally damped, were not considered for a least-square fit of $\mathrm{Og}(w,\Sigma\mathrm{N_r})$ against $\mathrm{Og}(w,T)$. The former criteria was applied for discarding spikes. Finally, the optimization factor corresponded to the damping factor.

### 2.3.5 In-situ power-spectral method [IPS]

Application of the in-situ power spectral method (IPS) after Ibrom et al. (2007) was executed using EddyPro. It uses measured power spectra of a reference scalar and of the trace gas of interest, here $\mathrm{Ps}(T)$ and $\mathrm{Ps}(\Sigma\mathrm{N_r})$. The first step - the estimation of $\tau_{\mathrm{r}}$ or the cut-off frequency $f_{\mathrm{c}}$ - is similar to the in-situ cospectra method (Eq. 2), but the transfer function is different.

$$\frac{\mathrm{Ps}(\Sigma\mathrm{N_r})}{\mathrm{Ps}(T)} = \frac{1}{1 + (f/f_{\mathrm{c}})^2} \tag{4}$$





For estimating $f_c$ Eddy Pro uses quality selected and averaged power spectra. We set 0.4 Hz as lowest noise frequency in the option 'removal of high frequency noise' and adjusted the threshold values for removing power spectra and cospectra from

the analysis accordingly. Additionally we forced EddyPro to filter the spectra after statistical (Vickers and Mahrt, 1997) and micrometeorological (Mauder and Foken, 2004) quality criteria. We applied the correction of instrument separation after Horst and Lenshow (2009) for crosswind and vertical wind and took the suggested lowest and highest frequency (0.006 Hz and 5 Hz) as fitting range for $Ps(T)$ and $Ps(\Sigma N_r)$ for FOR. Applying IPS to BOG data was not possible, because we did not measure $CO_2$ and $H_2O$ with the EC setup installed at BOG. Adding high-frequency $CO_2$ and $H_2O$ data from the EC setup described

in (Hurkuck et al., 2016) which was placed next to the $\Sigma N_r$ setup allows us to calculate correction factors after IPS for $\Sigma N_r$. We changed the highest frequency to 8 Hz and took the lowest frequency from standard settings (0.006 Hz). For comparing the results of IPS to our cospectral methods we chose the same half-hours which passed the automatic selection criteria and the manual screening (see Sec. 2.2). Finally, the correction factors were calculated after the following equation

$$\alpha^{-1} = \frac{A_1 \bar{u}}{A_2 + f_c} + 1 \qquad (5)$$

$A_1$ and $A_2$ were estimated for stable and unstable stratification using degraded time series of sonic temperature. The degradation was done by a varying low pass recursive filter (Ibrom et al., 2007; Sabbatini et al., 2018). In general, a summary of processing eddy-covariance data including high-frequency spectral correction methods is given in Sabbatini et al. (2018).

## 3    Results

### 3.1    Characterization of power spectra and cospectra

Figure 2 shows exemplary cospectra and power spectra of the two measurement sites. We compare cospectra which were measured during unstable daytime conditions and at similar wind speed. All in all, the cospectral densities of the gas and heat fluxes are quite similar. It indicates that the chosen sampling interval and frequency were sufficient to capture flux-carrying eddies. However, $Co(w, \Sigma N_r)$ shows a stronger variation than the other cospectra. The effect of different measurement height is quite obvious. It results in a shift of all cospectra to the left for the FOR site. The stronger drop of $Co(w, \Sigma N_r)$ compared to

$Co(w, CO_2)$ and $Co(w, H_2O)$ in the high-frequency range is likely related to damping by the $\Sigma N_r$ inlet tubes, which did not affect the $CO_2/H_2O$ open-path measurements. It also appears that the damping (difference of cospectra in the high-frequency range) at BOG is higher than the one at FOR for the selected averaging interval.



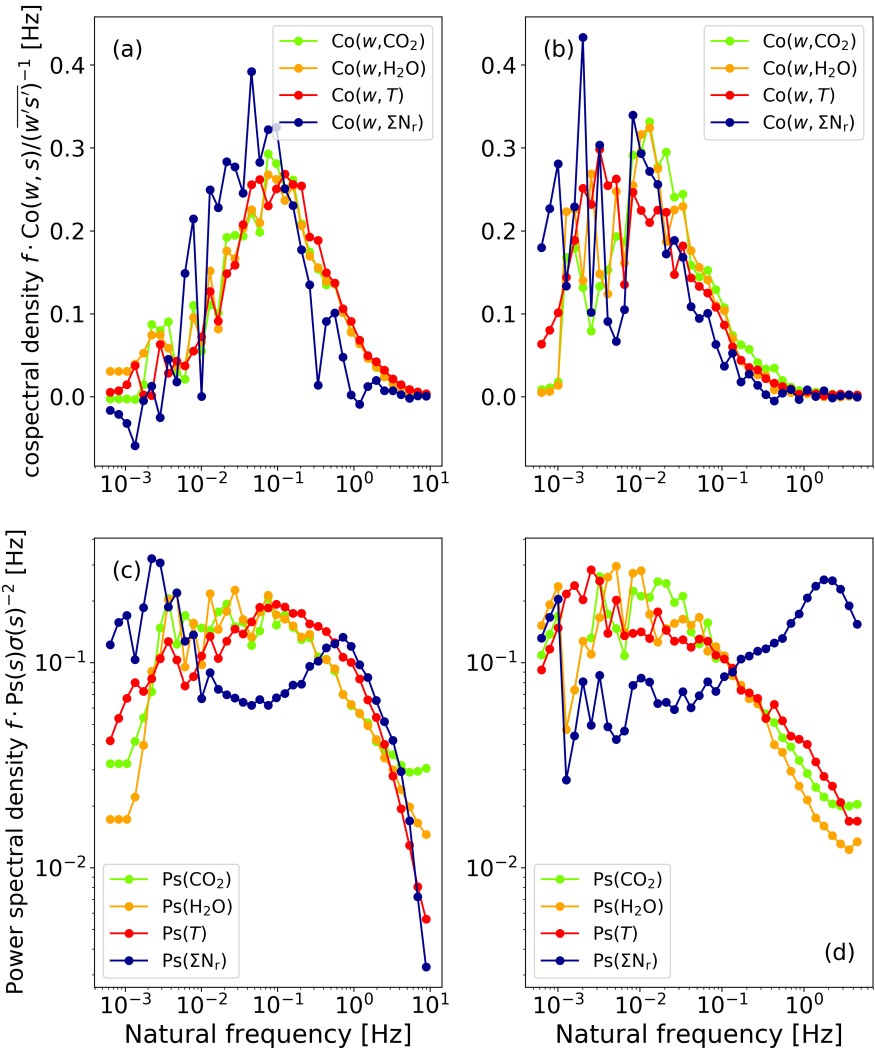

**Figure 2.** Normalised cospectra and power spectra of $T$ (red), $\Sigma N_r$ (blue), $CO_2$ (green) and $H_2O$ (orange) for Bourtanger Moor (a and c) and Bavarian Forest (b and d) measurement site. (Co)spectra were averaged from 11.10.2012 09:00 to 11.10.2012 16:30 ($\zeta$=-0.31, $\bar{u}$=1.36 ms$^{-1}$) at BOG and from 16.10.2016 10:00 to 16.10.2016 15:30 ($\zeta$=-3.27, $\bar{u}$=1.89 ms$^{-1}$) at FOR. $CO_2$ and $H_2O$ (co)spectra of BOG were adjusted to the aerodynamic measurement height of the $\Sigma N_r$ setup.

The shapes of the power spectra for $T$, $CO_2$ and $H_2O$ are comparable to those found in other studies (e.g., Ammann, 1999; Ibrom et al., 2007; Rummel et al., 2002; Aubinet et al., 2012; Ferrara et al., 2012; Fratini et al., 2012; Min et al., 2014). For Ps($T$) a slope of -0.62 (BOG) and -0.63 (FOR) was determined. Differences to the theoretical shape, -2/3 for power spectra, may be related to slight damping of Ps($T$) in the high-frequency range. The stronger drop of Co($w,\Sigma N_r$) compared to Co($w,CO_2$) and Co($w,H_2O$) in the high-frequency range is likely related to damping by the tubes, which is not relevant





for open-path instruments. $Ps(CO_2)$ and $Ps(H_2O)$ have nearly the same slope in the inertial subrange and exhibit the excepted shape. In contrast, $Ps(\Sigma N_r)$ is lower than $Ps(CO_2)$ and $Ps(H_2O)$ at lower frequencies ($< 0.1\,Hz$) and starts to rise afterwards.

A maximum is reached around $1\,Hz$. This phenomenon was found in almost all $Ps(\Sigma N_r)$ at the measurement sites. We further estimated the slope of $Ps(\Sigma N_r)$ in the high-frequency range. We applied a variance filter of $w$, $T$, $\Sigma N_r$ and excluded PS if the variance was higher than 1.96 standard deviation which corresponds to confidence limit of 95%. Additionally we excluded low-quality fluxes (flag=2) of sensible heat and $\Sigma N_r$ after Mauder and Foken (2004). Figure 3 shows a distribution of the estimated slopes at both measurement sites.

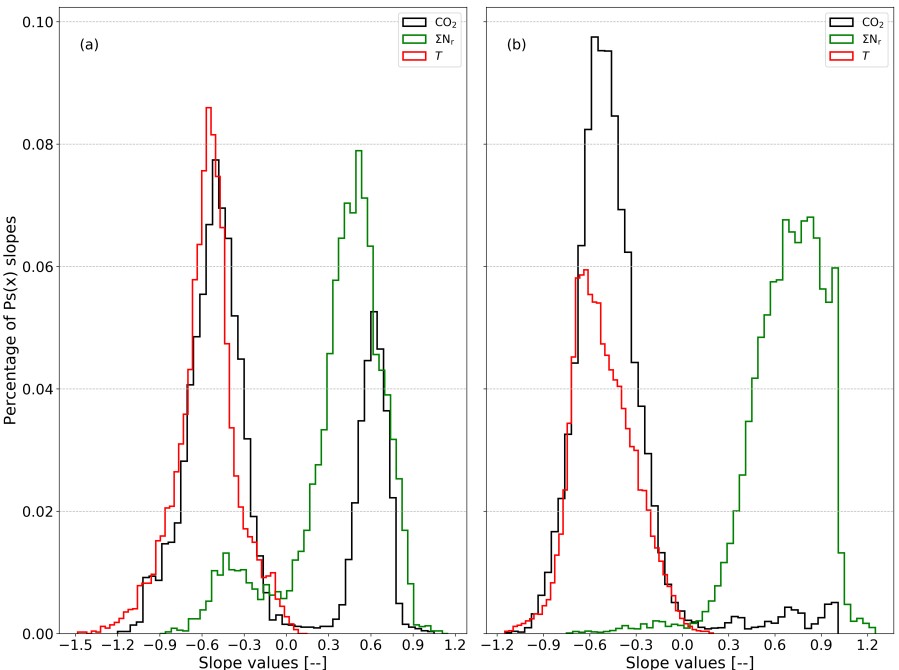

**Figure 3.** Distribution of spectral slopes in the high-frequency range ($> 0.1\,Hz$) of $Ps(\Sigma N_r)$ (green), $Ps(CO_2)$ (black) and $Ps(T)$ (red) for Bourtanger Moor (a) and for the Bavarian Forest (b). Slopes were estimated for half hourly power spectra from 02.10.2012 to 17.07.2013 and from 01.06.2016 to 28.06.2018 at BOG and FOR, respectively.

The slope of $Ps(T)$ is between -0.5 and -0.7, which is close to the theoretical value and the shape of the histogram seems to be narrower around the theoretical value at BOG than at FOR. In contrast, the slope of $Ps(\Sigma N_r)$ was mostly positive at both sites (87% at BOG and 98% at FOR). At BOG a second maximum is observed at around -0.5. The amount of $Ps(\Sigma N_r)$ slopes around -2/3 is rather small at BOG and even negligible at FOR. A positive slope for nearly all Ps of a certain trace gas is rather unexpected. Therefore we did a similar analysis for $Ps(CO_2)$. We used equivalent filtering criteria (see above) and additionally

applied a precipitation filter due to the open-path characteristics of the LI-7500. In general, most slopes of $Ps(CO_2)$ are negative in contrast to $Ps(\Sigma N_r)$. At BOG 70% of $Ps(CO_2)$ show a negative slope and at FOR nearly all slopes of $Ps(CO_2)$ are negative (95%). Furthermore, slopes of $Ps(CO_2)$ are closer to the slopes of $Ps(T)$ than $Ps(\Sigma N_r)$, but their maximum is slightly higher





than -2/3 (-0.53 for BOG and -0.58 for FOR). The shape of Ps($CO_2$) coincides well with the shape of Ps($T$) at both sites, but the agreement of Ps($CO_2$) with Ps($T$) was slightly better at the forest site. Positive slopes of Ps($CO_2$) were detected at both sites. More Ps($CO_2$) of BOG exhibit a positive slope between 0.50 and 0.75 (24%) than the Ps($CO_2$) of FOR (2%) in the same range.

### 3.2 Comparison of different damping correction methods

In the following, we present the results of the damping correction methods introduced in Sec. 2.3. Firstly, we describe the results of the in-situ power spectral method (IPS) and the four cospectral methods. Secondly, we demonstrate findings of dependencies on meteorological variables. Figures 4 and 5 show a time series of $\alpha$ which were calculated by each method on monthly (BOG) or bimonthly (FOR) basis depicted as box plots. It was possible to estimate $\alpha$ with all methods for 816 half-hours for BOG and 811 half-hours for FOR. All damping correction methods were evaluated for the same half-hours.

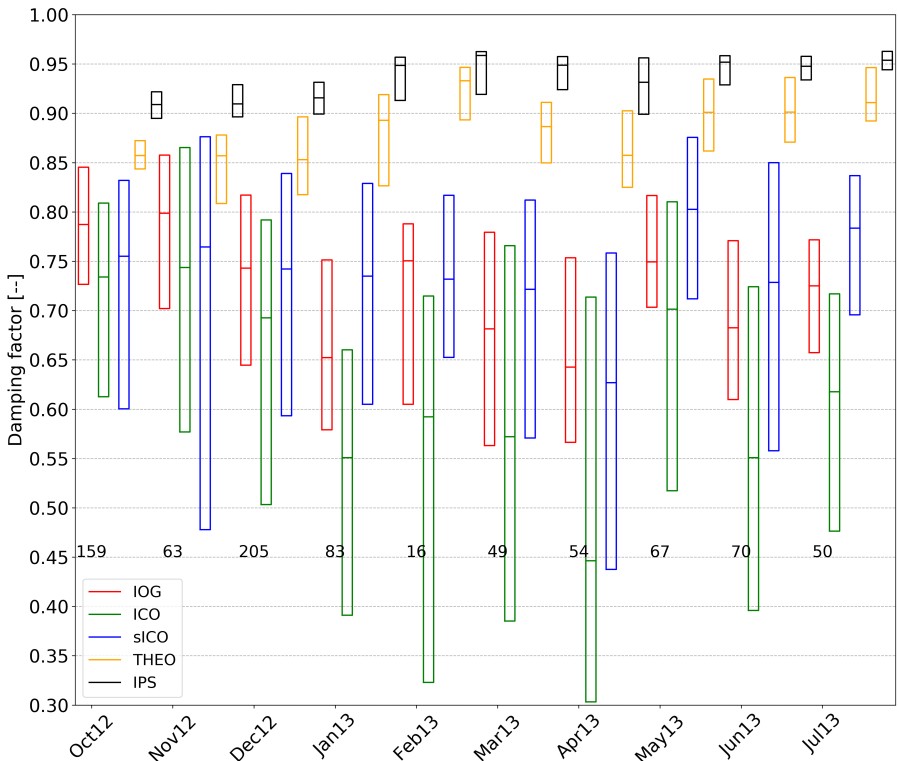

**Figure 4.** Boxplots of the flux damping factor ($\alpha$) for BOG without whiskers and outliers (box frame = 25 % to 75 % interquartile range (IQR), bold line = median). The number of observations which are displayed in the plot are the same for every method.





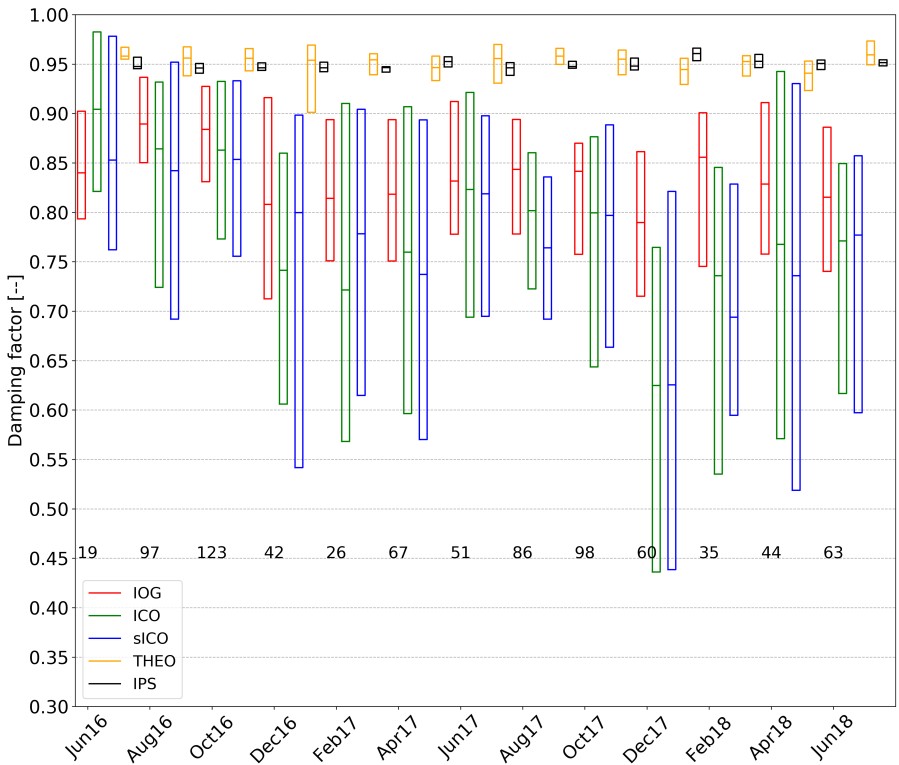

**Figure 5.** Boxplots of the flux damping factor ($\alpha$) for FOR without whiskers and outliers (box frame = 25 % to 75 % interquartile range (IQR), bold line = median). The number of observations which are displayed in the plot are the same for every method.

Monthly $\alpha$ calculated with the IPS method show no temporal drift at FOR (Fig. 5). The median $\alpha$ is around 95% for nearly every month. Additionally, the interquartile range (IQR; 25 to 75 %-quartile) is very small (1 to 2%). At BOG monthly median

$\alpha$ calculated with IPS were also mostly around 95%, only the first three month were sightly lower by $\sim 4$%. Their (IQR) is around 4% on average. It is obvious that $\alpha$ of IPS is the highest compared to the cospectral methods and they exhibit the lowest IQR during the measurement period.

At both sites the median $\alpha$ of the in-situ cospectral methods ICO, sICO and IOG show only moderate temporal variations during the whole measurement campaigns. While slightly higher values in summer and lower values in winter were found at

the FOR site (Fig. 5), the opposite pattern was observed at the BOG site (Fig. 4). Their IQR is more variable and ranges from 0.13 to 0.26 at BOG and from 0.16 to 0.31 at FOR. Changes in the range of the IQR and fluctuations of the medians may be related to different meteorological conditions, to changes in composition of $\Sigma N_r$ or to a degeneration of instrumental response. During field visits for maintenance, parts of the TRANC like the heating tube or platinum gaze were exchanged or cleaned, which could influence the results. At both sites $\alpha$ by THEO were always higher than in-situ cospectral methods (IOG, ICO,

sICO) and their medians were about 90% at BOG and about 95% at FOR. Their IQR is smaller than IOG, ICO and sICO, too.





At FOR the median $\alpha$ of ICO and sICO are similar for every month showing a difference of 0.03 on average and their IQR cover mostly the same range (Table 2 and Fig. 5). Values for $\alpha$ by IOG are mostly higher and exhibit a difference of 0.06 on average to sICO and ICO. The IQR by IOG is roughly half of the IQR of ICO and sICO (Table 2). During the months of December in 2016 and 2017 as well as January in 2017 and 2018 and April to May in 2018 IQR of ICO and sICO is relatively

large. These periods have in common that the average vertical wind was quite low in January 2017 and 2018 (less than $0.01\,\mathrm{ms^{-1}}$). Additionally, we had some instrumental performance problems (exchange of the pump and heating tube, power failure) with the TRANC in the mentioned months which has an influence on the quality of the cospectra/ogives. Consequently, IOG, ICO and sICO exhibit a wide IQR from 15 to 40% and differences in the median from 6 to 16% which could be related to the low number of valid cospectra/ogives. Therefore classifying $\alpha$ at FOR bimonthly (Fig. 5) was a needed approach to

enhance the quality when the amount of valid cospectra is not enough for a robust estimation of $\alpha$. Overall a good agreement of IOG, ICO and sICO was found.

At BOG the median $\alpha$ of ICO are the lowest and the median $\alpha$ of sICO and IOG are nearly the same for every month (Table 2 and Fig. 4). The difference of ICO to IOG varies by 5% and 20% and to sICO by 2% and 18%. A systematic difference in $\alpha$ between ICO and sICO was not observed for FOR. At the beginning of the measurements the difference was rather small,

but it started increasing after December 2012. The range of the quartiles is similar for IOG and sICO for certain months (see Table 2 and Fig. 4), but their IQR is lower than the IQR of ICO. Again the IQR of IOG is roughly half of ICO IQR. It seems that theoretical cospectra could not reproduce the shape of $\mathrm{Co}(w,T)$ well under certain site conditions, although $\tau_\mathrm{r}$ of sICO and ICO were quite similar. They show a correlation of 0.75 and an average absolute difference of 0.48. Comparing $\alpha$ between the sites shows that the damping is stronger at BOG than at FOR. Table 2 shows the averaged $\alpha$ at FOR and BOG. In total

we got a loss of 16-22% at FOR and about 26-38% at BOG which is much more than the damping estimated by THEO and IPS. These values are in common with other EC studies done on $\Sigma\mathrm{N_r}$) and other reactive nitrogen compounds (Ammann et al., 2012; Ferrara et al., 2012; Brümmer et al., 2013; Stella et al., 2013; Zöll et al., 2016).





**Table 2.** Averages of monthly medians, lower and upper quartiles of $\alpha$ over the whole measurement period for all applied methods at both sites.

| Site | method | median | lower quartile | upper quartile |
|---|---|---|---|---|
| Bavarian Forest | IOG | 0.84 | 0.77 | 0.90 |
| | ICO | 0.78 | 0.64 | 0.89 |
| | sICO | 0.78 | 0.63 | 0.89 |
| | THEO | 0.95 | 0.93 | 0.96 |
| | IPS | 0.95 | 0.94 | 0.95 |
| Bourtanger Moor | IOG | 0.72 | 0.64 | 0.80 |
| | ICO | 0.62 | 0.45 | 0.76 |
| | sICO | 0.74 | 0.59 | 0.83 |
| | THEO | 0.88 | 0.85 | 0.91 |
| | IPS | 0.94 | 0.91 | 0.95 |

The IQR is comparable for ICO and sICO at both sites. It ranges from 0.24 to 0.31, which is caused by the large variety in the shape of measured cospectra. The IQR of IOG is roughly half as large and ranges from 0.13 to 0.16. The IQR of THEO is rather small (0.02 at FOR and 0.07 at BOG). Flux loss and its IQR estimated by IPS are the lowest of all methods. For investigating deviations of the different methods more precisely, we computed correlation, bias and the precision as the standard deviation of the difference between two methods. The results are summarized in Table B1. IOG exhibits a bias of not more than 10% to ICO and sICO and is rather small at BOG (3%). The bias and precision between sICO and ICO is lowest at FOR. Additionally, the scattering of sICO $\alpha$ is more pronounced, which results in a lower precision of sICO against the IOG. Both sites have in common that the correlation of IOG with sICO was inferior to ICO. Checking ICO $\alpha$ against sICO $\alpha$ demonstrates a high correlation at both sites (0.78 for FOR and 0.66 for BOG). This is excepted, since theoretical cospectra are based on $Co(w, T)$. IOG, ICO and sICO show a strong bias, low precision and nearly no correlation to THEO. The correlation between the sICO with THEO is somewhat higher because of utilizing Kaimal cospectra for both methods. IPS shows a negative bias and high precision to IOG, ICO and sICO at FOR and 0.05 larger bias than THEO at BOG. The correlation of IPS with THEO is quite high at both sites which is reasonable, since bias and precision are quite low. Both methods give almost equal $\alpha$.

For investigating a trend in meteorological variables such as temperature, relative humidity, stability and wind speed, we classified them into bins, calculated $\alpha$ for each bin and display them as box plots (Fig. 6). In the following figure only wind speed and stability are shown. These are two variables for which we except a dependence, since the shape and position of a Kaimal cospectrum varies with wind speed and stability. We checked for dependencies on the other variables such as global radiation, temperature and humidity, but no significant influence was found.





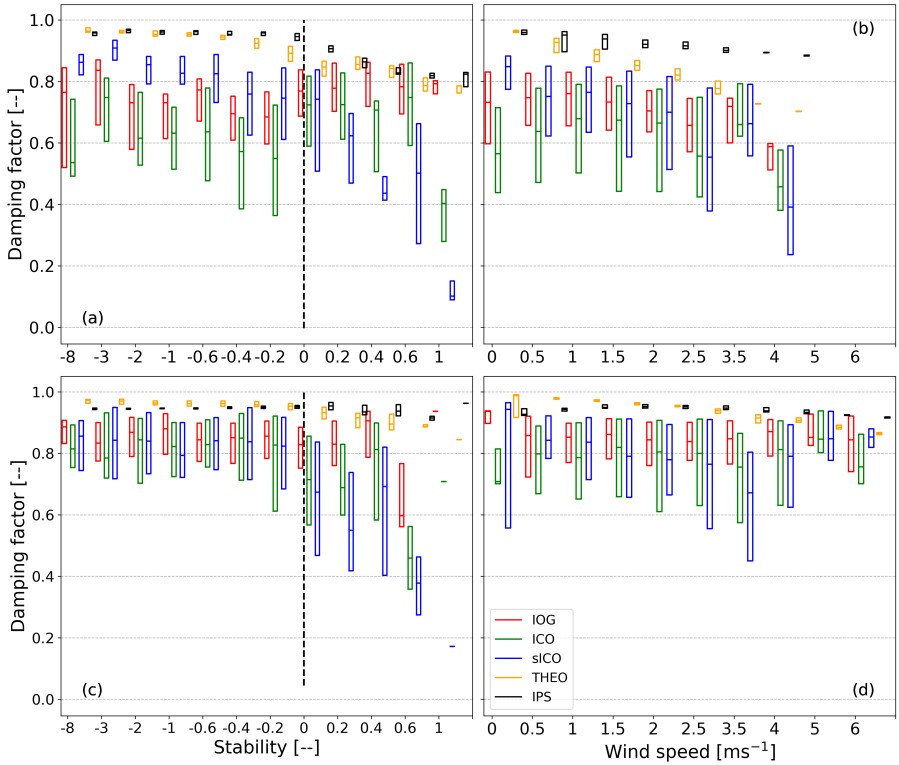

**Figure 6.** Dependency of the flux damping factor ($\alpha$) on stability and wind speed classes as box plots without whiskers and outliers (box frame = 25 % to 75 % interquartile range (IQR), bold line = median). Each damping estimation method is assigned with a different color (red: IOG, green: ICO, blue: sICO, orange: THEO, black: IPS). (a) and (b) refer to the BOG site and (c) and (d) to the FOR site.

A slight dependence on wind speed for BOG $\alpha$ is starting to be relevant at wind speeds above $1\,\mathrm{ms}^{-1}$, which is confirmed by IOG, ICO and sICO. The influence on wind speed predicted by THEO begins already at low wind speed, which means that stronger damping was found at higher wind speed values. It shows a (linear) decrease from the beginning. A bias to IOG, ICO and sICO (Table B1) exists for all wind speed classes. Considering the medians we observed an increase in attenuation

from 15% till 20% over the whole wind speed regime. The bias with sICO (Table B1) is mostly visible for wind speeds up to $1.5\,\mathrm{ms}^{-1}$ and gets negligible afterwards. Values of $\alpha$ at FOR were nearly invariant to changes in wind speed. The bias with THEO diminished for wind speed larger than $4\,\mathrm{ms}^{-1}$. The predicted drop due to wind speed by THEO is roughly 10%. IPS shows the weakest $\alpha$ for all wind speed classes at both sites. The decrease of $\alpha$ with wind speed is less than 10% at BOG and hence lower than the cospectral methods. IPS exhibit no significant drop in $\alpha$ with wind speed at FOR.

Values of $\alpha$ estimated by THEO are almost equal for unstable conditions and decline for stable situations. As before the theoretical drop in attenuation is stronger at BOG (up to 20%) than at FOR (not exceeding 10%). The FOR $\alpha$ of IOG, ICO and sICO are nearly equal ($\sim 0.85$) for unstable cases. During stable situations IOG and ICO exhibit no distinct trend and their IQR is similar. The IQR of sICO gets wider and their values are lower than the other methods. No significant trend is observed. The





linear decline is given for BOG $\alpha$, too, but does not exists for IOG and ICO. The $\alpha$ are similar for unstable cases, but showing

no decrease with increasing stability. Again the IQR of the sICO increases for positive stability and is smaller than IOG and ICO for negative values. The bias to IOG and ICO is obvious for the negative values. Similar to THEO, IPS shows a drop of $\alpha$ with increasing stability at BOG, but values are higher than for the cospectral methods. As observed for wind speed at FOR no significant drop in $\alpha$ for IPS occurs under stable conditions.

### 3.3  Analysis of response time

After comparing $\alpha$ of the individual methods we focus on variation of $\tau_\mathrm{r}$ in time. Therefore we show a time series of $\tau_\mathrm{r}$ of both measurement sites. Figure 7 shows a time series of $\tau_\mathrm{r}$ which were calculated by ICO on bimonthly basis depicted as box plots.

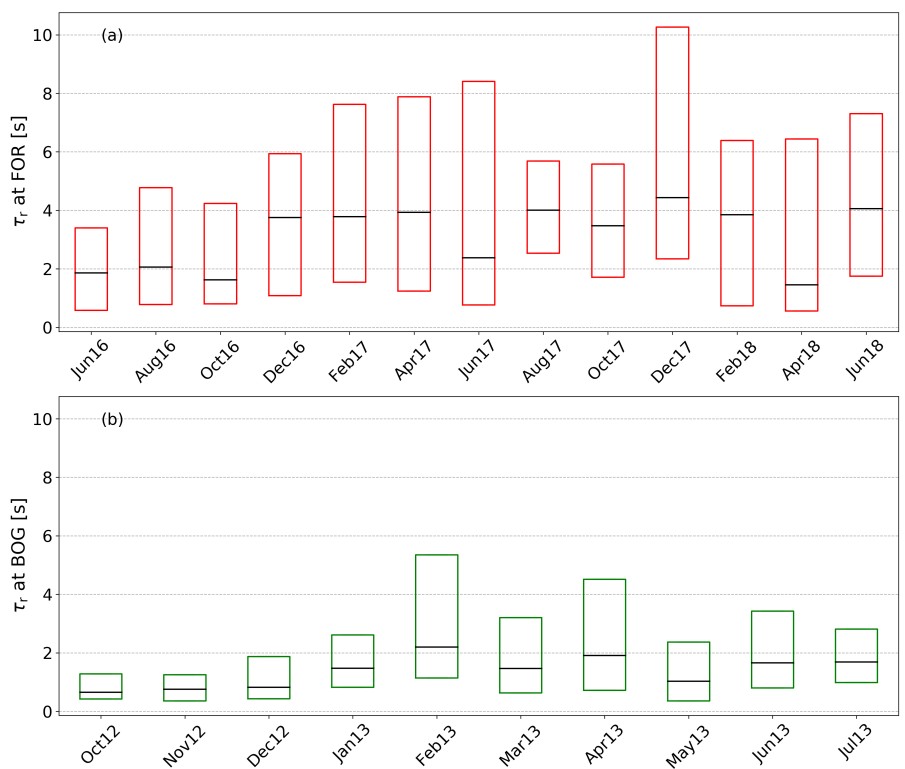

**Figure 7.** Time series of the response time ($\tau_\mathrm{r}$) depicted as box plot without whiskers and outliers (box frame = 25 % to 75 % interquartile range (IQR), bold line = median) for the FOR site (a) and the BOG site (b).

It is obvious that medians of $\tau_\mathrm{r}$ of FOR are generally larger than medians of $\tau_\mathrm{r}$ of BOG. The averaged median $\tau_\mathrm{r}$ is 1.37 s for BOG and 3.13 s for FOR (Table B2). Both sites have in common that $\tau_\mathrm{r}$ was sightly lower at the start of the measurements and their medians were quite constant until December 2012 at BOG and October/November 2016 at FOR. Afterwards $\tau_\mathrm{r}$ and its

IQR increased significantly, especially at FOR. The variation of $\tau_\mathrm{r}$ follows no trend and seems to be rather random. The IQR





of FOR was larger indicating that scattering of $\tau_r$ was enhanced at FOR. On average $\tau_r$ increased from 0.74 s to 1.63 s at BOG and from 1.85 s to 3.51 s at FOR on average (Table B2).

## 4 Discussion

### 4.1 Noise effects on power spectra and cospectra

Measured fluxes of $\Sigma N_r$ are heavily affected by white and red noise. They are caused by low and non-stationary ambient trace gas concentrations and fluxes, typically low fluxes due to weak sources and inhomogenously distributed N sources, limited resolution and precision of the CLD and varying proportions of different $N_r$ compounds. This leads to a high rejection rate of cospectra and power spectra during quality screening and is challenging for every spectral anaylsis using in-situ measurements. While the influence on cospectra is mainly limited to low-frequency range, power spectra show systematic deviations in the

low and high frequency range. The positive slope (Fig. 2 and Fig. 3) is related to white noise which compromise the $Ps(\Sigma N_r)$ in the high-frequency domain. White and red noise are more present at FOR, because the site was located in a remote area with no nearby sources of $\Sigma N_r$. At BOG white noise is weaker since more sources of $\Sigma N_r$ were next to the EC station. The disturbance due to red noise is also visible in Fig. 2. The variability (scattering) of cospectra and power spectra is more pronounced at FOR than at BOG as visible in the shown example. Some $Ps(\Sigma N_r)$ in Fig. 3, mainly at BOG, show a slope near the theoretical

value of -2/3 and were not affected by white noise. Therefore we examined the environmental conditions such as wind speed, friction velocity, concentration and flux values at that site during half-hours which were attributed to slopes less than -0.25 and compared them to half-hours with a slope greater than -0.25. Only the distribution of concentration was different for the two regimes: Most $Ps(\Sigma N_r)$ with a slope less than -0.25 were associated with concentration values between 25 ppb and 40 ppb, whereas $Ps(\Sigma N_r)$ with a slope greater than -0.25 were associated with concentration values between 10 ppb and 25 pp which is

in common with the background concentration level of $\Sigma N_r$ at BOG. It was about 21 ppb, whereas only 5 ppb on average were measured at FOR. Thus it seems that the concentration is an important factor for regulating the quality of $Ps(\Sigma N_r)$. The slope of $Ps(T)$ shows a clear peak between -0.5 and -0.7 for both sites, which is close to the theoretical value of -2/3. The differences in the distribution may be related to different site characteristics like surface roughness length, inhomogenous canopy height, turbulence or to large-scale eddies which gain more influence on the fluxes at higher aerodynamic measurement height. In

contrast to $\Sigma N_r$, noise was more present in $Ps(CO_2)$ at BOG than at FOR, although the site is located in a remote area. The stronger influence of white noise to the $Ps(CO_2)$ measured at BOG could be induced by reduced sensitivity of the open-path instrument due to humidity. During the whole campaign relative humidity was always on a high level. Such humid conditions could reduce the sensitivity of the sensor and introduce noise in power spectra. Above the forest the air was less humid and consequentially, less $Ps(CO_2)$ were affected by white noise. Removing high-frequency variations which consist mainly of white

noise is easier for $Ps(CO_2)$ because their signal is higher than those of $Ps(\Sigma N_r)$ in the low-frequency domain and the observed noise is limited to high frequencies ($> 2$ to $5$ Hz). Additionally, the noise is strictly linear and exhibits no parabolic structure such as $Ps(\Sigma N_r)$ (Fig. 2). The drop of $Ps(\Sigma N_r)$ in the highest-frequency range is unexpected. It may be caused by uncorrelated noise which is induced by some components of the setup like pump, logger or other electrical components. This uncorrelated





noise occurred mostly around $1\,\text{Hz}$ and decreased towards the highest frequencies. Handling the impact of unknown noise on
power spectra may not be completely possible for IPS. Wolfe et al. (2018) installed an EC setup in an aircraft and measured
$CO_2$, $H_2O$ and $CH_4$ with Los Gatos Research analyzers and $H_2O$ with an open-path infrared absorption spectrometer. They
found a slope of $\sim 1$ in $Ps(CO_2)$, $Ps(H_2O)$ and $PS(CH_4)$ above $0.4\,\text{Hz}$, but not in the $Ps(H_2O)$ of the open-path analyzer.
They concluded that the white noise was related to insufficient precision of the closed-path analyzers at higher frequencies.
No white noise was detected in the corresponding cospectra, because it does not correlate with $w$. Kondo and Tsukamoto
(2007) did $CO_2$ flux measurements above the Equatorial Indian Ocean. They concluded that white noise was related to a lack
of sensitivity to small $CO_2$ density fluctuations. Density fluctuations of $CO_2$ above open ocean surfaces are much smaller than
over vegetation. Similar to the present study, they detected no white noise in their $Co(w, CO_2)$. Their site characteristics and
related low fluctuations of trace gas are comparable to the forest site. The latter was located in a remote area and therefore far
away from potential (anthropogenic) nitrogen scoures. This led to low concentrations and less variability in concentrations and
deposition fluxes. Very small fluctuations of $\Sigma N_r$ are probably not detectable by the instrument. This is further confirmed by the
time lag analysis we did before flux estimation. The broad shape of the empirical lag distribution around the physical lag (not
shown) and the random time lag scattering demonstrated that most of the fluxes were near or below the detection limit and thus
quality of (co)spectra suffered from noise. Additionally physical reasons such as an inhomogeneous surface roughness length
and canopy height in the footprint of the tower and different range of relevant eddy sizes may have been reasons for fewer
valid high-quality (co)spectra compared to the BOG site. The findings indicate that using Ps for estimating correction factors
of gases with low turbulent fluctuations, which are measured by a closed-path instrument, can be problematic. Therefore we
recommend using cospectra to estimate $\tau_r$ and $\alpha$ of reactive gases which have less variability in concentrations and deposition
fluxes. White noise was observed in power spectra of $CO_2$ and $H_2O$, too. Both gases were measured with an open-path analyzer,
but their concentrations are higher and the variability in concentrations of these gases is much larger than for $\Sigma N_r$. It indicates
that $Ps(CO_2)$ are clearly less affected by white noise and the instrument is able to capture the high-frequent variability of $CO_2$
well. The assumption of spectral similarity was valid for $Ps(CO_2)$, but was not fullfilled for $Ps(\Sigma N_r)$ due to the influence of
red and white noise. Consequentially, an optimization fit with an infinite impulse response function gives unrealistic results
for $\tau_r$. Most likely automatic filtering criteria are not sufficient enough to extract good quality (co)spectra of $\Sigma N_r$ efficiently
and thereby the averaged $Ps(\Sigma N_r)$ used for fitting procedure is dominated by low quality and invalid cases. However, using
more restrictive quality selection criteria or narrowing the frequency range for the fitting of the transfer function produced
rapidly changing values or even negative values for $\tau_r$. This demonstrates that the estimation of $\tau_r$ with $Ps(\Sigma N_r)$ via IPS is
very uncertain and the number of $Ps(\Sigma N_r)$ with sufficient quality was not high enough for a robust fitting. Therefore, IPS is
likely to be inappropriate for correcting flux measurement of trace gases with a high white and/or red noise level. The number
of good quality (co)spectra for $CO_2$ and $H_2O$ was at least one order of magnitude higher than for $\Sigma N_r$. Monthly averaged $\alpha$
for $CO_2$ and $H_2O$ by IPS were in the range of 5% and 10% which is quite reasonable for and open-path instrument and in
agreement with studies dealing with the same instrument (Burba et al., 2010; Butterworth and Else, 2018).



## 4.2 Assessment of cospectral approaches

### 4.2.1 THEO vs. (semi-)empirical approaches

In general, $\alpha$ values determined by the (semi-) empirical cospectral methods (sICO, ICO and IOG) were considerably lower
than the results of THEO. The difference indicates a strong additional damping effect whose impact on $\Sigma N_r$ fluxes is not
detected by the fluid dynamics related transfer functions used in THEO. This additional damping must be caused by adsorption
processes at the inner surfaces of the inlet system, for example in the converter, the sample lines or the CLD. Studies from
Aubinet et al. (1999); Bernhofer et al. (2003); Ammann et al. (2006); Spank and Bernhofer (2008) also have shown that
the damping factor by the THEO approach is often too high. Beside disregarded damping processes, this could have also
been caused by deviations of the site specific cospectra from theoretical cases. Therefore it is advisable to apply empirical
methods to measurements of gases with unknown properties or to setups and instrument devices with flux loss sources which
are difficult to quantify. Empirical methods take the sum of all potential flux losses into account and do not take care of an
individual or specific flux loss. The difference between THEO and empirical methods in total flux losses at the two study sites
can be explained by the different aerodynamic measurement height. With increasing measurement height turbulence cospectra
are shifted to lower frequencies (Fig. 1 and Fig. 2) and hence a weaker high-frequency damping is expected. Vertical sensor
separation was not considered by the spectral transfer function in the THEO approach. However, the impact of vertical sensor
separation on the flux loss is very low if the gas analyzer is placed below the anemometer as in the present study. Kristensen
et al. (1997) determined a flux loss of only 2% at the vertical separation of $20\,cm$ and measurement height of $1\,m$. This effect
gets even smaller with increasing measurement height. Beside the measurement height, also the wind speed and stability are
expected to have an influence on the position and shape of the cospectrum and thus on the damping factor. Yet, no clear
systematic dependencies of (s)ICO and IOG results on these parameters were found. The dependency on wind speed at BOG is
only valid for high wind speed classes. Values of $\alpha$ at BOG appear to be invariant to changes in stability, whereas $\alpha$ at FOR are
quite constant at unstable conditions. In constrast, sICO follows the expected drop at stable conditions as observed for sICO
or THEO. The reason for the difference between sICO and ICO is discussed in Sec. 4.2.2. There could be other effects which
superpose the wind speed and stability dependencies, which are relevant for (chemical) damping processes, e.g. humidity and
$\Sigma N_r$ concentration could influence the aging of the tubes and consequentially the adsorption at inner tube walls. However, we
found no dependency on these parameters.

$\Sigma N_r$ is a complex trace gas signal since it consists of many reactive N gases which have various reaction pathways and
concentrations of the single compounds are unknown. As shown by Hurkuck et al. (2014) $N_r$ concentrations at BOG were
relatively high and showed a distinct diurnal cycle due to intensive livestock and crop production in the surrounding region
which is in contrast to the situation at the remote FOR site. There were almost no nearby anthropogenic sources (Zöll et al.,
2019) and hence much lower concentration (Beudert and Breit, 2010) and a weaker daily cycle were observed. This has
an influence on the variability of (co)spectra and strengthens their susceptibility to noise. It could make them invariant to
wind speed and stability. Due to the measurement of the sum of individual $N_r$ compounds we can only roughly estimate the
contribution of individual species to the flux and its high frequency loss. Hurkuck et al. (2014) and Zöll et al. (2016) detected





strong $NH_3$ deposition at BOG accounting for more than 80% of deposited $\Sigma N_r$. Therefore, mostly $NH_3$ seems to influence the damping of $\Sigma N_r$. According to DELTA-Denuder measurements presented in Zöll et al. (2019), $NH_3$ concentrations were relatively low at FOR site (Beudert and Breit, 2010). 33% of $\Sigma N_r$ were $NH_3$ and 32% were attributed to $NO_2$. However, both species are still the main $\Sigma N_r$ flux contributors, thereby holding an important role for the detected flux loss at the forest site.

All in all, a general or site-specific parametrisation of the damping as a function of environmental parameters was not possible. Since half-hourly $\alpha$ of the empirical methods vary with time and due to the limited amount of high-quality $\Sigma N_r$ cospectra it is advisable to use averages over certain time periods. Therefore we decided to use monthly median values for correcting fluxes at BOG. A bimonthly classification was conducted for FOR because the rejection rate was higher due to higher uncertainty of cospectra in the lower frequency range. For estimating $\alpha$ a reliable determination of $\tau_r$ is needed. Using a constant $\tau_r$ is

possible but not recommended for our $\Sigma N_r$ setup since $\tau_r$ varied with time and started to increase after a few month. It seems that the variation of $\alpha$ in time was mainly driven by the change of $\tau_r$. The increase of $\tau_r$ and the enhanced variation of $\tau_r$ after a few month could be related to instrumental performance problems caused by an aging of tubes and filters, reducing pump performance, problems with the CO supply and TRANC temperature or a sensitivity loss of the CLD.

### 4.2.2 ICO vs. sICO approach

The difference between the ICO and sICO method is the usage of Kaimal cospectra for determining $\alpha$ after equation (1). One reason for using theoretical cospectra is that it lowers the computation time for estimation of $\alpha$. Moreover, due to site or experimental setup related reasons the $Co(w,T)$ may be influenced by noise in the low-frequency range which compromises the determination of $\alpha$. In such cases using Kaimal cospectra can be good alternative. The usage of standard Kaimal cospectra leads to a loss of site specific information. Differences to measured $Co(w,T)$ can lead to uncertainties in the damping esti-

mation of sICO. The consequence is an observed bias of unstable $\alpha$ between sICO and ICO at BOG (Fig. 6) or induced wind speed and stability dependencies by the usage of Kaimal cospectra which are not confirmed by ICO or IOG. Mamadou et al. (2016) computed $\alpha$ of $CO_2$ with locally measured cospectra and Kansas cospectra (Kaimal et al., 1972), which are slightly different from the theoretical cospectra used in this study. They found that theoretical and measured $Co(w,T)$ differ significantly in shape which resulted in large differences of correction factors during stable conditions, although their investigated

site exhibited no complex terrain or vegetation. It led to an overestimation of nighttime fluxes of 14-28% if Kansas cospectra were used. Therefore, we selected $\alpha$ of ICO and sICO estimated at stable conditions during day and nighttime. Comparing stable ($\zeta > 0.05$), nighttime/dawn ($R_g < 20\,\mathrm{W m^{-2}}$) with stable, daytime half-hourly $\alpha$ showed that stable, nighttime $\alpha$ had a higher variability and were mostly overestimated by 0.14-0.35, whereas stable, daytime $\alpha$ were overestimated by 0.1-0.2. Some $\alpha$ were underestimated, but the discrepancy was about 0.15 on average. Using Kaimal cospectra can be problematic

for estimating $\alpha$ under stable conditions. If typical wind speed or stability dependencies are not approved by other cospectral methods, we do not recommend the usage of theoretical methods such as Kaimal cospectra since it may lead to a bias or unproven dependency.





### 4.2.3 ICO vs. IOG approach

The main difference between ICO and the IOG method is that IOG utilizes the low-frequency part and (s)ICO the high-
frequency part of the cospectrum. The low-frequency part is much more variable than the high-frequency one, especially on
half-hourly basis. As a consequence the ratio between $Og(w, \Sigma N_r)$ and $Og(w, T)$ is often not well-defined in the fitting range
and hence the linear regression between $Og(w, \Sigma N_r)$ and $Og(w, T)$ gives erroneous results. Strong attenuation is possibly
underestimated by IOG because damping can extend into the fit range. IOG may perform better for averaged cospectra since
impact of scattering in the low-frequency part of the spectrum would be reduced. The variability (scattering) of cospectra in the
high-frequency part is comparatively small and differences in the decay of $Co(w, \Sigma N_r)$ and $Co(w, T)$ are easier to identify than
differences in the low-frequent part. The transfer function used in the ICO fitting routine has to consider the relevant damping
processes. While the transfer functions for physical damping effects are relatively well defined ((cf. Mamadou et al., 2016);
Table A1), chemical damping effects are rather unknown although they can be very important for reactive gases such as $NH_3$ or
$\Sigma N_r$. The empirical transfer function was chosen with regard to different response times of the individual sensors. Since both
sensors are first-order system filters the dynamic frequency response can be described by a the first-order filter transfer function
(A1). Additionally, the TRANC/CLD has a slower response than the sonic. The mismatch in the response times introduce a
phase shift in the time series which is accounted for by applying the phase shift mismatch function (Table A1) after Zeller et al.
(1988); Ammann (1999). The impact of phase shift mismatch is visible in the high-frequency range of $Co(w, \Sigma N_r)$ where it
leads to a steep decay and significant variations. Until now there is no ideal transfer function which can capture all damping
processes. The transfer function can differ depending on trace gas and site setup. Our empirical transfer function was chosen
especially for reactive gases such as $\Sigma N_r$ or $NH_3$ measured with a closed-path instrument. The usage of equation 3 for other
gases like $CO_2$ or $H_2O$ is not recommended without knowing any spectral characteristics. In case of $CO_2$ and $H_2O$ measured
with Li-7500 at FOR and BOG we have to modify equation 3. We would leave out the phase shift mismatch since the Li-7500
has a faster response and consider using the sensor separation and/or path averaging transfer function (Moore, 1986).

## 5 Conclusions


We investigated flux losses of total reactive nitrogen ($\Sigma N_r$) measured with a custom-built converter (TRANC) coupled to fast-
response CLD above a mixed forest and a semi-natural peatland. We compared five different methods for the quantification and
correction of high-frequency attenuation: the first is adapted from Moore (1986) (THEO), the second uses measured cospectra
of sensible heat and trace gas flux (ICO), the third uses response time calculated from measured cospectra and estimates
damping with modified Kaimal cospectra (Ammann, 1999) (sICO), the fourth uses the measured ogives (IOG) and the fifth
method is the power spectral method by Ibrom et al. (2007) (IPS). The flux losses by IPS for our closed-path eddy covariance
setups were around 6% at the peatland site (BOG) and around 5% at the forest site (FOR). The attenuation after THEO was
about 12% at BOG and about 5% at FOR. The methods using measured cospectra or ogives (ICO, sICO and IOG) showed a
flux loss of roughly 16-22% for the forest measurements and around 26-38% for the peatland measurements, with ICO showing
the strongest damping at both sites.





We found that $Ps(\Sigma N_r)$ were heavily affected by white and red noise. No robust estimation of the response time by using measured power spectra was possible. THEO could not capture strong damping processes of $\Sigma N_r$ fluxes which are likely caused by adsorption processes occurring at inner surfaces of the inlet system or missing information about the contribution of specific gases to $\Sigma N_r$. Consequently, THEO and IPS are not recommended for estimating reliable flux losses of $\Sigma N_r$.

Differences in flux losses are related to measurement height and hence to the variable contribution of small and large-scale eddies to the flux. No systematic or only partly significant dependencies of the empirical methods (ICO, sICO, and IOG) on parameters such as atmospheric stability and wind speed, which have an influence on the shape and position of cospectrum, were observed. Damping factors ($\alpha$) were found to be invariant to stability and only a minor dependence on high wind speed was determined at the peatland site. At the forest site, $\alpha$ values were quite constant at unstable conditions, but exhibited no

drop with increasing stability. We suppose that other factors like varying atmospheric concentration, distribution and strength of sources and sinks, enhanced chemical activity of $\Sigma N_r$ compared to $CO_2$ and $H_2O$ and vegetation could influence $\alpha$ stronger and may superpose slight effects of wind speed and stability. General or site-specific parameterisation of the damping was not possible.

The empirical methods perform well at both sites and median $\alpha$ are in the range of former studies about reactive nitrogen

compounds. However, we detected significant discrepancies to ICO which were related to site-specific problems or to using different frequency ranges of the cospectrum for the assessment. We discovered a bias between $\alpha$ computed with ICO and sICO for the BOG measurements. No significant bias for ICO and sICO was detected at the FOR site. We supposed that Kaimal cospectra may underestimate the attenuation of fluxes under certain site conditions (cf. Mamadou et al., 2016). Differences in $\alpha$ to IOG are induced by utilizing the low-frequency part of the cospectrum. The low-frequency part is more variable than

the high-frequency part on half-hourly basis. Strong attenuation cases could be underestimated by IOG since damping already occurs in the fit range.

Our investigation of different spectral correction methods showed that ICO is most suitable for capturing damping processes of $\Sigma N_r$. However, not all damping processes of reactive gases are fully understood yet and current correction methods have to be improved with regard to quality selection of cospectra. Power spectral and purely theoretical methods which are established

in flux calculation software worked well for inert gases, but are not suitable for reactive nitrogen compounds. Estimating damping of EC setups designed for highly reactive gases with an empirical method may be a considerable and reliable option. For further correction of fluxes we will use monthly median $\alpha$ since half-hourly values will lead to significant uncertainties in fluxes. Using a constant $\tau_r$ is not recommended as we noticed variation of $\tau_r$ with time, which is caused by altering the inlet system. Correcting fluxes after meteorologically classified $\alpha$ is possible if dependencies are exhibited by the EC setup.

*Data availability.* All data are available upon request from the first author of this study (pascal.wintjen@thuenen.de).





## Appendix A

### A1 Transfer functions of the $\Sigma N_r$ setup

Transfer functions used for validation of $\alpha$ after THEO, ICO and sICO are listed in Table A1. A detailed description is given in the mentioned literature. Table 1 contains physical parameters of the setup which are necessary to estimate $\alpha$.

**Table A1.** Transfer functions used for evaluation of the $\Sigma N_r$ damping factors.

| Transfer Function | physical parameters |
|---|---|
| first-order filter<br>$TF_R(f) = \dfrac{1}{\sqrt{1 + (2\pi\tau_r f)^2}}$ | response time $\tau_r$ ((Moore, 1986; Moncrieff et al., 1997)) |
| sensor separation<br>$TF_s(f) = \exp\left(-9.9(fd_s/u)^{1.5}\right)$ with $d_s = d_{sa}|\sin(\alpha_d)|$ | $u$ wind speed, effective lateral separation distance $d_s$ , measured separation distance $d_{sa}$, $\alpha_d$ angle between the line joining the sensors and windirection (Moore, 1986; Aubinet et al., 2012), |
| path averaging anemometer<br>$TF_w(f_p) = \dfrac{2}{\pi f_p}\left(1 + \dfrac{1}{2}\exp(-2\pi f_p) - 3\dfrac{1-\exp(-2\pi f_p)}{4\pi f_p}\right); f_p = \dfrac{fp_1}{u}$ | $p_1$ sonic path length ((Moore, 1986; Moncrieff et al., 1997; Aubinet et al., 2012)) |
| tube attenuation<br>$TF_{t,lam}(f) = \exp\left(-0.82\mathrm{Re}\mathrm{Sc}f_t^2\right)$ with $f_t = f \cdot (0.5DL)^{0.5}/v_t$ | $D$ Diameter of tube, $L$ length of tube, $\mathrm{Sc}$ Schmidt Number, $\mathrm{Re}$ Reynolds Number, $v_t$ flow speed inside the tube ((Ammann, 1999; Aubinet et al., 1999, 2012)) |
| phase shift mismatch<br>$TF_{\Delta R}(f) \approx \cos\left[\arctan\left(2\pi f\tau_r\right) - 2\pi f\tau_r\right]$ | $\tau_r$ response time ((Zeller et al., 1988; Ammann, 1999)) |

**A2   Kaimal cospectrum used in THEO and sICO**

The cospectrum for stable conditions after Ammann (1999) has the following form

$$\mathrm{Co}_{mod}(f, a, u) = \frac{f \cdot (a/u)}{0.284 \cdot (1 + 6.4 \cdot \zeta)^{0.75} + 9.345 \cdot (1 + 6.4 \cdot \zeta)^{-0.825} \cdot (f \cdot (a/u))^{2.1}} \tag{A1}$$





where $a$ is the aerodynamic measurement height and is given by the difference of measurement height $z$ and the zero-plane displacement height $d$ with $a = z - d$ (Spank and Bernhofer, 2008). $\zeta$ is the stability parameter and is defined by $\zeta = a/L$. $L$
is the Obukov-Length. The cospectrum for unstable conditions is determined by two parts

$$\text{Co}_{\text{mod}}(f,a,u) = \begin{cases} 12.92 \cdot f\,(a/u) \cdot (1 + 26.7 \cdot f\,(a/u))^{-1.375} & f\,(a/u) < 0.54 \\ 4.378 \cdot f\,(a/u) \cdot (1 + 3.8 \cdot f\,(a/u))^{-2.4} & f\,(a/u) \geq 0.54 \end{cases} \tag{A2}$$

## Appendix B

### B1    Results of different damping correction methods

**Table B1.** Result of the comparison between different damping determination methods at the two measurement sites. Bias ($\Delta$) is computed as averaged difference between $\alpha$. Precision is given as 1.96 standard deviation of the difference. $r$ is the correlation coefficient.

| method | Bavarian Forest | | | Bourtanger Moor | | |
|---|---|---|---|---|---|---|
| | $\Delta$ | $1.96\sigma$ | $r$ | $\Delta$ | $1.96\sigma$ | $r$ |
| ICO, IOG | -0.07 | 0.33 | 0.50 | -0.10 | 0.31 | 0.67 |
| ICO, sICO | 0.0 | 0.25 | 0.78 | -0.07 | 0.33 | 0.66 |
| ICO, THEO | -0.19 | 0.37 | 0.09 | -0.25 | 0.43 | -0.08 |
| ICO, IPS | -0.19 | 0.38 | -0.09 | -0.30 | 0.43 | -0.14 |
| | | | | | | |
| sICO, IOG | -0.07 | 0.36 | 0.36 | -0.03 | 0.36 | 0.42 |
| sICO, THEO | -0.20 | 0.33 | 0.22 | -0.18 | 0.37 | 0.36 |
| sICO, IPS | -0.20 | 0.37 | -0.05 | -0.23 | 0.38 | 0.38 |
| | | | | | | |
| IOG, THEO | -0.12 | 0.22 | 0.0 | -0.15 | 0.26 | 0.01 |
| IOG, IPS | -0.12 | 0.22 | -0.08 | -0.20 | 0.26 | -0.16 |
| | | | | | | |
| THEO,IPS | 0.0 | 0.05 | 0.47 | -0.05 | 0.07 | 0.70 |



## B2 Analysis of the response time estimated by ICO

**Table B2.** Median $\tau_r$ averaged over certain measurement periods at both sites.

| Site | time period | averaged $\tau_r$ [s] | lower quartile [s] | upper quartile [s] |
|------|-------------|----------------------|--------------------|--------------------|
| Bavarian Forest | Jun 2016 - Nov 2016 | 1.85 | 0.72 | 4.14 |
|  | Dec 2016 - Jun 2018 | 3.51 | 1.43 | 7.15 |
|  | whole period | 3.13 | 1.26 | 6.46 |
| Bourtanger Moor | Oct 2012 - Dec 2012 | 0.74 | 0.40 | 1.47 |
|  | Jan 2013 - Jul 2013 | 1.63 | 0.78 | 3.47 |
|  | whole period | 1.37 | 0.67 | 2.87 |

*Author contributions.* PW wrote the manuscript, carried out the measurements at the forest site and performed data analysis and interpretation. CA gave scientific advice. FS helped with coding and evaluated meteorological measurements. CB conducted the measurements at the peatland site and gave scientific advice. All authors discussed the results and FS, CA and CB reviewed the manuscript.

*Competing interests.* The authors declare that they have no conflict of interest.

*Acknowledgements.* This research was funded by the German Enviroment Agency (UBA) (project FORESTFLUX, support code FKZ
3715512110) and by the German Federal Ministry of Education and Research (BMBF) within the framework of the Junior Research Group NITROSPHERE (support code FKZ 01LN1308A). We thank Undine Zöll for scientific and logistical help, Jeremy Rüffer and Jean-Pierre Delorme for technical support, particularly during the field campaigns and the Bavarian Forest Nationalpark Administration, namely Burkhard Beudert, Wilhelm Breit and Ludwig Höcker for excellent technical and logistical support at BF site.





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
