# Peer review of "Correcting high-frequency losses of reactive nitrogen flux measurements"

_Atmospheric Measurement Techniques, 2019_

## Referee Comment (RC1) · Anonymous Referee #1 · 30 Jan 2020

The manuscript outlined that high-frequency measurements of reactive nitrogen species – required for eddy covariance observations of reactive nitrogen fluxes – are subject to high-frequency attenuation due to chemical reactions and adsorption/desorption along the intake tubing. The authors investigated five methodologies for correcting these losses, and applied them in a critical manner for two datasets taken over different surfaces. They were able to show that theoretical spectral corrections were lacking in characterizing the chemical losses in the eddy covariance system, and hence concluded that experimentally-derived corrections in the high-frequency range – in particular in-situ cospectral corrections – were the most appropriate method for their dataset.

The manuscript presents a good overview of the usual methods to correct for inlet

attenuation with a closed path flux sensor. The conclusion that corrections based on spectral similarity with an in-situ (non attenuated) scalar cospectrum like sensible heat work best, is very reasonable and undoubtedly correct.

Overall, methods and conclusions seem appropriate for terrestrial flux measurements where you have big signals, but will be tricky or impossible to apply when signals are weak.

While my work overlaps with eddy covariance flux research, I lack the depth of expertise to thoroughly evaluate the applied methods and their evaluation. The manuscript is overall very well organized and written. It considers and nicely builds on previous literature. Without doubt, the authors are leading experts in this field. This gives me confidence that this part of the paper is of high quality.

While overall the English language is easily understandable and pretty good, the paper would benefit from careful editing by a native English speaker. I believe the Copernicus staff will do that during the proofsetting, which will likely cover this need.

Here are a few pointers for a start: Commas should probably be inserted in lines

2,9,12,21,22,36,42,59,61,77,116,119,119,129,129,131,131,132,135,138,140, 146,148,180,187,190,196,197,202,215,220,242,245,247,247,252,254,256,269, 273,273,277,279,284,284,284,287,289,290,296,314,316,317,337,330,337,340, 341,360,366,382,396,403,439,440,444,451,452,463,498.

Please review and consider placing commas according to 'Oxford comma' rule.

Other minor comments:

Abstract: I suggest adding a sentence or two that summarize the value of this work to a general audience.

Line 13: I recommend using past tense (underestimated) here and in similar situations (Line 101, ....).

Line 75: Please provide detail about the sampling inlet.

Line 92: This study may also be of relevance: "Characterization and mitigation of water vapor effects in the measurement of ozone by chemiluminescence with nitric oxide", by Boylan, P.; Helmig, D.; Park, J. -H., ATMOSPHERIC MEASUREMENT TECHNIQUES Volume: 7 Issue: 5 Pages: 1231-1244 Published: 2014.

Lines 204, 213, 221: Can the authors provide reasons for choosing their frequency cutoff limits?

Figs. 4+5: Values for n are in an awkward location – perhaps at the very top or bottom of figure?

Table 1: Why would the lag time be different by a factor of 8, when the tubing length is different by a factor of 4, at similar flow rates?

Line 125: Isn't the time lag actually zero given the nature of the measurement?

Line 145: So, do I understand this right that actually only  ${\sim}10\%$  of the observations were retained for the data analyses after all filters had been applied?

Line 147: I suggest to delete 'as well as'.

Figure 4, 5: I found it hard to keep track of the many abbreviations used in the text and figures. Maybe the authors could provide a table that lists them all in one place? It would also help to explain/spell out abbreviations again in the Figure caption.

Line 241: What are the likely reasons for dampening of the temperature power spectrum?

Line 265: I would not call these graphs time series plots. They show statistical analyses of monthly data.

Line 278: "platinum gaze" = "platinum gauze"?

Line 287: It is not entirely clear in this sentence whether these periods of poor instru-

**СЗ**

ment performance were removed or included in the analysis. Perhaps a minor reword to make this clear.

Table 2: Consider swapping location of FOR and BOG for consistency with other figures.

Lines 303 to 305: Appears to repeat already-presented information. Consider beginning paragraph at "For investigating..."

Fig. 7: Swapped ordering of FOR and BOG again, as opposed to order in Fig. 6.

Line 330: Can you explain what the response time actually represents? It's obviously not the time from the entry of the sample in the inlet to when the instrument gave a signal?

Sec. 3.3: Could the analysis given between meteorological variables and alpha be repeated for tau? It seems that alpha and tau have an inverse relationship (covered briefly in the conclusions) and therefore might it be possible to include this in Sec. 3.2.

Line 364: Correct to 'ppb'.

Line 371: Can a similar analysis to that in the later section of Sec. 3.2 be used to investigate the influence of humidity on CO2?

Line 379: How can a logger contribute to the noise?

Line 486: Correct to 'low-frequency'.

Line 499: Based on your results and experience, can you comment on the possible complication from water vapor and NO having different lag times in the converter and sampling system, and how that then effects the interference from quenching of the NO signal?

Conclusions section: It was not clear to me whether the damping factor values translated directly into flux value reductions. Therefore, as the results/discussion was related largely to damping factor values, the introduction of flux value reductions in the conclusions seemed somewhat new. I would also suggest here to place again these flux reduction values within the previous literature, as cited earlier in the paragraph beginning in Line 45.

---

## Referee Comment (RC2) · Anonymous Referee #2 · 30 Jan 2020

**GENERAL COMMENTS**

The manuscript "Correcting high-frequency losses of reactive nitrogen flux measurements" by Wintjen et al. represents an important study which investigates the performance and applicability of different methods to correct for the high frequency attenuation of reactive nitrogen eddy covariance fluxes. This topic is very relevant since observed losses of reactive nitrogen fluxes are substantial and parametrizations that correct them are difficult to obtain due to the complexity of chemical and physical interactions of the reactive nitrogen species within the inlet system and instrumental setup.

Wintjen et al. compare five different methods that are commonly used for the flux correction of typically less reactive gases such as $H_2O$ and $CO_2$ and apply them to the reactive nitrogen flux measurements at a peatland and a forest site. Since they

performed flux measurements over several years, the authors have an extraordinary dataset to conduct this analysis covering a wide range of environmental conditions. Although they do not find significant dependencies of the flux loss with environmental conditions, they are able to propose that using an empirical co-spectral approach is most the most suited of the five methods they investigated.

While the scientific analysis and conclusions are mostly sound (see for more details the specific comment below), I found the manuscript very difficult to read and it should be improved from a reader's perspective. At many points, sentences lack clear links to each other, making it difficult to follow the line of arguments. In some sections, this creates the first impression that they are placed quite arbitrarily, which is enhanced by that fact that some statements are restated at multiple times within one paragraph. Furthermore, often it is not clear what is referred to (at least it can be ambiguous), interrupting the reading flow. To improve the readability, I strongly recommend to better link individual sentences, to remove redundant statements and to make referrals more precise. I believe this will significantly improve the quality of the manuscript, making for a stronger case for the suitability of each flux correction method.

In the following, scientific comments are listed that should be addressed. Furthermore, I provided additional comments where the manuscripts needs further clarifications. These do not need to be answered individually but should be considered in the revised version.

SPECIFIC COMMENTS

[1] L. 47-50: Since they used a different approach to correct NH3 fluxes, I suggest to include the recent study by Moravek et al. (2019) here.

[2] L. 173-174, Figure 1 + Figure2: The authors use averaged co/power-spectra of several hours measured on one specific day for each site. How representative are the chosen spectra for (daytime) conditions during the entire measurement period? While it is clear that the shown spectra are exemplary, it would be beneficial to state on how

many days spectra with the same described features were observed. Furthermore, is there a reason why different days were chosen for the co-spectra in Figure 1 and 2? If so, these should be highlighted.

[3] L. 182-193: How does this approach compare to the approach by Aubinet et al. (2000), which uses a normalization factor instead of normalizing by the total covariances? I find it important to highlight the methodological differences here as the method by Aubinet et al. (2000) is also referred to widely in literature (Foken et al., 2012). Furthermore, it should be shown here how Eq. (2) was derived from Eq (1). Co(...) and TF_exp are functions of the frequency, but then solving for alpha is not straight forward, unless I miss something here. In general, the description of how the fits to obtain Tau_exp were performed can be improved. For example it should be mentioned what kind of least-square fit was performed, linear, non-linear?

[4] L. 200-207: To support the descriptive text, the Ogive equation should be included here. Also, the "optimization factor" (L. 207) has not been defined yet.

[5] L. 223-224: It should be explained here briefly why this parameterization of alpha is used, which uses the horizontal wind speed. While it provides the opportunity to apply the correction to a large dataset (once fc is known), the methodology is different from the other approaches used here, that determine alpha more directly. The mean horizontal wind speed (u) should be defined in the text.

[6] L. 241: It should be stated that the slopes in the inertial subrange are meant here. In addition, shouldn't a weaker slope (-0.62 and -0.63 compared to -2/3) result an increased flux contribution in the high frequency range, which is the opposite as stated here? Was the entire inertial subrange used to derive the slopes? This is probably problematic since both positive and negative slopes are observed in the same power spectra.

[7] L. 254-255: Was the precipitation filter for the Li-7500 data only used for the evaluation of the presented slopes? If so, shouldn't it have been applied for filtering the power

spectra and cospectra as well?

[8] L. 370-373: A stronger white noise of Ps(CO2) at BOG than FOR is not visible to me in the power spectra presented in Figure 2. Which frequency range the authors referring to? May the slight increase in the very high frequencies be due to the aliasing effect?

[9] L. 350-410: The power spectra used for the IPS method do not follow the expected shape and the authors relate the increase of power densities at high frequency to white noise. This is a good demonstration of the shortcomings of the IPS method, for conditions where instrumental noise impacts over a certain frequency range. As the instrumental noise is uncorrelated with the vertical wind speed, it is not detected in the co-spectra. Still, it has to be discussed how the instrumental noise impacts the detection of small mixing ratio fluctuations that relate to the trace gas flux, which then would also impact the Co-spectra. In my view this is not clearly discussed in the manuscript. While the authors state that the instrument was probably unable to resolve small concentration differences, the power spectra at the BOG site show a steep decline in the inertial subrange that is similar to the one from the temperature time series. This would suggest that the instrument was capable of capturing the concentration differences in the high frequency range. Also, it would be useful to include in the discussion under which conditions the IPS method can be used.

[10] L. 397-398: The authors state that under conditions with "less variability in concentrations and depositions fluxes" the IPS method fails. Shouldn't it be just under low flux conditions (regardless whether deposition or emission fluxes prevail)?

[11] L. 434-437: The authors list here parameters that could affect the time response. However they do not discuss which component of the TRANC-CLD setup they expect to have the largest impact on the time response. Since all Nr compounds are converted to NO by the TRANC, interactions with the tubing wall may be less important than for example interactions of NH3 at the inlet, which is much "stickier" than NO. Therefore,

adding a short discussion on the expected high frequency attenuation processes would help to better understand the observed/non-observed dependencies of alpha with environmental and instrumental parameters. I think this is important to discuss since it would show whether the system's time response was more similar to that of a NO (i.e. time response of tubing and CLD) or to that of a NH3 analyzer (as a sticky compound with potentially large flux contribution).

[12] L. 450: The authors state that a general parameterization of alpha was not possible. Still, there are some dependencies of the empirical method with stability and wind speed. While it is difficult to derive a robust parameterization, shouldn't at least be distinguished between night time and daytime alpha values?

[13] L. 493-494: This sentence is misleading as the phase shift is obviously not the only cause for steep decay in the high frequency range. Rather it would be important to state here what the percentage contributions of both transfer functions are to the overall alpha values.

[14] L. 516-518: From the results shown in Figure 6, there seems to be clear differences between stable/unstable conditions for both sites, as well as a dependency with wind speed at a BOG site. To me, these differences/trends/dependencies – despite some uncertainties - should be mentioned in the conclusions.

ADDITIONAL COMMENTS

L. 34-35: Add that these measurements were mainly for NH3.

L. 36: "...which have...": from the sentence structure it is not clear that the authors are referring to the QCL and TRANC analyzers. I suggest making a new sentence here.

L. 79-83: What were the concentration ranges of Nr species observed at the site? They should be mentioned as well as it was done for the forest site.

L. 84-86: To make it easier to follow, I would to state which of the previously mentioned compounds is converted at each step.

L. 89: Since the sensor separation distance is very critical for the presented study, I suggest referring already here to Table 1.

L. 119-124: Since the time lag determination is influenced by the damping of high frequencies, it would be important to mention here what the observed variation of calculated time lags were.

L. 124: Did the authors use the range of the time lag computation as filtering criteria? If so, this should be stated here clearly since it is not stated in Sect. 3.1.

L. 140-141: Use "see above" instead of Sect. 2.2.

L. 164: Since the authors describe the response of a first-order system, better describe tau_r as the "time constant", since the "response time" can be interpreted as a multiple of that.

L. 184-185: I suggest to denote TF_R and TF_deltaR in text and use ":" at the end of sentence.

Figure 1-6: I suggest adding the measurement site (BOR, FOR) in each subplot. This will provide easier readability without having always to refer to the Figure captions. Furthermore use either the full or abbreviated site names consistently in all Figure caption.

L. 218-220: These sentences sound misleading. If the authors used $CO_2$ and $H_2O$ data from another eddy covariance setup that was installed in a certain distance from the reactive nitrogen flux measurements, then they did apply IPS to the BOG date, just with an additional uncertainty.

L. 230/Figure 2: It should be stated that the periods used to calculate the spectra in Figure 2 are different from the periods in Figure 1.

L. 245-246: The sentences are confusing as it sounds that another analysis was performed and it is not clear on which data set. Instead of using "We further estimated..."

I suggest to connect both sentences, for example like "at measurement sites, for which we estimated the slope…".

L. 250-260: I find this description of Figure 3 difficult to read since it seems to "jump" between positive and negative slopes, different scalars and sites. To make the description more concise one could for example speak of a "bi-modal distribution" of the $CO_2$ (for both sites) and Nr (in case of BOG) slopes.

L. 268: For alpha, either use "%" or ratio (as shown in Figure 5).

L. 303-304: This seems repetitive information as the IQR ranges of ICO and sICO were already described in L. 275-277.

L. 323-325: Do the authors mean here a differences compared to THEO or a bias "between" IOG, ICO and sICO?

L. 333: Do the authors mean no significant trend of the IQR? Alpha for sICO is decreasing with increasing stability.

L. 350- 410: As this subsection discusses several aspects of the noise effect, it should be divided into several paragraphs.

L. 377-378: "It may be caused…": This sounds like as if the drop in the high-frequency range was caused by white noise, but it is rather that the white noise (occurring at about 1 Hz as stated later).

L. 438-458: This paragraph discusses the general applicability of the correction methods for Nr fluxes, and less the differences between approaches. I suggest therefore to move this discussion to section at the end as it also relates to the Sections 4.3.2 and 4.3.3.

L. 471-473: In this sentence it is not clear whether alpha values from ICO or sICO were overestimated.

REFERENCES

Aubinet, M., Grelle, A., Ibrom, A., Rannik, U., Moncrieff, J., Foken, T., Kowalski, A. S., Martin, P. H., Berbigier, P., Bernhofer, C., Clement, R., Elbers, J., Granier, A., Grunwald, T., Morgenstern, K., Pilegaard, K., Rebmann, C., Snijders, W., Valentini, R. and Vesala, T.: Estimates of the annual net carbon and water exchange of forests: The EUROFLUX methodology, Adv. Ecol. Res. Vol 30, 30, 113–175, 2000.

Foken, T., Leuning, R., Oncley, S., Mauder, M. and Aubinet, M.: Corrections and Data Quality Control, in Eddy Covariance, edited by M. Aubinet, T. Vesala, and D. Papale, pp. 85–131, Springer Netherlands., 2012.

Moravek, A., Singh, S., Pattey, E., Pelletier, L. and Murphy, J. G.: Measurements and quality control of ammonia eddy covariance fluxes: a new strategy for high-frequency attenuation correction, Atmos. Meas. Tech., 12(11), 6059–6078, doi:10.5194/amt-12-6059-2019, 2019.

---

## Author Response (AR1)

**Response to reviewers' comments - manuscript *AMT-2019-375* "Correcting high-frequency losses of reactive nitrogen flux measurements"**

We thank the reviewers for their constructive comments. We have addressed all of them and modified the manuscript accordingly. Our detailed answers are given below. Referee comments are given in italic, the answers in standard font. The comments by Reviewer 1 were numbered from R1.1 to R1.27 titled as other minor comments, and the specific comments of Reviewer 2 range from R2.1 to R2.14. The additional comments start at R2.15 and end at R2.37. The line and figure numbers in the answers, where we will add the new information into the manuscript, refer in this document to the originally submitted version. The text which is enclosed by "..." will be implemented in the revised manuscript.

**Response to Reviewer 1**

**General Comments**

*The manuscript outlined that high-frequency measurements of reactive nitrogen species – required for eddy covariance observations of reactive nitrogen fluxes - are subject to high-frequency attenuation due to chemical reactions and adsorption/desorption along the intake tubing. The authors investigated five methodologies for correcting these losses, and applied them in a critical manner for two datasets taken over different surfaces. They were able to show that theoretical spectral corrections were lacking in characterizing the chemical losses in the eddy covariance system, and hence concluded that experimentally-derived corrections in the high-frequency range – in particular in-situ cospectral corrections – were the most appropriate method for their dataset.*
*The manuscript presents a good overview of the usual methods to correct for inlet attenuation with a closed path flux sensor. The conclusion that corrections based on spectral similarity with an in-situ (non attenuated) scalar cospectrum like sensible heat work best, is very reasonable and undoubtedly correct.*
*Overall, methods and conclusions seem appropriate for terrestrial flux measurements where you have big signals, but will be tricky or impossible to apply when signals are weak. While my work overlaps with eddy covariance flux research, I lack the depth of expertise to thoroughly evaluate the applied methods and their evaluation. The manuscript is overall very well organized and written. It considers and nicely builds on previous literature. Without doubt, the authors are leading experts in this field. This gives me confidence that this part of the paper is of high quality.*
*While overall the English language is easily understandable and pretty good, the paper would benefit from careful editing by a native English speaker. I believe the Copernicus staff will do that during the proofsetting, which will likely cover this need.*

Thank you for your compliments on this work. Your comments and recommendations are answered below.

**Other minor comments:**

**Comment R1.1** *Here are a few pointers for a start: Commas should probably be inserted in lines 2, 9, 12, 21, 22, 36, 42, 59, 61, 77, 116, 119, 119, 129, 129, 131, 131, 132, 135, 138, 140, 146, 148, 180, 187, 190, 196, 197, 202, 215, 220, 242, 245, 247, 247, 252, 254, 256, 269, 273, 273, 277, 279, 284, 284, 284, 287, 289, 290, 296, 314, 316, 317, 337, 330, 337, 340, 341, 360, 366, 382, 396, 403, 439, 440, 444, 451, 452, 463, 498. Please review and consider placing commas according to 'Oxford comma' rule*
**Response to R1.1** We added commas in the mentioned lines after the Oxford comma rule.

**Comment R1.2** *Abstract: I suggest adding a sentence or two that summarize the value of this work to a general audience*
**Response to R1.2** We appended the following sentences at the end of the abstract:
"Flux measurements of reactive nitrogen compounds are of increasing importance to assess the impact of unintended emissions and on sensitive ecosystems and to evaluate the efficiency of mitigation strategies. Therefore, it is necessary to determine the exchange of reactive nitrogen gases with the highest possible accuracy. This study gives insight in the performance of flux correction methods and their usability for reactive nitrogen gases."

**Comment R1.3** *Line 13. I recommend using past tense (underestimated) here and in similar situations (Line 101, . . ...)*
**Response to R1.3** Changed.

**Comment R1.4** *Line 75: Please provide detail about the sampling inlet.*
**Response to R1.4** We added the following sentences to the corresponding section (Line 83):
"The sampling inlet was designed after Marx et al. (2012) and Ammann et al. (2012). The inlet tube is 15 cm long, consists of FeNiCr, has an outer diameter of 1/4", and is actively heated from the edge of the tube. Inner temperatures are higher than 100°C".
Further details about the sampling inlet are given in the cited publications.

**Comment R1.5** *Line 92: This study may also be of relevance: "Characterization and mitigation of water vapor effects in the measurement of ozone by chemiluminescence with nitric oxide", by Boylan, P.; Helmig, D.; Park, J. -H., ATMOSPHERIC MEASUREMENT TECHNIQUES Volume: 7 Issue: 5 Pages: 1231-1244 Published: 2014*
**Response to R1.5** We thank the reviewer for the literature advice. The instrument presented by Boylan et al. (2014) is a custom-built chemical luminescene analyzer and suited for the detection of ozone. They determined a reduction in sensitivity of 4.15% in ozone signal per $10 \, \text{mmol} \, \text{mol}^{-1}$ water vapor. The reduction in sensitivity of our CLD was determined to 0.19% per $1 \, \text{mmol} \, \text{mol}^{-1}$ water vapor increase by Marx (2004). Both values are comparable. Since the instrument is not the same as the one we used, a comparison would be of low value and we decided not to mention the ozone analyzer. Brümmer et al. (2013) and Ammann et al. (2012) discussed the impact of water vapor on nitrogen concentrations measured with the CLD TR780 and give an equation for the correction of estimated fluxes.

**Comment R1.6** *Lines 204, 213, 221: Can the authors provide reasons for choosing their frequency cutoff limits?*
**Response to R1.6** The limits were chosen with regard to former studies and subjective decisions explained in the following lines. The limits of the fitting range for IOG were based on the values and suggestions of Ammann et al. (2006) and Ferrara et al. (2012). The missing information about the examination of the power spectral cut-off limit has been added to Sec. 2.3.5.
"The value for the 'lowest noise frequency', which was set in EddyPro for running IPS, was a subjective decision based on visual screening through power spectra. Therefore, we calculated slopes of $\Sigma N_r$ power spectra in the inertial subrange and estimated the frequency, at which noise started to increase and slopes got positive."
Zöll et al. (2016) conducted $NH_3$ flux measurements based on the EC method in close proximity to our tower. The cut-off limits of their high-frequency damping analysis were similar to our values (personal communication), which is probably related to the high amount of $NH_3$ in the $\Sigma N_r$ signal (Hurkuck et al., 2014; Zöll et al., 2016). Zöll et al. (2016) estimated the attenuation of their EC system with the ogive method.
In case of cospectral approach, the following lines were added to Sec. 2.3.5.
"The decision of the lower frequency limits were further proven by the examination of the ogives ratio, which shows constant values in a certain frequency range. The position exhibits the frequency, at which high-frequency attenuation mostly starts to increase."

**Comment R1.7** *Figs. 4+5: Values for n are in awkward location - perhaps at the very top or bottom of figure?*
**Response to R1.7** We changed the position of the values and placed them at the top of the figures.

**Comment R1.8** *Table 1: Why would be the lag time be different by a factor of 8, when the tubing length is different by a factor of 4, at similar flow rates?*
**Response to R1.8** The dimensions of the critical orifice were different causing the different lag times and therefore mostly responsible for the differences. The calculation through the equation for volumetric flow rate and uniform movement is not possible, because the equations do not consider the pressure reduction induced by the critical orifice. The pressure reduction in the tube results in a higher gas flow.

**Comment R1.9** *L. 125: Isn't the time lag actually zero given the nature of the measurements?*
**Response to R1.9** A time lag for the open-path $CO_2/H_2O$ analyzer arised due to separation distance between the sonic and the gas analyzer. Thus, the time lag was less than a second, but usually greater than zero.

**Comment R1.10** *L. 145: So, do I understand this right that actually only $\sim 10\%$ of the observations were retained for the data analyses after all filters had been applied?*
**Response to R1.10** Yes, about 10% of the cospectra were used for the damping analysis. These cospectra passed different filtering criteria ensuring a high quality for the attenuation analysis. We confirm that the number of valid cospectra is not quite high. This is mostly related to physical and chemical properties of $\Sigma N_r$ (lines 438-443). Due to the variability in concentrations and generally a low concentration level of $\Sigma N_r$, the influence of noise on cospectra can vary significantly. Consequently, instruments need to detect very low fluctuations of $\Sigma N_r$ precisely, which is probably not possible for the instrument (lines 388-393). Inert gases like $CO_2$ or $H_2O$ have much higher concentrations and distinctive daily cycles. Therefore, the impact of noise on cospectra is much lower, and thus the amount of high-quality cospectra is much higher. Consequentially, the low amount of valid cospectra is related to the complexity of compounds in $\Sigma N_r$ and to current limitations in the detection limit of the devices.

**Comment R1.11** *L 147: I suggest to delete 'as well as'.*
**Response to R1.11** Accepted and replaced with 'and'. We further rearranged the sentence and slightly modified it.
"The theoretical slope for power spectra of temperature and inert trace gas concentrations is -2/3."

**Comment R1.12** *Figure 4,5: I found it hard to keep track of the many abbreviations used in the text and figures. Maybe the authors could provide a table that lists them all in one place? It would also help to explain/spell out abbreviations again in the Figure caption.*
**Response to R1.12** We included a table with all necessary abbreviations at the end of Sec 2.3.
"Table R1 gives an overview about abbreviations used in this study."

| Parameter or Term | Abbreviation |
| --- | --- |
| Theoretical damping calculation | THEO |
| In-situ cospectral method | ICO |
| Semi in-situ cospectral method | sICO |
| In-situ ogive method | IOG |
| In-situ power-spectral method | IPS |
| (Power) spectrum | Ps(..) |
| Cospectrum | Co(..) |
| Ogive | Og(..) |
| Transfer function | TF |
| response time | $\tau_r$ |
| damping factor | $\alpha$ |
| Bourtanger Moor (semi-natural peatland) | BOG |
| Bavarian Forest (mixed forest) | FOR |
| Total Reactive Atmospheric Nitrogen Converter | TRANC |
| Chemiluminescence dectector | CLD |

Table R1: Important terms and corresponding shortcuts used in this study.

**Comment R1.13** *Line 241: What are the likely reasons for dampening of temperature power spectrum?*
**Response to R1.13** A slight high-frequency damping of $Ps(T)$ can be caused by the path averaging of the sonic anemometer (e.g. Moore, 1986). In addition, the observed shape of the spectrum (slope) can deviate from the theoretical shape due to non-ideal environmental conditions (e.g. non-homogeneous turbulence, influence of roughness sublayer). We added this information at line 241.

**Comment R1.14** *Line 265: I would not call these graphics time series plots. They show statistical analyses of monthly data.*
**Response to R1.14** Revised. We corrected it according to the reviewer's suggestion. Terms were exchanged to "statistical analyses" at further locations, e.g. line 340, 341, and the corresponding figure captions.

**Comment R1.15** *Line 268: "platinum gaze" = "platinum gauze"?*
**Response to R1.15** It is platinum gauze.

**Comment R1.16** *Line 287: It is not entirely clear in this sentence whether these periods of poor instrument performance were removed or included in the analysis. Perhaps a minor reword to make this clear.*
**Response to R1.16** We thank the reviewer for his/her advice. We added some information to line 111 and rephrased the corresponding sentence:
"Periods of maintenance and insufficient instrument performance were removed from damping analysis by manual screening and monitoring performance parameters such as TRANC heating temperature or flow rate."
As a matter of fact, not all affected fluxes can be excluded by the selection criteria. Thus, an

influence on the damping analysis can't be excluded.

**Comment R1.17** *Table 2: Consider swapping location of FOR and BOG for consistency with other figures.*
**Response to R1.17** Revised.

**Comment R1.18** *Line 303 to 305: Appears to repeat already-presented information. Consider beginning paragraph at "For investigating..."*
**Response to R1.18** We deleted the corresponding lines.

**Comment R1.19** *Fig. 7: Swapped ordering of FOR and BOG again, as opposed to order Fig. 6.*
**Response to R1.19** We changed the position of the subplots.

**Comment R1.20** *Line 330: Can you explain what the response time actually means? It's obiously not the time from the entry of the sample in the inlet to when the instrument gave a signal?*
**Response to R1.20** We agree that the term 'response time' in the manuscript needs further clarification. We added a corresponding paragraph at the end of Sec. 2.3.1:
"Physically, the analyser response time $\tau_{r,a}$ represents the time, at which the difference between the measured signal and the measured quantity is reduced by 1/e after a step change. Thus, it is also called e-folding time. If it is zero, changes will be recognized instantaneously. This is mostly not possible for common gas analysers. Our TRANC-CLD system, which has proven to be suitable for EC measurements (Marx et al., 2012; Brümmer et al., 2013), has an e-folding time of about 0.3-0.35 s. $\tau_{r,a}$ is used for the first-order filter transfer function (Table A1) in the THEO approach. In this manuscript $\tau_r$, which is also called response time, is a fitting parameter used in equation (2). It is linked to the cut-off frequency $f_c = 1/2\pi\tau_r$, at which the cospectrum is damped to $1/\sqrt{2} \approx 0.71$ or the power spectrum to 50%."

**Comment R1.21** *Sec 3.3: Could the analysis given between meteorological variables and alpha be repeated for tau? It seems that alpha and tau have an inverse relationship (covered briefly in the conclusions and therefore might it be possible to include this in Sec. 3.2.*
**Response to R1.21** According to the reviewers suggestion, we did a statistical analysis of $\tau_r$ classified by meteorological variables. We further determined the correlation between monthly averaged $\tau_r$ and $\alpha$. Correlations of -0.83 for BOG and -0.72 for FOR show that there is significant inverse relation between both parameters, which is expected due to the inverse dependency of $\tau_r$ in the empirical transfer function. The analysis of $\tau_r$ stratified by meteorological variables can be useful in order to investigate, if the scattering in $\alpha$ is related either to the variability in cospectra or to the instrument performance. $\tau_r$ is mostly a device-specific parameter. It should have a higher affinity to instrument or measurement setup parameters such as measurement height, pump and heating efficiency, altering of the inlet, and sensitivity of the analyser than to atmospheric, turbulent variations. Changes in gas concentrations may also affect $\tau_r$. Therefore, we classified the meteorological parameters into bins, calculated $\tau_r$ for each bin and display them as box plots. The box plot is shown below.

[Figure]

Figure R1: Dependency of the response time ($\tau_r$) on stability and wind speed classes as box plots without whiskers and outliers (box frame = 25 % to 75 % interquartile range (IQR), bold line = median). (a) and (b) refer to the BOG site and (c) and (d) to the FOR site.

$\tau_r$ is mostly constant for medium and high wind speed at BOG and exhibit slightly higher values at low wind speeds (0-0.5 m/s). During highly stable and unstable conditions $\tau_r$ reaches up to 3.5 s. It seems rather constant during medium, unstable conditions, but increases under stable conditions. The same is valid for $\tau_r$ at FOR. $\tau_r$ exhibits highest values under both highly unstable and stable conditions. However, $\tau_r$ is strongly affected by wind speed at FOR. It decreases with wind speed and seems to follow a non-linear relationship. The analysis confirms the statements of our conclusions that the usage of constant $\tau_r$ or $\alpha$ after meteorologically classified parameters is problematic. Generally, it is not known how much the variability in $\tau_r$ contributes to the scattering in $\alpha$ for certain wind speeds or stability values. Thus, usage of $\tau_r$ classified by meteorological parameters is only recommended for medium or high wind speeds at BOG or near-neutral and unstable atmospheric conditions at both sites. As mentioned in the conclusion, using a constant $\tau_r$ is problematic due to its variation with time, which is probably related to the instrument performance or changes in the composition of $\Sigma N_r$.

For covering the additional aspects, we extended the results Sec. 3.3 and introduce a new section in the discussion. In Sec. 4.3 we give recommendations for correcting high-frequency flux losses of $\Sigma N_r$, for example, which correction factor seem to be most suitable to correct our flux data. Additionally, we moved parts of line 438 to 458 to this subsection (see comment of Reviewer 2, R2.36).

**Comment R1.22** *Line 364: Correct to 'ppb'.*
**Response to R1.22** Corrected.

**Comment R1.23** *Line 371: Can a similar analysis to that in the later section of Sec. 3.2*

*be used to investigate the influence of humidity on CO2?*

**Response to R1.23** In principle, it is possible for $CO_2$ or other trace gases, too. However, such an analysis is beyond the scope of this study, which is focused on $\Sigma N_r$ rather than on $CO_2$. The determined slopes of $Ps(CO_2)$ can be separated into different humidity classes and displayed as box plots. A similar analysis could be done for the $CO_2$ damping factors. A comparison may help to investigate the impact of white noise on the damping under wet or humid conditions.

**Comment R1.24** *Line 379: How can a logger contribute to noise?*

**Response to R1.24** We noticed the mistake, deleted the logger from the enumeration, and changed the enumeration as follows: "[..] components of the setup like pump, air-conditioning system or electrical components."

**Comment R1.25** *Line 486: Correct to 'low-frequency'.*

**Response to R1.25** Revised.

**Comment R1.26** *Line 499: Based on your results and experience, can you comment on the possible complication from water vapor and NO having different lag times in the converter and sampling system, and how that then affects the interference from quenching of the NO signal?*

**Response to R1.26** The effect of water vapor quenching on the NO signal is very small, which is caused by the low sensitivity reduction of 0.19% per $1\,\mathrm{mmol\,mol^{-1}}$ water vapor. The impact on determined fluxes is corrected by the procedure given in Brümmer et al. (2013) and Ammann et al. (2012). Brümmer et al. (2013) found only $25\,\mathrm{g\,N\,ha^{-1}}$ higher deposition during 11-month field campaign. We measured $\Sigma N_r$ exchange above a remote, mixed forest for 2.5 years. According to Brümmer et al. (2013), we got a shift towards less deposition by approximately $100\,\mathrm{g\,N\,ha^{-1}}$ on the remaining quality-controlled fluxes. Total dry nitrogen deposition was around $12\,\mathrm{kg\,N\,ha^{-1}}$ during the 2.5 year field campaign. The correction is only applied to the determined fluxes.

**Comment R1.27** *Conclusion section: It was not clear to me whether the damping factor values transform directly into flux value reductions. Therefore, as the results/discussion was related largely to damping factor values, the introduction of flux value reductions within in the conclusions seemed somewhat new. I would also suggest here to place again these flux reduction values within the previous literature, as cited earlier in the paragraph beginning in line 45.*

**Response to R1.27** We agree with the reviewers comment and added the presented lines to the results to differentiate between damping factors and flux loss values (line 300). Furthermore, we put the citations of the previous literature in the conclusion (line 510):

"By subtracting the damping factor from an ideal, unattenuated system, which has a damping factor of one, the result will be the flux loss value (=$1-\alpha$). This loss value shows how much of the signal is lost from the inlet to the analysis of the signal by the instrument. Thus, flux losses calculated by IPS for our TRANC-CLD setup are around 6% at BOG and around 5% at FOR. The flux loss after THEO was approximately 12% at BOG and about 5% at FOR. The methods using measured cospectra or ogives (ICO, sICO and IOG) showed a flux loss of roughly 16-22% for FOR and around 26-38% for BOG. ICO shows the strongest damping at both sites."

**Response to Reviewer 2**

**General Comments**

*The manuscript "Correcting high-frequency losses of reactive nitrogen flux measurements" by Wintjen et al. represents an important study which investigates the performance and applicability of different methods to correct for the high frequency attenuation of reactive nitrogen eddy covariance fluxes. This topic is very relevant since observed losses of reactive nitrogen fluxes are substantial and parametrizations that correct them are difficult to obtain due to the complexity of chemical and physical interactions of the reactive nitrogen species within the inlet system and instrumental setup.*

*Wintjen et al. compare five different methods that are commonly used for the flux correction of typically less reactive gases such as H2O and CO2 and apply them to the reactive nitrogen flux measurements at a peatland and a forest site. Since they performed flux measurements over several years, the authors have an extraordinary dataset to conduct this analysis covering a wide range of environmental conditions. Although they do not find significant dependencies of the flux loss with environmental conditions, they are able to propose that using an empirical co-spectral approach is most the most suited of the five methods they investigated.*

*While the scientific analysis and conclusions are mostly sound (see for more details the specific comment below), I found the manuscript very difficult to read and it should be improved from a reader's perspective. At many points, sentences lack clear links to each other, making it difficult to follow the line of arguments. In some sections, this creates the first impression that they are placed quite arbitrarily, which is enhanced by that fact that some statements are restated at multiple times within one paragraph. Furthermore, often it is not clear what is referred to (at least it can be ambiguous), interrupting the reading flow. To improve the readability, I strongly recommend to better link individual sentences, to remove redundant statements and to make referrals more precise. I believe this will significantly improve the quality of the manuscript, making for a stronger case for the suitability of each flux correction method.*

*In the following, scientific comments are listed that should be addressed. Furthermore, I provided additional comments where the manuscripts needs further clarifications. These do not need to be answered individually but should be considered in the revised version.*

We thank the Reviewer for his/her comments and criticism on this work. We addressed all mentioned points and implemented most of the suggestions in the revised manuscript. A detailed reply to your comments is given below.

**Specific comments**

**Comment R2.1** *L. 47-50: Since they used a different approach to correct NH3 fluxes, I suggest to include the recent study by Moravek et al. (2019) here.*
**Response to R2.1** We added it to the corresponding chapter:
"Moravek et al. (2019) proposed a new approach for correcting high-frequency flux losses of $NH_3$ measured by a QCL. The method is based on frequently measuring the analyzer's time response. The application of this method resulted in 46% flux loss."

**Comment R2.2** *L. 173-174, Figure 1+Figure 2: The authors use averaged co/power-spectra of several hours measured on one specific day for each site. How representative are the chosen*

*spectra for (daytime) conditions during the entire measurement period? While it is clear that the shown spectra are exemplary, it would be beneficial to state on how many with the same described features were observed. Furthermore, is there a reason why different days were chosen for the cospectra in Figure 1 and 2? If so, these should be highlighted.*

**Response to R2.2** On average, wind speed and stability were approximately $1.65\,\text{ms}^{-1}$ and -0.22 at BOG during daytime. At FOR, the average wind speed and stability were $1.91\,\text{ms}^{-1}$ and -0.44 during daytime. Wind speed conditions of the averaged power spectra and cospectra are almost similar to the average values during daytime for the entire period. Stability values of the displayed case are in agreement with daytime average for BOG. At FOR, the shown example refers to stable conditions whereas an unstable average is exhibited during daytime. In general, daytime stability values of both sites are rather low and close to neutral conditions. At both sites, approximately 10% of the analyzed cospectra were in the range of $\pm 0.5\,\text{ms}^{-1}$ to the average wind speed and $\pm 0.15$ to the average stability. Using only the wind speed restriction resulted in 40% agreement at FOR and 55% at BOG. It seems that the stability is more diverse and not correlated to wind speed. The correlation between wind speed and stability for the analyzed cospectra used for the damping analysis is rather low for both sites (0.26 for BOG and 0.15 for FOR). The choice of different days was caused by data gaps in the measurements.

The discussion of the long-term wind speed and stability averages compared to the exemplary case were added to the manuscript beginning at line 178.

**Comment R2.3** *L. 182-193: How does this approach compare to the approach by Aubinet et al. (2000), which uses a normalisation factor instead of normalizing by the total covariances? I find it important to highlight the methodological differences here as the method by Aubinet et al. (2000) is also referred to widely in literature (Foken et al., 2012). Furthermore, it should be shown here how Eq. (2) was derived from Eq. (1). Co(...) and TF_exp are functions of the frequency, but then solving for alpha is not straight forward, unless I miss something here. In general, the description of how the fits were performed can be improved. For example, it should be mentioned what kind of of a least-square fit was performed, linear, non-linear?*

**Response to R2.3** We improved the description of the ICO method and the estimation of $\alpha$ according to the suggestion of Reviewer 2. The following sentences were added after line 185.

"The approach used in this study is somewhat different to other methods that are also based on using measured cospectra of heat and gas flux, for example the method of Aubinet et al. (1999). The latter uses a normalisation factor, which corresponds to the ratio of the heat flux cospectrum to gas flux cospectrum. Both cospectra are integrated until frequency $f_\text{o}$, which should not be affected by high-frequency damping, but high enough to allow an accurate calculation of the normalisation factor. However, the definition of $f_\text{o}$ is rather imprecise and thus, an incorrect setting of $f_\text{o}$ can lead to significant uncertainties in the damping analysis. In our approach cospectra are normalized by their corresponding total covariance. In order to consider the damping of the gas flux cospectrum and its covariance, the damping factor is introduced in Eq. (2). Thus, we assume that both approaches give similar results, since both approaches cover the damping of the gas flux cospectrum."

We also expand Eq. (1) to clarify the definition of $\alpha$ (line 155). Furthermore, we add the following flow chart to support the descriptive text about the procedure for determining $\alpha$ after ICO.

[Figure]

Figure R2: Illustration of the calculation of $\alpha$ and $\tau_r$ by ICO.

**Comment R2.4** *L. 200-207: To support the descriptive text, the Ogive equation should be included here. Also, the "optimization factor (L.207) has not been defined yet.*
**Response to R2.4** We add the mathematical definition of the ogive to the corresponding section (line 200). We rewrite line 207 to further specify the optimization factor.
"Finally, the optimization factor, which minimizes the difference between $Og(w, \Sigma N_r)$ against $Og(w, T)$, is the result of the least-squares problem and corresponds to the damping factor."

**Comment R2.5** *L. 223-224: It should be explained here briefly why this parameterization of alpha is used, which uses the horizontal wind speed. While it provides the opportunity to apply the correction to a large dataset (one fc is known), the methodology is different fro the other approaches used here, that determine alpha more directly. The mean horizontal wind speed (u) should be defined in the text.*
**Response to R2.5** The following sentences were added to the Sec. 2.3.5 (after line 224).
"In general, the idea of IPS is that the EC system can be simulated by a recursive filter. Thereby, $\alpha$ is determined by the ratio of the unfiltered covariance $\overline{w'T'}$ to the filtered covariance $\overline{w'T'_f}$, and applying the recursive filter to degrade the time series of sonic temperature (Ibrom et al., 2007). However, Ibrom et al. (2007) argued that this ratio gives erroneous results for small fluxes. Therefore, they parameterized $\alpha$ by the mean horizontal wind speed ($\bar{u}$), stability, and $f_c$. Ibrom et al. (2007) investigated a proportionality between $\alpha^{-1}$ and $u \cdot f_c^{-1}$. By introducing a proportionality constant $A_1$ and a second constant $A_2$, which should consider for spectral properties of the time series, the following equation for calculating the correction factor was proposed (for details see Ibrom et al., 2007, Sec. 2.4):."

**Comment R2.6** *L.241: It should be stated that the slopes in the inertial subrange are meant here. In addition, shouldn't a weaker slope (-0.62 and -0.63 compared to -2/3) result in an increased flux contribution, which is the opposite as stated here? Was the entire inertial subrange used to derive the slopes? This is probably problematic since both positive and negative slopes are observed in the same power spectra*

**Response to R2.6** We specified that the slope of $Ps(T)$ was estimated for the inertial subrange (line 240) and corrected the sentence about the interpretation of the slope value (line 241). The fitting range used for the derivation of the slopes is smaller than the inertial subrange, for example, to exclude slightly positive slopes at the very high frequencies of the inert trace gases.

**Comment R2.7** *L. 254-255: Was the precipitation filter for the Li-7500 data only used for the evaluation of the presented slopes? If so, shouldn't it have been applied for filtering the power and cospectra as well?*

**Response to R2.7** The precipitation filter was also applied for filtering the lower quality cases of $CO_2$ and $H_2O$ shown in Fig. 3. Since the TRANC-CLD setup is a closed-path system, the precipitation filter was not applied to $\Sigma N_r$ measurements. Quality assurance of $Ps(T)$ and $Co(w, T)$ were made by the criteria of Mauder and Foken (2006), which already account for insufficient conditions compromising flux calculation. Due to the stricter criteria for $\Sigma N_r$ filtering most of the less quality cases were rejected.

**Comment R2.8** *L. 370-373: A stronger white noise of Ps(CO₂) at BOG than at FOR is not visible to me in the power spectra presented in Figure 2. Which frequency range are the authors referring to? May the slight increase in the very high frequencies be due to aliasing effect?*

**Response to R2.8** We apologise for not including a figure reference here. In fact the text refers to data shown in Figure 3, not Figure 2. We improved the mentioned lines as shown below. The slight increase at very high-frequencies of $Ps(CO_2)$ observed in Fig. 2 is mostly probably not induced by aliasing (of the real flux contributions above 5 Hz), because of the very steep slope and the lack of a similar increase effect in $Ps(T)$ and $Ps(H_2O)$.

"Before, we argued that concentration of $\Sigma N_r$ leads to differences in the slope distribution. Concentrations of $CO_2$ were not significantly different between the sites. As a consequence, there has to be another parameter responsible for discrepancy in the contribution of positive $Ps(CO_2)$ slopes at the measurement sites. We suppose that the discrepancy of positive $Ps(CO_2)$ slopes corresponds to different levels of humidity at the measurement sites."

**Comment R2.9** *L.350-410: The power spectra used for the IPS method do not follow the expected shape and the authors relate the increase of power densities at high frequency to white noise. This is a good demonstration of the shortcomings of the IPS method, for conditions where instrumental noise impacts over a certain frequency range. As the instrumental noise is uncorrelated with the vertical wind speed, it is not detected in the co-spectra. Still, it has to be discussed how the instrumental noise impacts the detection of small mixing ratio fluctuations that relate to the trace gas flux, which then would also impact the Co-spectra. In my view this is not clearly discussed in the manuscript. While the authors state that the instrument was probably unable to resolve small concentration differences, the power spectra at the BOG site show a steep decline in the inertial subrange that is similar to the one from the temperature time series. This would suggest that the instrument was capable of capturing the concentration differences in the high frequency range. Also, it would be useful to include in the discussion under which conditions the IPS method can be used.*

**Response to R2.9** Since the question is quite long and treats several aspects of Sec. 4.1, we divided the answer into several paragraphs. At first, we complemented the lines 390-393 by the discussion about the impact of the instrumental noise:

"Instrumental noise affects the shape of the covariance function. It can lead to a broadening of the covariance peak and generally enhances the scattering of the covariance values. Both effects are already enlarged in case of small mixing ratio fluctuations. Thus, instrumental noise further compromises the time lag estimation and leads to additional noise in cospectra. Due to the applied time lag criterion the effect of instrumental noise is mostly cancelled out. The position of the cospectral peak is less impacted, and thus instrumental noise can only lead to an enhancement of scattering of cospectral values, preferentially in the low-frequency range of the cospectrum. In other words, instrumental noise mostly contributes to low-frequent noise, the red noise."

Based on the reviewer's advice we improved our suggestion using IPS for reactive gases. We added the following aspects to the section about the impact of noise on IPS beginning at line 398:

"However, Fig. 2 reveals that $Ps(\Sigma N_r)$ shows a steep decline in the high-frequency range after the peak at BOG, which is similar to the decline of $Ps(T)$. $\Sigma N_r$ concentration was 24.4 ppb on average and exhibits a standard deviation of 9.6 ppb for the averaging period in Fig. 2 suggesting significant differences in concentration levels. It confirms the statement that the concentration is an important driver for the quality of $Ps(\Sigma N_r)$. This leads us to the assumption that the instrument was in principle able to capture differences in concentration levels in the high-frequency range if mixing ratio fluctuations are relatively high."

Basically, for time periods with remarkable differences in concentration levels IPS should perform well. Yet, the opposite is the case which is probably related to the negligible difference in the slopes of $Ps(\Sigma N_r)$ and $Ps(T)$, and to the violation of the critical assumption requesting spectral similarity of the power spectra (Fig. 2). This assumption is clearly not valid for $Ps(\Sigma N_r)$ due to the low-frequent noise.

"Consequently, for estimating damping factors with IPS certain conditions seem to be fulfilled. For example, instruments need a low detection limit for detecting low turbulent fluctuations, sources and impact of noise and strategies for the elimination of noise should be known, gases should be rather inert or have little interaction with surfaces or other chemical compounds, and, in case of IPS, show a wind speed dependency on damping factors. Similar to cospectral methods, IPS will also benefit from a well-defined footprint, equal canopy height, and sufficient turbulence. Satisfying these aspects is quite difficult for a custom-built EC system, which is rather new and thus not all attenuation processes are identified, and designed for measuring a trace gas, which consists of several compounds with unknown contribution, complex reaction pathways, and generally low fluctuations." Our conclusion is that IPS seems not be the optimal choice for correcting high-frequency flux losses of $\Sigma N_r$ due to large red and white noise.

**Comment R2.10** *L. 397-398: The authors state that under conditions with "less variability in concentrations and deposition" fluxes the IPS method fails. Shouldn't it be just under low flux conditions (regardless whether deposition or emission fluxes prevail)?*

**Response to R2.10** We agree with the reviewer's comment and correct the sentence as follows: "Therefore, we recommend using cospectra to estimate $\tau_r$ and $\alpha$ of reactive gases, since these gases exhibit normally low density fluctuations."

**Comment R2.11** *L. 434-437: The authors list here parameters that could affect the time response. However they do not discuss which component of the TRANC-CLD they expect to have the largest impact on the time response. Since all Nr compounds are converted to NO by the TRANC,*

*interactions with the tubing walls may be less important than for example interactions of NH3 at the inlet, which is much stickier than NO. Therefore, adding a short discussion on the expected high-frequency attenuation processes would help to better understand the observed/non-observed dependencies of alpha with environmental and instrumental parameters. I think it is important to discuss since it would show whether the system's time response was more similar to that of NO (i.e. time response of tubing and CLD) or to that of a NH3 analyzer (as a sticky compound with potentially large flux contribution).*

**Response to R2.11** We agree with the reviewer that such a discussion is helpful for understanding, which part of the TRANC-CLD setup has the strongest impact on the system's time response and damping. We rephrased the mentioned lines and consider the new aspects in the revised manuscript as follows:

"There could be other effects which superpose the wind speed and stability dependencies, for example, (chemical) damping processes occurring inside the TRANC-CLD system. Humidity and $\Sigma N_r$ could affect the aging of the tube and consequentially the adsorption at inner tube walls. However, we found no dependency of these parameters on damping factor and time response. Interactions with tube walls is probably less important, especially for the tube connecting the end of the TRANC to the CLD, because the main trace gas within the line is NO, which acts rather inert in the absence of ozone and $NO_2$. Because $NO_2$ and $O_3$ are converted in the TRANC, it can be assumed that the influence of interaction with tube walls on time response and high-frequency flux losses is mostly negligible compared to effects, which happen in the CLD and TRANC. The CLD contributes more to the total attenuation than the tubing, but supposedly not as much as the TRANC. Rummel et al. (2002) also used the CLD 780 TR as device for measuring NO fluxes. High-frequency flux losses were rather low and ranged between 21% (close to the ground) and 5% (11 m above ground). Also, Wang et al. (2020) observed low flux losses of NO by approximately 12% by measuring with a QCL (1.7 m above ground). Consequently, the strongest contributor to the overall damping has to be the TRANC. $NH_3$ is, considering all possible convertible compounds, the most abundant in certain ecosystems, highly reactive, and rather "sticky". In absolute terms it has the highest influence on the damping of $\Sigma N_r$. QCL devices, which may be used for the detection of $NH_3$ (Ferrara et al., 2012; Zöll et al., 2016; Moravek et al., 2019), were equipped with a special designed, heated, and opaque inlet to avoid sticking of $NH_3$ at tube walls, water molecules, and preventing unwanted molecules entering the analyser cell. Thus, $NH_3$ has high flux loss factors ranging from 33 to 46% (Ferrara et al., 2012; Zöll et al., 2016; Moravek et al., 2019). These damping factors are closer to the damping factors of $\Sigma N_r$, in particular for BOG, at which high $NH_3$ concentrations were measured and most of $\Sigma N_r$ can be attributed to $NH_3$ (Hurkuck et al., 2014; Zöll et al., 2016). At FOR, flux losses were lower due to physical reasons (Sec. 4.1, lines 393-394) and due to lower contribution of $NH_3$ to $\Sigma N_r$ at FOR (Sec. 4.2.1, lines 447-449). $NH_3$ is converted inside the TRANC at the platinum gauze after passing through the actively heated inlet and iron-nickel-chrome (FeNiCr) alloy tube. Since the main part of the pathway is heated and isolated against environmental impacts, the inlet of the TRANC and the distance to the sonic seem to be critical for the detection and attenuation of $NH_3$. Finally, we suppose that the response time and attenuation of our TRANC-CLD system is more similar to that of an $NH_3$ analyzer under a high ambient $NH_3$ load."

**Comment R2.12** *L. 450: The authors state that a general parameterization of alpha was not possible. Still, there are some dependencies of the empirical method with stability and wind speed. While it is difficult to derive a robust parameterization, shouldn't at least be distinguished between night time and day time alpha values?*

**Response to R2.12** The motivation behind this formulation was that there was no explicit dependency on the investigated parameters. Dependencies on wind speed and stability were only valid for certain ranges and under certain site conditions. For example, a parameterization can be performed for unstable conditions and for wind speeds above $1.5\,\mathrm{ms^{-1}}$ at BOG. As mentioned in the manuscript, other parameters showed no clear dependency on $\alpha$. We classified $\alpha$ in different radiation classes, but found no significant difference between day and night time values. The exchange pattern of $\Sigma\mathrm{N_r}$ is rather bi-directional during the entire day. The exchange pattern of inert gases like $CO_2$ is largely related to photosynthesis and respiration. During daytime $CO_2$ exhibits also bi-directional exchange characteristics. During the night the exchange of $CO_2$ is mostly unidirectional. Thus, we would expect a diurnal variation of the $CO_2$ attenuation. The influence of global radiation on the biosphere-atmosphere exchange of $\Sigma\mathrm{N_r}$ and $CO_2$ was explicitly shown by (Zöll et al., 2019) for FOR. They also investigated drivers of $\Sigma\mathrm{N_r}$. However, global radiation explained only 22% of the variability in $\Sigma\mathrm{N_r}$ fluxes, whereas 66% of the variability in $CO_2$ fluxes were related to global radiation. $\Sigma\mathrm{N_r}$ had the concentration as a second driver, which was approximately 24%. Consequently, there are additional factors controlling the biosphere-atmosphere exchange of total reactive nitrogen, which may be of chemical nature and challenging to quantify. We revised the mentioned sentence and expanded the discussion with certain aspects written here.

**Comment R2.13** *L. 493-494: This sentence is misleading as the phase shift is obviously not the only cause for steep decay in the high-frequency range. Rather it would be important to state here what the percentage contributions of both transfer functions are to the overall alpha values.*
**Response to R2.13** We rephrased the sentence to clarify the statement: "The inclusion of the shift mismatch in Eq. (3) leads to a steeper slope in the empirical transfer function and variations around zero at higher frequencies (see Fig. 1) compared to a first-order function alone (not shown). If $\alpha$ is calculated without phase-shift effect, we get an overestimation of the damping up to 10% for both sites. This could be expected and indicates that most of the damping is related to a time shift."

**Comment R2.14** *L. 516-518: From the results shown in Fig. 6, there seems to be clear differences between stable/unstable conditions for both sites, as well as a dependency with wind speed at a BOG site. To me, these differences/trends/dependencies - despite some uncertainties should be mentioned in the conclusions.*
**Response to R2.14** We agree that the mentioned lines sound quite general (lines 518-520) and therefore, added some details to the description.
"In case of the empirical methods, we found a wind speed dependency on damping factors ($\alpha$), apparently a linear decrease in $\alpha$ with increasing wind speed at BOG. However, the trend is limited to wind speeds higher than $1.5\,\mathrm{ms^{-1}}$. At FOR, $\alpha$ of IOG, sICO, and ICO seem to be invariant to changes in wind speed. For unstable cases $\alpha$ values are rather constant at FOR ($\sim 0.85$). At BOG, $\alpha$ of IOG and ICO were similar and vary between 0.6 and 0.8 at unstable conditions, whereas sICO values were higher by approximately 0.05-0.15. The expected decline of $\alpha$ with increasing stability was only observed in sICO at both sites, probably related to the usage of Kaimal cospectra. IOG and ICO showed no clear trend for stable cases."

**Additional comments**

**Comment R2.15** *L. 34-35: Add that these measurements were mainly for NH3.*

**Response to R2.15** Revised.

**Comment R2.16** *L. 36: "...which have...": from the sentence structure it is not clear that the authors are referring to the QCL and TRANC analyzers. I suggest making a new sentence here.*
**Response to R2.16** Revised and changed to: "Both measurement systems..."

**Comment R2.17** *L. 79-83: What were the concentration ranges of Nr species observed at the site? They should be mentioned as well as it was done for the forest site.*
**Response to R2.17** Information about the concentration levels were appended to the site characteristics of the semi-natural peatland site (line 81).
"Average $NH_3$ concentrations ranged from 8 to 22 ppb, HONO was mostly below 0.1 ppb, $HNO_3$ had an average concentration of 0.04 ppb, NO was approximately 3.6 ppb, and $NO_2$ showed 8.6 ppb on average (Hurkuck et al., 2014; Zöll et al., 2016). Concentrations of NO and $NO_2$ were provided by the 'Air Quality Monitoring Lower Saxony' (Lower Saxony Ministry of Environment, Energy and Climate Protection) (for data availability please see `https://www.umwelt.niedersachsen.de/startseite/themen/luftqualitat/lufthygienische_uberwachung_niedersachsen/aktuelle_messwerte_messwertarchiv/`)."

**Comment R2.18** *L. 84-86: To make it easier to follow, I would state which of the previously mentioned compounds is converted at each step.*
**Response to R2.18** We thank the reviewer for his/her advise and added the following information to the revised manuscript (Line 85 and Line 86):
"Inside the FeNiCr tube, $NH_4NO_3$ is thermally split up to gaseous $NH_3$ and $HNO_3$. $HNO_3$ is thermally converted to $NO_2$, $H_2O$, and $O_2$. $NH_3$ reacts at a platinum gauze with $O_2$ to NO and $H_2O$. HONO is thermally split up to NO and a hydroxyl radical. [...] resulting in a reduction of the remaining nitrogen compounds, $NO_2$ and other higher nitrogen oxides, to NO inside the gold tube."

**Comment R2.19** *L. 89: Since the sensor separation distance is very critical for the presented study, I suggest referring already here to Table 1.*
**Response to R2.19** Revised.

**Comment R2.20** *L. 119-124: Since the time lag determination is influenced by the damping of high frequencies, it would be important to mention here what the observed variation of time lags were.*
**Response to R2.20** According to the reviewer's suggestion, we appended the additional information written here to line 124.
"The chosen range for the time lag computation coincides with range of the highest time lag density. The variation of time lags around the physical lag were almost constant for both measurement campaigns and not correlated to the temporal variation of the damping factors. The difference in ranges may be related to different site characteristics, different mixing ratio fluctuations of $\Sigma N_r$ compounds at the sites, and performance of the TRANC-CLD setup."

**Comment R2.21** *L. 124: Did the authors use the range of time lag computation as filtering criteria? If so, this should be stated here clearly since is it not stated in Sect. 3.1.*
**Response to R2.21** The reference to Sec. 3.1 in the discussion manuscript is misleading and was deleted. Of course, the time lag window was used for the whole analysis. In the revised manuscript the time lag criterion used for the slope distribution is identical to the one used for

the determination of $\alpha$.

**Comment R2.22** *L. 140-141: Use "see above" instead of Sect. 2.2.*
**Response to R2.22** Revised.

**Comment R2.23** *L. 164: Since the authors describe the response of a first-order system, better describe tau_r as the "time constant", since the "response time" can be interpreted as a multiple of that.*
**Response to R2.23** Based on the comments of Reviewer 1 and Reviewer 2 to the same topic, we added some text in order to clarify the difference between the analyzer response time $\tau_{r,a}$ and the empirical cospectral response time $\tau_r$ used in this manuscript. Since the term "response time" is well established to characterize instrument performance, we consider it as better suited for the present purpose than the term "time constant", which is very unspecific. In addition, it could be misleading, since $\tau_r$ is not really constant, considering the results of Fig. 7. Also, Moravek et al. (2019) measured a significant variability in $\tau_{r,a}$ during their measurement campaign.

**Comment R2.24** *L. 184-185: I suggest to denote TF_R and TF_deltaR in text and use ":" at the end of the sentence*
**Response to R2.24** Revised.

**Comment R2.25** *Figure 1-6: I suggest adding the measurement site (BOG, FOR) in each subplot. This will provide easier readability without having always to refer to Figure captions. Furthermore, use the full or abbreviated site names consistently in all Figure captions.*
**Response to R2.25** We added the site names and improved the figure captions.

**Comment R2.26** *L. 218-220: These sentences sound misleading. If the authors used CO2 and H2O data from another eddy covariance setup that was installed in a certain distance from the reactive nitrogen flux measurements, then they did apply IPS to BOG data, just with an additional uncertainty.*
**Response to R2.26** We agree. We rephrased the mentioned lines according to the reviewer's recommendation:
"Using the IPS through EddyPro for $\Sigma N_r$ at BOG requires $CO_2$ and $H_2O$ measurements. Since both inert gases were not measured at $\Sigma N_r$ tower, we used $CO_2$ and $H_2O$ data from the EC setup described in Hurkuck et al. (2016), which was placed next to the $\Sigma N_r$ setup. Then, the application of IPS to $\Sigma N_r$ at BOG was done, thereby inducing additional uncertainty."

**Comment R2.27** *L. 230/Figure 2: It should be stated that the periods used to calculate the spectra in Figure 2 are different from the periods in Fig. 1.*
**Response to R2.27** Revised.

**Comment R2.28** *L. 245-246: The sentences are confusing as it sounds that another analysis was performed and it is not clear on which data set. Instead of using "We further estimated..." I suggest to connect both sentences, for example like at measurement sites for which we estimated the slope.*
**Response to R2.28** Revised. The analysis of the slope distribution was done after we had noticed that a significant amount of $Ps(\Sigma N_r)$ was affected by white noise. Thus, we applied the filtering criteria to exclude rather low fluxes, instrument performance issues, and conditions of insufficient

turbulence, which could have been responsible for the observed white noise.

**Comment R2.29** *L. 250-260: I find this description of Figure 3 difficult to read since it seems to "jump" between positive and negative slopes, different scalars and sites. To make the description more concise one could for example speak of a "bi-modal distribution" of the $CO_2$ (for both sites and Nr (in case of BOG) slopes.*
**Response to R2.29** We rearranged and rephrased the mentioned lines.

**Comment R2.30** *L.268: For alpha, either use "%" or ratio.*
**Response to R2.30** Done.

**Comment R2.31** *L. 303-304: This seems to be repetitive information as the IQR ranges of ICO and sICO were already described in L. 275-277.*
**Response to R2.31** These sentences are deleted in the revised manuscript.

**Comment R2.32** *L. 323-325: Do authors mean here a difference compared to THEO or a bias "between" IOG, ICO and sICO?*
**Response to R2.32** Here, the bias with THEO is meant. We corrected the corresponding sentence and rearranged the description sightly.

**Comment R2.33** *L. 333: Do the authors mean no significant trend of the IQR? Alpha for sICO is decreasing with increasing stability.*
**Response to R2.33** We specified the description in the revised manuscript as follows:
"During stable situations ICO, IOG, and sICO exhibit no distinct trend through all positive stability classes. Only for stability values above 0.4 a decrease in $\alpha$ is visible. However, this decline in $\alpha$ is rather uncertain, since the IQR is relatively large compared to the unstable classes and the amount of cospectra, which are attributed to stable conditions, is relatively small."

**Comment R2.34** *L.377-378: "It may be caused...": This sounds like as if the drop in the high-frequency range was caused by white noise, but it is rather that the white noise (occurring at about 1Hz as stated later.*
**Response to R2.34** We replaced "may" by "most likely". The words were chosen in such a way because we do not know, which component or combination is most responsible for the observed uncorrelated noise. For this reason, we mentioned several components, which may be sources of uncorrelated noise such as the pump or electrical components.

**Comment R2.35** *L. 350-410: As this subjection discusses several aspects of the noise effect, it should be divided into several paragraphs.*
**Response to R2.35** We agree with the reviewer's suggestion. For a better readability we separated Sec 4.1 into three subsections. The first paragraph is called "Sources of spectral noise (Lines 350-374), the second "Impact of noise on power spectra and cospectra" (Lines 374-395), and the third "Impact of noise on IPS" (Lines 395-411). The second and third paragraph were extended due to the reviewer comment R2.8.

**Comment R2.36** *L. 438-458: This paragraph discusses the general applicability of the correction methods for Nr fluxes, and less the differences between the approaches. I suggest therefore to move this discussion to the section at the end as it also relates the sections 4.3.2 and 4.3.3.*

**Response to R2.36** For implementing the discussion about the expected high-frequency damping processes (R2.8), some information of the mentioned lines are also needed, but in total we agree with the reviewer's comment. We moved parts of the text and introduced a section about the applicability of the correction methods for $\Sigma N_r$ fluxes after Sec. 4.2.3.

**Comment R2.37** *L. 471-473: In this sentence it is not clear whether alpha values from ICO or sICO were overestimated.*
**Response to R2.37** We clarified to: "...if Kaimal cospectra (sICO) were used."

[revised manuscript text omitted]

Application of the EC technique to $N_r$ or $NH_3$ is challenging, because most $N_r$ compounds are highly reactive and water soluble, and background concentrations are typically low. In close proximity to sources like stables, managed fields (Sutton et al., 2011; Flechard et al., 2013), traffic, or industry (Sutton et al., 2011; Fowler et al., 2013), compounds of $N_r$ like $NH_3$ or $NO_2$ can reach high concentrations. In the past, low-cost measurement devices like passive samplers (Tang et al., 2009), DELTA Denuder (DEnuder for Long-Term Atmospheric sampling) (Sutton et al., 2001) or wet chemistry analyzers (von Bobrutzki et al., 2010) were mainly used in $N_r$ measurement studies. However, these instruments typically have a low time resolution and require inferential modeling for estimating fluxes (e.g., Hurkuck et al., 2014). Recently, new measurement techniques for $N_r$ compounds were developed, such as quantum cascade lasers (QCL) using Tunable Infrared Laser Differential Absorption Spectroscopy (TILDAS), mainly for $NH_3$, (Ferrara et al., 2012; Zöll et al., 2016; Moravek et al., 2019) or the total reactive nitrogen converter (TRANC) (Marx et al., 2012; Ammann et al., 2012; Brümmer et al., 2013; Zöll et al., 2019) coupled to a fast-response chemiluminescene detector (CLD). Both measurement systems have a certain robustness, a high sampling frequency, and are sensitive enough to allow EC measurements of $NH_3$ or $\Sigma N_r$.

Evaluating fluxes with these closed-path EC systems leads to underestimation of fluxes due to damping in the high and low-frequency range. An EC setup, like any measurement setup, is comparable with a filter which removes high and low-frequency parts from measured signals. High-frequency losses are for example related to sensor separation (Lee and Black, 1994), air transport through tubes in closed-path systems (Leuning and Moncrieff, 1990; Massman, 1991; Lenschow and Raupach, 1991; Leuning and Judd, 1996), different response characteristics of the instruments, and phase-shift mismatching (Ammann, 1999). These processes inducing flux losses are usually described by spectral transfer functions (Moore, 1986; Zeller et al., 1988; Aubinet et al., 1999).

The magnitude of the high-frequency flux loss depends on the trace gas of interest, the experimental setup, wind speed, and atmospheric stability. In the recent literature, different estimates of flux losses due to high-frequency damping have been reported. For example Zöll et al. (2016) found flux losses of 33% for $NH_3$ at an ombrotrophic, moderately drained peatland site. Ferrara et al. (2012) used the same QCL instrument and estimated flux losses from 23% to 43% depending on the correction method. Moravek et al. (2019) proposed a new approach for correcting high-frequency flux losses of $NH_3$ measured by a QCL.

The method is based on frequently measuring the analyzer's time response. The application of this method resulted in 46% flux loss. Ammann et al. (2012) measured $\Sigma N_r$ with a TRANC-CLD system at an intensively managed grassland site and estimated flux losses between 19% and 26%. Brümmer et al. (2013) operated a TRANC-CLD system at a managed agricultural site and calculated flux losses of roughly 10%. Stella et al. (2013) calculated flux losses of 12–20% for NO and 16–25% for $NO_2$.

60 Evidently, the range and magnitude of flux losses of $\Sigma N_r$ and several compounds is quite large. Correction factors for $CO_2$ and $H_2O$ are usually lower. $CO_2$ shows for a closed-path EC setup attenuation factors from 2% up to 15% (Su et al., 2004; Ibrom et al., 2007; Mammarella et al., 2009; Burba et al., 2010; Butterworth and Else, 2018). $H_2O$ shows a stronger damping than $CO_2$ that depends on humidity and age of intake tube due to interactions of sample air water vapor with the inner tube surfaces. The corresponding flux loss varies from 10% to 42% (Su et al., 2004; Ibrom et al., 2007; Mammarella et al., 2009; Burba et al.,

65 2010). Mammarella et al. (2009) reported that strong damping (up to 40%) of $H_2O$ occurs in wintertime and during night due to high relative humidity and only 10% to 15% during summertime.

In the past decades, several methods for calculating spectral correction factors have been proposed based on theoretical cospectra (Kaimal et al., 1972; Moore, 1986; Moncrieff et al., 1997), measured power spectra (Ibrom et al., 2007; Fratini et al., 2012), and measured cospectra or ogives (Ammann et al., 2006). Some of these methods are implemented in ready-to-use eddy

70 covariance post-processing packages like EddyPro (LI-COR Biosciences, Lincoln, USA). In principle, it is possible to calculate flux losses without measuring trace gas concentrations, if all physical parameters of the setup and process losses are known. Such a method does not consider gas-specific properties and may not be suitable for highly reactive gases. In general, all these methods are optimized for inert greenhouse gases and not for $N_r$ species. It is therefore questionable if common methods for spectral correction are applicable for $N_r$ given the high reactivity and chemical characteristics of single compounds. Recently,

75 Polonik et al. (2019) found that the applied correction method depends strongly on the gas of interest ($CO_2$ and $H_2O$) and the type of gas analyzer used. They suggest that high-frequency attenuation of closed and enclosed devices measuring $H_2O$ should be corrected empirically. Consequently, common methods are not perfectly suited for dealing with specific EC setups. In this study, we test five different spectral damping correction methods for EC fluxes of $\Sigma N_r$ that were measured at two different sites using a TRANC-CLD system. We investigate (1) quantitative differences between the methods, (2) their sensitivity to the input

80 data and (3) dependencies on meteorological conditions (wind speed, atmospheric stability, etc.) and measurement height.

**2 Methods**

**2.1 Sites and experimental setup**

We analyzed data from two measurement sites. At both sites we installed a custom-built $\Sigma N_r$ converter (total reactive atmospheric nitrogen converter, TRANC) after Marx et al. (2012), a 3-D ultrasonic anemometer (GILL-R3, Gill Instruments,

85 Lymington, UK), a fast-response chemiluminescence detector (CLD 780 TR, ECO PHYSICS AG, Dürnten, Switzerland), and a dry vacuum scroll pump (BOC Edwards XDS10, Sussex, UK).

The first site (52°39'N 7°11'N, 19 m a.s.l) is a semi-natural peatland in Northwest Germany, called 'Bourtanger Moor' (BOG). It is an ombrotrophic, moderately drained bog with high ambient $NH_3$ concentrations (Zöll et al., 2016) dominating

the local deposition of $\Sigma N_r$ (Hurkuck et al., 2014). Average $NH_3$ concentrations ranged from 8 to 22 ppb, HONO was mostly below 0.1 ppb, $HNO_3$ had an average concentration of 0.04 ppb, NO was approximately 3.6 ppb, and $NO_2$ showed 8.6 ppb on average (Hurkuck et al., 2014; Zöll et al., 2016). Averaged values refer to the entire measurement campaign of the cited publications. Concentrations of NO and $NO_2$ were requested from the 'Air Quality Monitoring Lower Saxony' (Lower Saxony Ministry of Environment, Energy and Climate Protection) (for data availability please see https://www.umwelt.niedersachsen.de/startseite/themen/luftqualitat/lufthygienische_uberwachung_niedersachsen/aktuelle_messwerte_messwertarchiv/). A detailed description of the site is given in Hurkuck et al. (2014, 2016). The EC system was operated from October 2012 to mid of July 2013.

TRANC and sonic anemometer were installed at 2.50 m above ground. The sampling inlet was designed after Marx et al. (2012) and Ammann et al. (2012). The inlet tube was 15 cm long, consisted of FeNiCr, had an outer diameter of 1/4", and was actively heated from the edge of the tube. Inner temperatures were higher than 100°C. While passing through the TRANC, air samples undergo two conversion steps. The first one is a thermal pathway inside an iron-nickel-chrome (FeNiCr) alloy tube at approx. 870 °C. Inside the FeNiCr tube, $NH_4NO_3$ is thermally split up into gaseous $NH_3$ and $HNO_3$. $HNO_3$ is thermally converted to $NO_2$, $H_2O$, and $O_2$. $NH_3$ reacts at a platinum gauze with $O_2$ to NO and $H_2O$. HONO is thermally split up to NO and a hydroxyl radical. In a passively heated gold tube (approx. 300 °C) a catalytic conversion follows. Before reaching the gold tube, carbon monoxide (CO) is applied as a reducing agent resulting in a reduction of the remaining nitrogen compounds, $NO_2$ and other higher nitrogen oxides, to NO inside the gold tube. To sum it up, all $\Sigma N_r$ (except for $N_2O$ and $N_2$) are converted to NO. At the end of the converter a critical orifice was mounted, which ensured a pressure reduction at a constant flow rate of $\sim 2.0\,L\,min^{-1}$. After passing through a 12 m opaque Polytetrafluoroethylene (PTFE) tube the sample air was analyzed in the CLD with a sampling frequency of 20 Hz. The GILL-R3 was installed next to the inlet of the TRANC (Table 1). CLD and pump were located in an air-conditioned box. For further details of converter and field applications, we refer to Marx et al. (2012), Ammann et al. (2012), and Brümmer et al. (2013). It was shown that concentrations measured by the CLD are affected by water vapour due to quantum mechanical quenching. To compensate for this effect, calculated fluxes were corrected following the approach by Ammann et al. (2012) and Brümmer et al. (2013). Next to the $\Sigma N_r$ setup, another EC system for $CO_2$ and $H_2O$ measurements was placed (Hurkuck et al., 2016) using a GILL-R3 and a fast-response, open-path infrared gas analyzer (IRGA, LI-7500, LI-COR Biosciences, Lincoln, USA).

Our second site (48°56'N 13°56'N, 807 m a.s.l) was located in the Bavarian Forest (FOR) National Park, Germany. The same TRANC and sonic anemometer were mounted on different booms next to each other at a height of 30 m above ground and approximately 10 m above the forest canopy. Next to the sonic, an open-path LI-7500 infrared gas analyzer (IRGA) for measuring $CO_2$ and $H_2O$ concentrations was installed. CLD and pump were placed in an air-conditioned box at the bottom of the tower. A 45 m long, opaque PTFE tube connected the TRANC with the CLD. A critical orifice at the end of the TRANC restricted the flow to $2.1\,L\,min^{-1}$ and assured low pressure along the tube. Air temperature and relative humidity sensors (HC2S3, Campbell Scientific, Logan, Utah, USA) were mounted at four different heights along a vertical gradient (10, 20, 40 and 50 m). The site was located in a remote area, next to the Czech border, with no local industrial and agricultural emission hotspots (Beudert et al., 2018). Therefore, concentrations of $N_r$ species such as $NH_3$ (1.3 ppb), NO (0.4-1.5 ppb), and $NO_2$

(1.9-4.4 ppb) were very low (Beudert and Breit, 2010). A detailed description of the forest site can be found in Zöll et al.
125    (2019). For the attenuation analysis, data from June 2016 to end of June 2018 were selected. Important site-specific parameters
of both measurement sites are listed in Table 1. Table 2 gives an overview about abbreviations used in this study.

**Table 1.** Physical parameters of the EC-setups

| Parameter | Bourtanger Moor (BOG) | Bavarian Forest (FOR) |
|---|---|---|
| canopy height | 0.4 m | 20 m |
| measurement height (from ground) | 2.5 m | 31 m |
| displacement height | 0.268 m | 13.4 m |
| tube length | 12 m | 48 m |
| tube diameter (OD) | 6.4 mm | 6.4 mm |
| flow rate | 2.0 $\text{Lmin}^{-1}$ | 2.1 $\text{Lmin}^{-1}$ |
| horizontal sensor separation | 5 cm | 32 cm |
| vertical sensor separation (below the sonic) | 20 cm | 20 cm |
| sonic path length | 15 cm | 15 cm |
| CLD analyser response time ($\tau_{r,a}$) | 0.3 s | 0.3 s |
| acquisition frequency | 20 Hz | 10 Hz |
| kinematic viscosity | $1.46 \cdot 10^{-5}\,\text{m}^2\text{s}^{-1}$ | $1.46 \cdot 10^{-5}\,\text{m}^2\text{s}^{-1}$ |
| Schmidt number for NO | 0.87 | 0.87 |
| time delay | 2.5 s | 20 s |

**Table 2.** Important terms and corresponding shortcuts used in this study.

| Parameter or Term | Abbreviation |
|---|---|
| Theoretical damping calculation | THEO |
| In-situ cospectral method | ICO |
| Semi in-situ cospectral method | sICO |
| In-situ ogive method | IOG |
| In-situ power-spectral method | IPS |
| (Power) spectrum | Ps(..) |
| Cospectrum | Co(..) |
| Ogive | Og(..) |
| Transfer function | TF |
| Response time | $\tau_r$ |
| Damping factor | $\alpha$ |
| Bourtanger Moor (semi-natural peatland) | BOG |
| Bavarian Forest (mixed forest) | FOR |
| Total Reactive Atmospheric Nitrogen Converter | TRANC |
| Chemiluminescence dectector | CLD |

**2.2 Calculation and quality selection of fluxes and spectra**

[revised manuscript text omitted]

$$\alpha = \frac{\overline{w's'}^{\mathrm{m}}}{\overline{w's'}} = \frac{\int_{f=0}^{\infty} Co_{\mathrm{w,s}}^{\mathrm{m}}(f)\,\mathrm{d}f}{\int_{f=0}^{\infty} Co_{\mathrm{w,s}}(f)\,\mathrm{d}f} = \frac{\int_{f=0}^{\infty} TF(f)Co(f)\,\mathrm{d}f}{\int_{f=0}^{\infty} Co(f)\,\mathrm{d}f} \tag{1}$$

The flux attenuation factor is the ratio of the measured flux covariance $\overline{w's'}^{\mathrm{m}}$ of vertical wind $w'$ and scalar $s'$ to the true covariance $\overline{w's'}$, where the prime denotes fluctuations of the scalars. $\overline{w's'}$ is evaluated by the integral of $Co(f)$ over the frequency. Also, $\overline{w's'}^{\mathrm{m}}$ can be expressed by the integral of $Co(f)$ over the frequency, but it has to consider a transfer function. $TF$ is the overall spectral transfer function of the EC setup and is usually a product of several individual damping processes with specific transfer functions $TF_{\mathrm{i}}$. In the following subsections we describe the methods in detail.

**2.3.1 Theoretical damping calculation [THEO]**

The theoretical damping calculation [THEO] is the most commonly applied method (Spank and Bernhofer, 2008). It is independent of any measured data and works for open-path as well as closed-path EC systems (Leuning and Moncrieff, 1990; Lenschow and Raupach, 1991; Massman, 1991; Leuning and Judd, 1996; Moncrieff et al., 1997). It is based on the assumption, that all relevant attenuation processes are known and can be quantitatively described by spectral transfer functions $TF_{\mathrm{i}}$. Detailed descriptions of the $TF_{\mathrm{i}}$ are given in Moore (1986); Moncrieff et al. (1997); Ammann (1999); Aubinet et al. (1999, 2012). The $TF_{\mathrm{i}}$ and physical parameters for the EC setups used here, like the analyser response time $\tau_{\mathrm{r,a}}$, flow rate, tube length, and sensor separation, are listed in Tables A1 and 1. All $TF_{\mathrm{i}}$ were merged to a single total transfer function ($TF_{\mathrm{theo}}$), which was applied to theoretical (modified) Kaimal cospectra (from the original (Kaimal et al., 1972)). Subsequently, $\alpha$ was calculated after Eq. (1) for every quality selected flux averaging interval. Kaimal cospectra exclusively depend on stability, wind speed and measurement height above canopy (Moore, 1986; Ammann, 1999). Further in-situ measurements were not used for this approach.

In order to prevent a misunderstanding between $\tau_{\mathrm{r,a}}$ and the later introduced parameter $\tau_{\mathrm{r}}$, we state differences of them here. Physically, the analyser response time $\tau_{\mathrm{r,a}}$ represents the time, at which the difference between the measured signal and the measured quantity is reduced by 1/e after a step change. Thus, it is also called e-folding time. If it is zero, changes will be recognized instantaneously. This is mostly not possible for common gas analysers. Our TRANC-CLD system, which has proven to be suitable for EC measurements (Marx et al., 2012; Brümmer et al., 2013), has an e-folding time of about 0.3-0.35 s. $\tau_{\mathrm{r,a}}$ is used for the first-order filter transfer function (Table A1) in the THEO approach. In this manuscript $\tau_{\mathrm{r}}$, which is also called response time, is a fitting parameter used in equation (2). It is linked to the cut-off frequency $f_{\mathrm{c}} = 1/2\pi\tau_{\mathrm{r}}$, at which the cospectrum is damped to $1/\sqrt{2} \approx 0.71$ or the power spectrum to 50%.

**2.3.2 In-situ cospectral method [ICO]**

Theoretical cospectra could deviate from site-specific characteristics of the turbulent transfer, while theoretical transfer functions could miss important chemical or microphysical processes, which are more important for $\Sigma N_r$ than for inert gases like
210    $CO_2$, $H_2O$, $CH_4$, or $N_2O$. In the exemplary case of Fig. 1, the prescribed cospectrum of Kaimal  corresponds generally well with $Co(w,T)$, but a systematic deviation may exist in the low-frequency range for BOG. At both sites, differences to $Co(w, N_r)$ are also visible in the high-frequency range right of the cospectral maximum which is around 0.2 Hz for BOG and around 0.02 Hz for FOR in the present example.

[Figure]

**Figure 1.** Comparison of observed normalized cospectra with modified Kaimal cospectra (green) for similar wind speed and stability and their theoretical and experimental transfer functions at BOG ((a),(c))  ($\zeta$ = -0.23, $\bar{u}$ = 1.38 ms$^{-1}$) and FOR ((b),(d))  ($\zeta$ = 0.17, $\bar{u}$ = 2.04 ms$^{-1}$). Panels (c) and (d) show the theoretical cospectral transfer function ($TF_{theo}$) (black) and the experimental transfer function ($TF_{exp}$) (red). The experimental transfer functions were determined with the cospectra in (a) and (b). The displayed cospectra of heat (red) and $\Sigma N_r$ mass flux (blue) are averaged over half-hourly measurements on 10.10.2012 between 09:30 and 14:00 and on 28.10.2016 between 10:00 and 15:30 for BOG and FOR, respectively. The choice of different days was caused by data gaps in the measurements.

Cospectra of FOR are shifted to the left due to the larger measurement height above canopy and the increased contribution of low-frequentcy, large-scale eddies with height (Burba, 2013). The wind speed and stability values of the shown example are in close agreement with long-term, daytime averages of the corresponding sites. On average, wind speed and stability were approximately $1.65\,\mathrm{ms^{-1}}$ and -0.22 at BOG during daytime. At FOR, the average wind speed and stability were $1.91\,\mathrm{ms^{-1}}$ and -0.44 during daytime. Wind speed conditions of the averaged power spectra and cospectra are almost similar to the average values during daytime for the entire period. Stability values of the displayed case are in agreement with daytime average for BOG. At FOR, the shown example refers to stable conditions whereas an unstable average is exhibited during daytime. In general, daytime stability values of both sites are rather low and close to neutral conditions. At both sites, approximately 10% of the analyzed cospectra were in the range of $\pm 0.5\,\mathrm{ms^{-1}}$ to the average wind speed and $\pm 0.15$ to the average stability. Using only the wind speed restriction resulted in 40% agreement at FOR and 55% at BOG. It seems that the stability is more diverse and not correlated to wind speed. The correlation between wind speed and stability for the analyzed cospectra used for the damping analysis is rather low for both sites (0.26 for BOG and 0.15 for FOR). In conclusion, the shown example represents a common case of the selected cospectra, which were used for the empirical approaches, especially for wind speed.

The in-situ cospectra method (ICO) utilizes $\mathrm{Co}(w,T)$ instead of the Kaimal cospectrum in (Eq. 1). $\mathrm{Co}(w,T)$ is used as reference cospectrum, because it is almost unaffected by damping processes. Assuming spectral similarity between $\mathrm{Co}(w,T)$ and $\mathrm{Co}(w,\Sigma\mathrm{N_r})$, we can derive $TF_{\mathrm{exp}}$ as follows (Aubinet et al., 1999; Su et al., 2004):

$$\alpha \cdot \frac{\mathrm{Co}(w,\Sigma\mathrm{N_r})}{\overline{w'\Sigma\mathrm{N_r}'}} = TF_{\mathrm{exp}} \cdot \frac{\mathrm{Co}(w,T)}{\overline{w'T'}} \tag{2}$$

In principle, this equation compares the ratio of the cospectra, which corresponds to cospectral transfer function, to the empirical transfer function $TF_{\mathrm{exp}}$. Equation (2) allows us to determine $\tau_{\mathrm{r}}$. $TF_{\mathrm{exp}}$ consists of a first-order filter $TF_{\mathrm{R}}$ combined with a mismatching phase-shift $TF_{\Delta\mathrm{R}}$ for first-order systems (Ammann, 1999) (Table A1).:

$$TF_{\mathrm{exp}}(f) = TF_{\mathrm{R}}(f) \cdot TF_{\Delta\mathrm{R}}(f) \tag{3}$$

The approach used in this study is somewhat different to other methods that are also based on using measured cospectra of heat and gas flux, for example the method of Aubinet et al. (1999). The latter uses a normalisation factor, which corresponds to the ratio of the heat flux cospectrum to gas flux cospectrum. Both cospectra are integrated until frequency $f_{\mathrm{o}}$, which should not be affected by high-frequency damping, but high enough to allow an accurate calculation of the normalisation factor. However, the definition of $f_{\mathrm{o}}$ is rather imprecise and thus, an incorrect setting of $f_{\mathrm{o}}$ can lead to significant uncertainties in the damping analysis. In our approach cospectra are normalized by their corresponding total covariance. In order to consider the damping of the gas flux cospectrum and its covariance, the damping factor is introduced in Eq. (2). Thus, we assume that both approaches give similar results, since both approaches cover the damping of the gas flux cospectrum. The procedure of solving Eq. (2) is not straightforward. Thus, a flow chart of the important calculation steps is shown in Fig. 2.

[Figure]

**Figure 2.** Illustration of the calculation of $\alpha$ and $\tau_r$ by ICO.

The iteration was started with $\alpha_0 = 1$. Afterwards, a non-linear least-square fit of Eq. (2) was performed. For minimizing
both sides of Eq. (2)  $\tau_r$ was used as optimization parameter. After $\tau_r$ was calculated, $TF_{exp}(f)$ could be determined
and inserted into Eq. (1). $\alpha_1$ was estimated by Eq. (1) using $Co(w, T)$ as reference. Finally, the process was terminated, if
the difference between the first guess and $\alpha_1$ was sufficiently low ($< 0.035$). Otherwise, the whole process had been repeated.
Equation (2) was solved iteratively until $\alpha$ converged. Our experience was that three iteration steps were mostly enough to
fulfill the termination criterion.  The non-linear fit (Eq. 2) was done for frequencies larger
than 0.055 Hz for the BOG campaign. This frequency range is assumed to be affected by damping effects. A similar frequency
limit had been used in the damping analysis of Zöll et al. (2016) for the same site. For the FOR campaign the lower frequency
limit was set to 0.025 Hz. The decision of the lower frequency limits were further proven by the examination of the ogives
ratio, which shows constant values in a certain frequency range. The position exhibits the frequency, at which high-frequency
attenuation mostly starts to increase.
 Panel (c) and (d) of Fig. 1 show examples of the theoretical and
experimentally determined transfer functions for the two measurement sites. In both cases the experimental transfer function
drops earlier than the theoretical and reveals a significant variation in the high-frequency range.

**2.3.3 Semi in-situ cospectra method [sICO]**

The semi in-situ cospectra approach is similar to the one described in Sec. 2.3.2. The determination of $\tau_r$ follows the same procedure as for ICO, but instead of using $Co(w, T)$ in Eq. (1), this approach uses Kaimal cospectra (Eqs. (A1) and (A2)) as reference. This method is useful, if the quality of $Co(w, T)$ is not sufficient for estimating the damping factors, especially in the low-frequency range.

**2.3.4 In-situ ogive method [IOG]**

The in-situ ogive method (IOG) is based on Ammann et al. (2006) and Ferrara et al. (2012). An ogive is defined as the cumulative integral of the cospectrum from the lowest frequency $f_0$, which is given by the averaging interval, to the highest frequency, the Nyquist frequency $f_N$. The Nyquist frequency is the half of the sampling frequency.

$$\mathrm{Og}(f) = \int_{f_0}^{f_N} \mathrm{Co}(f)\, \mathrm{d}f \tag{4}$$

This method is similar to ICO, but does not rely on a specific form for the spectral transfer functions or cospectra and only requires $\mathrm{Og}(w, T)$ and $\mathrm{Og}(w, \Sigma N_r)$. Again, spectral similarity between $\mathrm{Og}(w, T)$ and $\mathrm{Og}(w, \Sigma N_r)$ is assumed. For estimating the damping, a linear regression between $\mathrm{Og}(w, T)$ and $\mathrm{Og}(w, \Sigma N_r)$ was performed in a specific frequency range. The range was constrained by frequencies for which $\mathrm{Og}(w, T) > 0.2$ and $\mathrm{Og}(w, \Sigma N_r) < 0.85$ was fulfilled. Frequencies lower than 0.002 Hz were excluded. The difference between the regression line and $\mathrm{Og}(w, \Sigma N_r)$ was calculated, and points exceeding a difference of 0.1 or frequencies above which the signal is totally damped, were not considered for a linear least-square fit of $\mathrm{Og}(w, \Sigma N_r)$ against $\mathrm{Og}(w, T)$. The former criterion was applied for discarding spikes. Finally, the optimization factor, which minimizes the difference between $\mathrm{Og}(w, \Sigma N_r)$ against $\mathrm{Og}(w, T)$, is the result of the least-squares problem and corresponds to the damping factor.

**2.3.5 In-situ power-spectral method [IPS]**

Application of the in-situ power spectral method (IPS) after Ibrom et al. (2007) was executed using EddyPro. It uses measured power spectra of a reference scalar and of the trace gas of interest, here $\mathrm{Ps}(T)$ and $\mathrm{Ps}(\Sigma N_r)$. The first step - the estimation of $\tau_r$ or the cut-off frequency $f_c$ - is similar to the in-situ cospectra method (Eq. 2), but the transfer function is different.

$$\frac{\mathrm{Ps}(\Sigma N_r)}{\mathrm{Ps}(T)} = \frac{1}{1 + (f/f_c)^2} \tag{5}$$

For estimating $f_c$ Eddy Pro uses quality-selected and averaged power spectra. We set 0.4 Hz as lowest noise frequency in the option 'removal of high frequency noise' and adjusted the threshold values for removing power spectra and cospectra from the analysis accordingly. The value for the 'lowest noise frequency', which was set in EddyPro for running IPS, was a subjective decision based on visual screening through power spectra. Therefore, we calculated slopes of $\Sigma N_r$ power spectra in the inertial subrange and estimated the frequency, at which noise started to increase and slopes got positive. Additionally, we

forced EddyPro to filter the spectra after statistical (Vickers and Mahrt, 1997) and micrometeorological (Mauder and Foken, 2004) quality criteria. We applied the correction of instrument separation after Horst and Lenshow (2009) for crosswind and vertical wind and took the suggested lowest and highest frequency (0.006 Hz and 5 Hz) as fitting range for $Ps(T)$ and $Ps(\Sigma N_r)$ for FOR. Applying the IPS through EddyPro for $\Sigma N_r$ at BOG requires $CO_2$ and $H_2O$ measurements. Since both inert gases were not measured at the $\Sigma N_r$ tower, we used high-frequency $CO_2$ and $H_2O$ data from the EC setup described in Hurkuck et al. (2016), which was placed next to the $\Sigma N_r$ setup. Then, the application of IPS to $\Sigma N_r$ at BOG was performed, thereby inducing additional uncertainty. We changed the highest frequency to 8 Hz and took the lowest frequency from standard settings (0.006 Hz). For comparing the results of IPS to our cospectral methods, we chose the same half-hours which passed the automatic selection criteria and the manual screening (see Sec. 2.2). In general, the idea of IPS is that the EC system can be simulated by a recursive filter. Thereby, $\alpha^{-1}$ is determined by the ratio of the unfiltered covariance $\overline{w'T'}$ to the filtered covariance $\overline{w'T_f'}$, and applying the recursive filter to degrade the time series of sonic temperature (Ibrom et al., 2007). However, Ibrom et al. (2007) argued that this ratio gives erroneous results for small fluxes. Therefore, they parameterized $\alpha$ by the mean horizontal wind speed ($\bar{u}$), stability, and $f_c$. Ibrom et al. (2007) investigated a proportionality between $\alpha^{-1}$ and $u \cdot f_c^{-1}$. By introducing a proportionality constant $A_1$ and a second constant $A_2$, which should consider for spectral properties of the time series, the following equation for calculating the correction factor was proposed (for details see Ibrom et al., 2007, Sec. 2.4):

$$\alpha^{-1} = \frac{A_1 \bar{u}}{A_2 + f_c} + 1 \tag{6}$$

$A_1$ and $A_2$ were estimated for stable and unstable stratification using degraded time series of sonic temperature. The degradation was done by a varying low pass recursive filter (Ibrom et al., 2007; Sabbatini et al., 2018). A general  summary of processing eddy-covariance data including high-frequency spectral correction methods is given in Sabbatini et al. (2018).

**3 Results**

**3.1 Characterization of power spectra and cospectra**

Figure 3 shows exemplary cospectra and power spectra of the two measurement sites. We compare cospectra which were measured during unstable daytime conditions and at similar wind speed. All in all, the cospectral densities of the gas and heat fluxes are quite similar. It indicates that the chosen sampling interval and frequency were sufficient to capture flux-carrying eddies. However, $Co(w, \Sigma N_r)$ shows a stronger variation than the other cospectra. The effect of different measurement height is quite obvious. It results in a shift of all cospectra to the left for the FOR site. The stronger drop of $Co(w, \Sigma N_r)$ compared to $Co(w, CO_2)$ and $Co(w, H_2O)$ in the high-frequency range is likely related to damping by the $\Sigma N_r$ inlet tubes, which did not affect the $CO_2/H_2O$ open-path measurements. It also appears that the damping (difference of cospectra in the high-frequency range) at BOG is higher than the one at FOR for the selected averaging interval.

[Figure]

**Figure 3.** Normalised cospectra and power spectra of $T$ (red), $\Sigma N_r$ (blue), $CO_2$ (green) and $H_2O$ (orange)  at BOG ((a),(c)) and FOR ((b),(d)). (Co)spectra were averaged at BOG from 11.10.2012 09:00 to 11.10.2012 16:30 ($\zeta$=-0.31, $\bar{u}$=1.36 ms$^{-1}$)  and at FOR from 16.10.2016 10:00 to 16.10.2016 15:30 ($\zeta$=-3.27, $\bar{u}$=1.89 ms$^{-1}$) . $CO_2$ and $H_2O$ (co)spectra of BOG were adjusted to the aerodynamic measurement height of the $\Sigma N_r$ setup. Note that the time period used for averaging is different from the periods of Fig. 1.

The shapes of the power spectra for $T$, $CO_2$ and $H_2O$ are comparable to those found in other studies (e.g., Ammann, 1999; Ibrom et al., 2007; Rummel et al., 2002; Aubinet et al., 2012; Ferrara et al., 2012; Fratini et al., 2012; Min et al., 2014). For Ps($T$) a slope of -0.62 (BOG) and -0.63 (FOR) was determined in the inertial subrange. The fitting range used for the derivation of the slopes is smaller than the inertial subrange, for example, to exclude slightly positive slopes of the inert trace gases at the very high frequencies. Differences to the theoretical shape, -2/3 for power spectra, may be related to slight damping of Ps($T$) in the high-frequency range. A slight high-frequency damping of Ps($T$) can be caused by the path averaging of the sonic anemometer (e.g. Moore, 1986). In addition, the observed shape of the spectrum (slope) can deviate from the theoretical shape due to non-ideal environmental conditions (e.g. non-homogeneous turbulence, influence of roughness sublayer). The stronger

drop of $Co(w, \Sigma N_r)$, compared to $Co(w, CO_2)$ and $Co(w, H_2O)$ in the high-frequency range, is likely related to damping by the tubes, which is not relevant for open-path instruments. $Ps(CO_2)$ and $Ps(H_2O)$ have nearly the same slope in the inertial subrange and exhibit the excepted shape. In contrast, $Ps(\Sigma N_r)$ is lower than $Ps(CO_2)$ and $Ps(H_2O)$ at lower frequencies

330   $(< 0.1\,\text{Hz})$ , starts to rise afterwards., and reaches a maximum  around 1 Hz. This phenomenon was found in almost all $Ps(\Sigma N_r)$ at the measurement sites.  for which we estimated the slope of $Ps(\Sigma N_r)$ in the high-frequency range. However, the amount of $Ps(CO_2)$, which were affected by this phenomenon was rather small compared to $Ps(\Sigma N_r)$. For an in-depth investigation of slope issue we applied a variance filter of $w$, $T$, $\Sigma N_r$ and excluded PS, if the variance was higher than 1.96 $\sigma$, which corresponds to confidence limit of 95%. Additionally, we excluded

335   low-quality fluxes (flag=2) of sensible heat and $\Sigma N_r$ after Mauder and Foken (2006) and applied the time lag filtering criteria. These criteria were used to exclude periods of rather low fluxes, instrument performance issues, and conditions of insufficient turbulence. We used equivalent filtering criteria for $CO_2$ and additionally applied a precipitation filter due to the open-path characteristics of the LI-7500. The precipitation filter was also applied for filtering the lower quality cases of $CO_2$ and $H_2O$ shown in Fig. 3. Figure 4 shows a distribution of the estimated slopes at both measurement sites.

[Figure]

**Figure 4.** Distribution of spectral slopes in the high-frequency range $(> 0.1\,\text{Hz})$ of $Ps(\Sigma N_r)$ (green), $Ps(CO_2)$ (black) and $Ps(T)$ (red) for the BOG site (a) and for the FOR site (b). Slopes were estimated for half hourly power spectra from 02.10.2012 to 17.07.2013 and from 01.06.2016 to 28.06.2018 at BOG and FOR, respectively.

340   The slopes of $Ps(T)$ are between -0.5 and -0.7, which is close to the theoretical value, and the shape of the histogram seems to be narrower around the theoretical value at BOG than at FOR. The distribution of the $Ps(CO_2)$ slopes is rather bi-modal at

BOG, but coincides well with the slope shape of the Ps($T$) slopes at FOR. In volume terms, most slopes of Ps($CO_2$) are negative at both sites (70% for BOG and nearly all for BOG (95%)), but their maximum is slightly higher than -2/3 (-0.53 for BOG and -0.58 for FOR). More Ps($CO_2$) slopes of BOG exhibit a positive slope between 0.50 and 0.75 (24%) than the Ps($CO_2$) slopes of FOR (2%) in the same range. In contrast, the slopes of Ps($\Sigma N_r$) are mostly positive at both sites (88% at BOG and 97% at FOR). Also at BOG, the slopes of Ps($\Sigma N_r$) exhibit a slight bi-modal distribution. The second maximum is observed at around -0.45. The amount of Ps($\Sigma N_r$) slopes around -2/3 is rather small at BOG (less than 10% are lower than -0.25) and even negligible at FOR (less than 1% are lower than -0.25). A positive slope for nearly all Ps of a certain trace gas is rather unexpected. ~~Therefore we did a similar analysis for Ps($CO_2$). We used equivalent filtering criteria (see above) and additionally applied a precipitation filter due to the open-path characteristics of the LI-7500. In general, most slopes of Ps($CO_2$) are negative in contrast to Ps($\Sigma N_r$). At BOG 70% of Ps($CO_2$) show a negative slope and at FOR nearly all slopes of Ps($CO_2$) are negative (95%). Furthermore, slopes of Ps($CO_2$) are closer to the slopes of Ps($T$) than Ps($\Sigma N_r$), but their maximum is slightly higher than -2/3 (-0.53 for BOG and -0.58 for FOR). The shape of Ps($CO_2$) coincides well with the shape of Ps($T$) at both sites, but the agreement of Ps($CO_2$) with Ps($T$) was slightly better at the forest site. Positive slopes of Ps($CO_2$) were detected at both sites. More Ps($CO_2$) of BOG exhibit a positive slope between 0.50 and 0.75 (24%) than the Ps($CO_2$) of FOR (2%) in the same range.~~

**3.2 Comparison of different damping correction methods**

In the following, we present the results of the damping correction methods introduced in Sec. 2.3. Firstly, we describe the results of the in-situ power spectral method (IPS) and the four cospectral methods. Secondly, we demonstrate findings of dependencies on meteorological variables. Figures 5 and 6 show  statistical analyses of $\alpha$ which were calculated by each method on monthly (BOG) or bimonthly (FOR) basis depicted as box plots. It was possible to estimate $\alpha$ with all methods for 816 half-hours for BOG and 811 half-hours for FOR. All damping correction methods were evaluated for the same half-hours.

[Figure]

**Figure 5.** Boxplots of the flux damping factor ($\alpha$) for BOG without whiskers and outliers (box frame = 25 % to 75 % interquartile range (IQR), bold line = median). The number of observations which are displayed  at the top of the plot are the same for every method.

[Figure]

**Figure 6.** Boxplots of the flux damping factor ($\alpha$) for FOR without whiskers and outliers (box frame = 25 % to 75 % interquartile range (IQR), bold line = median). The number of observations which are displayed  at the top of the plot are the same for every method.

Monthly $\alpha$ calculated with the IPS method show no temporal drift at FOR (Fig. 6). The median $\alpha$ is around 0.95 for
365   nearly every month. Additionally, the interquartile range (IQR; 25 to 75 %-quartile) is very small (0.01 to 0.02). At BOG, monthly median $\alpha$ calculated with IPS were also mostly around 0.95, only the first three month were sightly lower by $\sim$ 0.04. Their  is around 0.04 on average. It is obvious that $\alpha$ of IPS is the highest compared to the cospectral methods and they exhibit the lowest IQR during the measurement period.

Atbothsites, the median $\alpha$ of the in-situ cospectral methods ICO, sICO, and IOG show only moderate temporal variations
370   during the entire measurement campaigns. While slightly higher values in summer and lower values in winter were found at the FOR site (Fig. 6), the opposite pattern was observed at the BOG site (Fig. 5). Their IQR is more variable and ranges from 0.13 to 0.26 at BOG and from 0.16 to 0.31 at FOR. Changes in the range of the IQR and fluctuations of the medians may be related to different meteorological conditions, to changes in composition of $\Sigma N_r$, or to a degeneration of instrumental response. During field visits for maintenance, parts of the TRANC like the heating tube or platinum gauze were exchanged
375   or cleaned, which could influence the results. At both sites, $\alpha$ by THEO were always higher than in-situ cospectral methods (IOG, ICO, sICO) and their medians were about 0.90 at BOG and about 0.95 at FOR. Their IQR is smaller than IOG, ICO and sICO, too.

At FOR, the median $\alpha$ of ICO and sICO are similar for every month showing a difference of 0.03 on average, and their IQR cover mostly the same range (Table 3 and Fig. 6). Values for $\alpha$ by IOG are mostly higher and exhibit a difference of 0.06 on average to sICO and ICO. The IQR by IOG is roughly half of the IQR of ICO and sICO (Table 3). During the months of December in 2016 and 2017, as well as January in 2017 and 2018, and April to May in 2018, IQR of ICO and sICO is relatively large. Common to both periods  the average vertical wind was quite low in January 2017 and 2018 (less than $0.01\,\mathrm{m\,s^{-1}}$). Additionally, we had some instrumental performance problems (exchange of the pump and heating tube, power failure) with the TRANC in the mentioned months. As mentioned in Sec.2.2, these periods were not considered in the flux analysis. As a matter of fact, not all affected fluxes can be excluded by the selection criteria. Thus,  an influence on the quality of the cospectra/ogives can not be excluded. Consequently, IOG, ICO and sICO exhibit a wide IQR from 0.15 to 0.40 and differences in the median from 0.06 to 0.16 which could be related to the low number of valid cospectra/ogives. Therefore, classifying $\alpha$ at FOR bimonthly (Fig. 6) was a needed approach to enhance the quality, when the amount of valid cospectra is not enough for a robust estimation of $\alpha$. Overall, a good agreement of IOG, ICO and sICO was found.

At BOG, the median $\alpha$ of ICO are the lowest and the median $\alpha$ of sICO and IOG are nearly the same for every month (Table 3 and Fig. 5). The difference of ICO to IOG varies by 0.05 and 0.20 and to sICO by 0.02 and 0.18. A systematic difference in $\alpha$ between ICO and sICO was not observed for FOR. At the beginning of the measurements the difference was rather small, but it started increasing after December 2012. The range of the quartiles is similar for IOG and sICO for certain months (see Table 3 and Fig. 5), but their IQR is lower than the IQR of ICO. Again, the IQR of IOG is roughly half of ICO IQR. It seems that theoretical cospectra could not reproduce the shape of $\mathrm{Co}(w,T)$ well under certain site conditions, although $\tau_\mathrm{r}$ of sICO and ICO were quite similar. They show a correlation of 0.75 and an average absolute difference of 0.48. Comparing $\alpha$ between the sites shows that the damping is stronger at BOG than at FOR. Table 3 shows the averaged $\alpha$ at FOR and BOG.

By subtracting $\alpha$ from an ideal, unattenuated system, which has an damping factor of one, the result will be the flux loss value ($=1$-$\alpha$). This loss value shows how much of the signal is lost from the inlet to the analysis of the signal by the instrument. Thus, flux losses calculated by IPS for our TRANC-CLD setup are around 6% at BOG and around 5% at FOR. The flux loss after THEO was approximately 12% at BOG and about 5% at FOR. The methods using measured cospectra or ogives (ICO, sICO and IOG) showed a flux loss of roughly 16-22% for FOR and around 26-38% for BOG. ICO shows the strongest damping at both sites.  These values are in common with other EC studies conducted on $\Sigma\mathrm{N_r}$) and other reactive nitrogen compounds (Ammann et al., 2012; Ferrara et al., 2012; Brümmer et al., 2013; Stella et al., 2013; Zöll et al., 2016)(Moravek et al., 2019).

**Table 3.** Averages of monthly medians, lower and upper quartiles of $\alpha$ over the whole measurement period for all applied methods at both sites.

| Site | method | median | lower quartile | upper quartile |
|---|---|---|---|---|
| Bavarian Forest | IOG | 0.84 | 0.77 | 0.90 |
| | ICO | 0.78 | 0.64 | 0.89 |
| | sICO | 0.78 | 0.63 | 0.89 |
| | THEO | 0.95 | 0.93 | 0.96 |
| | IPS | 0.95 | 0.94 | 0.95 |
| Bourtanger Moor (BOG) | IOG | 0.72 | 0.64 | 0.80 |
| | ICO | 0.62 | 0.45 | 0.76 |
| | sICO | 0.74 | 0.59 | 0.83 |
| | THEO | 0.88 | 0.85 | 0.91 |
| | IPS | 0.94 | 0.91 | 0.95 |
| Bavarian Forest (FOR) | IOG | 0.84 | 0.77 | 0.90 |
| | ICO | 0.78 | 0.64 | 0.89 |
| | sICO | 0.78 | 0.63 | 0.89 |
| | THEO | 0.95 | 0.93 | 0.96 |
| | IPS | 0.95 | 0.94 | 0.95 |

The IQR is comparable for ICO and sICO at both sites. It ranges from 0.24 to 0.31, which is caused by the large variety in the shape of measured cospectra. The IQR of IOG is roughly half as large and ranges from 0.13 to 0.16. The IQR of THEO is rather small (0.02 at FOR and 0.07 at BOG). Flux loss and its IQR estimated by IPS are the lowest of all methods.

For investigating deviations of the different methods more precisely, we computed correlation, bias and the precision as the standard deviation of the difference between two methods. The results are summarized in Table B1. IOG exhibits a bias of not more than 10%0.10 to ICO and sICO and is rather small at BOG (3%0.03). The bias and precision between sICO and ICO is lowest at FOR. Additionally, the scattering of sICO $\alpha$ is more pronounced, which results in a lower precision of sICO against the IOG $\alpha$. Common to bBoth siteshave in commom that, the correlation of IOG with sICO was inferior to ICO. Checking ICO $\alpha$ against sICO $\alpha$ demonstrates a high correlation at both sites (0.78 for FOR and 0.66 for BOG). This is excepted, since theoretical cospectra are based on $\text{Co}(w,T)$. IOG, ICO and sICO show a strong bias, low precision and nearly no correlation to THEO. The correlation between the sICO with THEO is somewhat higher because of utilizing Kaimal cospectra for both methods. IPS shows a negative bias and high precision to IOG, ICO, and sICO at FOR and 0.05 larger bias than THEO at BOG. The correlation of IPS with THEO is quite high at both sites which is reasonable, since bias and precision are quite low. Both methods give almost equal $\alpha$.

For investigating a trend in meteorological variables such as temperature, relative humidity, stability, and wind speed, we classified them into bins, calculated $\alpha$ for each bin and display them as box plots (Fig. 7). In the following figure, only wind speed and stability are shown. These are two variables for which we except a dependence, since the shape and position of a

425 Kaimal cospectrum varies with wind speed and stability. We checked for dependencies on the other variables such as global radiation, temperature and humidity, but no significant influence was found.

[Figure]

**Figure 7.** Dependency of the flux damping factor ($\alpha$) on stability and wind speed classes as box plots without whiskers and outliers (box frame = 25 % to 75 % interquartile range (IQR), bold line = median). Each damping estimation method is assigned with a different color (red: IOG, green: ICO, blue: sICO, orange: THEO, black: IPS). (a) and (b) refer to the BOG site and (c) and (d) to the FOR site.

A slight dependence on wind speed for BOG $\alpha$ is starting to be relevant at wind speeds above $1\,\mathrm{ms}^{-1}$, which is confirmed by IOG, ICO and sICO. The influence on wind speed predicted by THEO begins already at low wind speed, which means that stronger damping was found at higher wind speed values. It shows a (linear) decrease from the beginning. A bias of

430 IOG, ICO, and sICO to THEO (Fig. 7) exists for all wind speed classes. Considering the medians, we observe an increase in attenuation from 0.15 till 0.20 over the whole wind speed regime. The bias of IOG and ICO with sICO (Fig. 7) is mostly visible for wind speeds up to $1.5\,\mathrm{ms}^{-1}$ and gets negligible afterwards.

$\alpha$ values of IOG, ICO, and sICO are nearly invariant to changes in wind speed at FOR. The predicted drop due to wind
speed by THEO is roughly 0.10 at FOR. The difference of the empirical cospectral methods with THEO diminished for
wind speed larger than $4\,\mathrm{ms}^{-1}$. IPS shows the weakest $\alpha$ for all wind speed classes at both sites. The decrease of $\alpha$ with wind
speed is less than 0.10 at BOG and hence lower than the cospectral methods. IPS exhibit no significant drop in $\alpha$ with
wind speed at FOR.

Values of $\alpha$ estimated by THEO are almost equal for unstable conditions and decline for stable situations. As before, the
theoretical drop in attenuation is stronger at BOG (up to 0.20) than at FOR (not exceeding 0.10). At FOR, $\alpha$ of
IOG, ICO and sICO are nearly equal ($\sim 0.85$) for unstable cases. ~~During stable situations IOG and ICO exhibit no distinct
trend and their IQR is similar. The IQR of sICO gets wider and their values are lower than the other methods. No significant
trend is observed~~. ICO, IOG, and sICO exhibit no distinct trend through all positive stability classes. Only for stability values
above 0.4 a decrease in $\alpha$ is visible. However, this decline in $\alpha$ is rather uncertain, since the IQR is relatively large compared
to the unstable classes and the amount of cospectra, which are attributed to stable conditions, is relatively small.

At BOG,  the linear decline in $\alpha$ is given for  sICO, but does not exist for IOG and ICO.  $\alpha$ of IOG and ICO
are similar for unstable cases, but show no clear decrease with increasing stability. The IQR of the sICO increases
for positive stability and is smaller than IOG and ICO for negative values. The bias of sICO to IOG and ICO is obvious for the
negative stability values. Similar to THEO, IPS shows a drop of $\alpha$ with increasing stability at BOG, but values are higher than
for the cospectral methods. As observed for wind speed at FOR, no significant drop in $\alpha$ for IPS occurs under stable conditions.

**3.3 Analysis of response time**

After comparing $\alpha$ of the individual methods we focus on variation of $\tau_\mathrm{r}$ in time. Therefore, we show statistical
analyses of $\tau_\mathrm{r}$ of both measurement sites. Figure 8 shows  statistical analyses of $\tau_\mathrm{r}$, which were calculated by ICO
on bimonthly basis depicted as box plots.

[Figure]

**Figure 8.** Statistical analysis of the response time ($\tau_r$) depicted as box plot without whiskers and outliers (box frame = 25 % to 75 % interquartile range (IQR), bold line = median) for the BOG site (a) and the FOR site (b).

455      It is obvious that medians of $\tau_r$ of FOR are generally larger than medians of $\tau_r$ of BOG. The averaged median $\tau_r$ is 1.37 s for BOG and 3.13 s for FOR (Table B2). Common to both sites , $\tau_r$ was sightly lower at the start of the measurements and their medians were quite constant until December 2012 at BOG and October/November 2016 at FOR. Afterwards $\tau_r$ and its IQR increased significantly, especially at FOR. The variation of $\tau_r$ follows no trend and seems to be rather random. The IQR of FOR was larger, indicating that scattering of $\tau_r$ was enhanced at FOR. On average, $\tau_r$ increased 460  from 0.74 s to 1.63 s at BOG and from 1.85 s to 3.51 s at FOR (Table B2).

     We further determined the correlation between monthly averaged $\tau_r$ and $\alpha$. Correlations of -0.83 for BOG and -0.72 for FOR show that there is significant inverse relation between both parameters, which is expected due to the inverse dependency of $\tau_r$ in the empirical transfer function. The analysis of $\tau_r$ stratified by meteorological variables can be useful in order to investigate, if the scattering in $\alpha$ is related either to the variability in cospectra or to the instrument performance. $\tau_r$ is mostly a device-specific 465  parameter. It should have a higher affinity to instrument or measurement setup parameters such as measurement height, pump and heating efficiency, altering of the inlet, and sensitivity of the analyser than to atmospheric, turbulent variations. Changes in gas concentrations may also affect $\tau_r$. Therefore, we classified the meteorological parameters into bins, calculated $\tau_r$ for each bin and display them as box plots (Fig. B1). $\tau_r$ is mostly constant for medium and high wind speed at BOG and exhibits

slightly higher values at low wind speeds (0-0.5 m/s). During highly stable and unstable conditions $\tau_r$ reaches up to 3.50 s. It seems rather constant during medium, unstable conditions, but increases under stable conditions. The same is valid for $\tau_r$ at FOR. $\tau_r$ exhibits highest values under both highly unstable and stable conditions. However, $\tau_r$ is strongly affected by wind speed at FOR. It decreases with wind speed and seems to follow a non-linear relationship.

**4 Discussion**

**4.1 Noise effects on power spectra and cospectra**

**4.1.1 Sources of spectral noise**

Measured fluxes of $\Sigma N_r$ are heavily affected by white and red noise. They are caused by low and non-stationary ambient trace gas concentrations and fluxes, typically low fluxes due to weak sources and inhomogenously distributed N sources, limited resolution and precision of the CLD, and varying proportions of different $N_r$ compounds. This leads to a high rejection rate of cospectra and power spectra during quality screening, which is challenging for every spectral anaysis using in-situ measurements. While the influence on cospectra is mainly limited to low-frequency range, power spectra show systematic deviations in the low and high-frequency range. The positive slope (Fig. 3 and Fig. 4) is related to white noise which compromise the $Ps(\Sigma N_r)$ in the high-frequency domain. White and red noise are more present at FOR, because the site was located in a remote area with no nearby anthropogenic sources of $\Sigma N_r$ (Zöll et al., 2019) resulting in low concentrations of $N_r$ compounds (see Sec. 2.1). At BOG white noise is weaker since more sources of $\Sigma N_r$ were next to the EC station. As shown by Hurkuck et al. (2014), $N_r$ concentrations at BOG were relatively high and showed a distinct diurnal cycle due to intensive livestock and crop production in the surrounding region. The disturbance due to red noise is also visible in Fig. 3. The variability (scattering) of cospectra and power spectra is more pronounced at FOR than at BOG in the low-frequency range as visible in the shown example.

Some $Ps(\Sigma N_r)$ in Fig. 4, mainly at BOG, show a slope near the theoretical value of -2/3 and were not affected by white noise. Therefore, we examined the environmental conditions such as wind speed, friction velocity, concentration and flux values at that site during half-hours, which were attributed to slopes less than -0.25, and compared them to half-hours with a slope greater than -0.25. Only the distribution of concentration was different for the two regimes: Most $Ps(\Sigma N_r)$ with a slope less than -0.25 were associated with concentration values between 25 ppb and 40 ppb, whereas $Ps(\Sigma N_r)$ with a slope greater than -0.25 were associated with concentration values between 10 ppb and 25 ppb which is in common with the background concentration level of $\Sigma N_r$ at BOG. It was about 21 ppb, whereas only 5 ppb on average was measured at FOR. Thus, it seems that the concentration is an important factor for regulating the quality of $Ps(\Sigma N_r)$. The slope of $Ps(T)$ shows a clear peak between -0.5 and -0.7 for both sites, which is close to the theoretical value of -2/3. The differences in the distribution may be related to different site characteristics like surface roughness length, inhomogenous canopy height, turbulence, or to large-scale

eddies which gain more influence on the fluxes at higher aerodynamic measurement height.  Before, we argued that concentration of $\Sigma N_r$ leads to differences in the slope distribution (Fig. 4). Concentrations of $CO_2$ were not significantly different between the sites. As a consequence, there has to be another parameter responsible for discrepancy in the contribution of positive Ps($CO_2$) slopes at the measurement sites. We suppose that the discrepancy of positive Ps($CO_2$) slopes corresponds to different levels of humidity at the measurement sites. Humid conditions could reduce the sensitivity of the open-path  instrument and introduce noise in power spectra. Above the forest the air was less humid and consequentially, less Ps($CO_2$) were affected by white noise.

**4.1.2 Impact of noise on power spectra and cospectra**

Removing high-frequency variations which consist mainly of white noise is easier for Ps($CO_2$) because their signal is higher than those of Ps($\Sigma N_r$) in the low-frequency domain and the observed noise is limited to highest frequencies ($> 2\,$Hz at FOR and $> 5\,$Hz at BOG). Additionally, the noise is strictly linear and exhibits no parabolic structure such as Ps($\Sigma N_r$) (Fig. 3).  The observed parabolic shape in Ps($\Sigma N_r$), which occurs around 1 Hz, is most likely caused by uncorrelated noise, which is induced by some components of the setup like pump, air-conditioning system, or  electrical components.  and decreased towards the highest frequencies. Handling the impact of unknown noise on power spectra is challenging for common, linear noise compensation methods. Thus, it is probably not possible to remove the uncorrelated noise from Ps($\Sigma N_r$) completely.

Wolfe et al. (2018) installed an EC setup in an aircraft and measured $CO_2$, $H_2O$, and $CH_4$ with Los Gatos Research analyzers and $H_2O$ with an open-path infrared absorption spectrometer. They found a slope of $\sim 1$ in Ps($CO_2$), Ps($H_2O$), and PS($CH_4$) above 0.4 Hz, but not in the Ps($H_2O$) of the open-path analyzer. They concluded that the white noise was related to insufficient precision of the closed-path analyzers at higher frequencies. No white noise was detected in the corresponding cospectra, because it does not correlate with $w$. Kondo and Tsukamoto (2007) did $CO_2$ flux measurements above the Equatorial Indian Ocean. They concluded that white noise was related to a lack of sensitivity to small $CO_2$ density fluctuations. Density fluctuations of $CO_2$ above open ocean surfaces are much smaller than over vegetation. Similar to the present study, they detected no white noise in their $Co(w, CO_2)$. Their site characteristics and related low fluctuations of trace gas are comparable to the forest site. The latter was located in a remote area and therefore far away from potential (anthropogenic) nitrogen sources. This led to low concentrations and less variability in concentrations and deposition fluxes. Very small fluctuations of $\Sigma N_r$ are probably not detectable by the instrument. This is further confirmed by the time lag analysis we did before flux estimation. The broad shape of the empirical lag distribution around the physical lag (not shown) and the random time lag scattering demonstrated that most of the fluxes were near or below the detection limit and thus quality of (co)spectra suffered from noise. Instrumental noise also affects the shape of the covariance function. It can lead to a broadening of the

covariance peak and generally enhances the scattering of the covariance values. Both effects are already enlarged in case of small mixing ratio fluctuations. Thus, instrumental noise further compromises the time lag estimation and leads to additional noise in cospectra. Due to the applied time lag criterion, the effect of instrumental noise is mostly cancelled out. The position of the cospectral peak is less impacted, and thus instrumental noise can only lead to an enhancement of scattering of cospectral values, preferentially in the low-frequency range of the cospectrum. In other words, instrumental noise mostly contributes to the low-frequent noise, the red noise. Additionally, physical reasons, such as an inhomogeneous surface roughness length, canopy height in the footprint of the tower, and different range of relevant eddy sizes, may have been reasons for fewer valid high-quality (co)spectra compared to the BOG site.

**4.1.3 Impact of noise on IPS**

The findings indicate that using Ps for estimating correction factors of gases with low turbulent fluctuations, which are measured by a closed-path instrument, can be problematic. Therefore, we recommend using cospectra to estimate $\tau_r$ and $\alpha$ of reactive gases , since these gases exhibit normally low density fluctuations. However, Fig. 3 reveals that $Ps(\Sigma N_r)$ shows a steep decline in the high-frequency range after the peak at BOG, which is similar to the decline of $Ps(T)$. $\Sigma N_r$ concentration was 24.4 ppb on average and exhibits a standard deviation of 9.6 ppb for the averaging period in Fig. 3 suggesting significant differences in concentration levels. It confirms the statement that the concentration is an important driver for the quality of $Ps(\Sigma N_r)$. This leads us to the assumption that the instrument was in principle able to capture differences in concentration levels in the high-frequency range if mixing ratio fluctuations are relatively high.

White noise was observed in power spectra of $CO_2$ and $H_2O$, too. Both gases were measured with an open-path analyzer, but their concentrations are higher and the variability in concentrations of these gases is much larger than for $\Sigma N_r$. It indicates that $Ps(CO_2)$ are clearly less affected by white noise and the instrument is able to capture the high-frequent variability of $CO_2$ well. The assumption of spectral similarity, which is a critical assumption for all in-situ methods, was valid for $Ps(CO_2)$, but was not fulfilled for $Ps(\Sigma N_r)$ due to the influence of red and white noise. Consequentially, an optimization fit with an infinite impulse response function gives unrealistic results for $\tau_r$. Most likely, automatic filtering criteria are not sufficient enough to extract good quality (co)spectra of $\Sigma N_r$ efficiently, and thereby the averaged $Ps(\Sigma N_r)$ used for fitting procedure is dominated by low quality and invalid cases. However, using more restrictive quality selection criteria or narrowing the frequency range for the fitting of the transfer function produced rapidly changing values or even negative values for $\tau_r$. This demonstrates that the estimation of $\tau_r$ with $Ps(\Sigma N_r)$ via IPS is very uncertain and the number of $Ps(\Sigma N_r)$ with sufficient quality was not high enough for a robust fitting. Consequently, for estimating damping factors with IPS certain conditions seem to be fulfilled. For example, instruments need a low detection limit for detecting low turbulent fluctuations, sources and impact of noise and strategies for the elimination of noise should be known, gases should be rather inert or have little interaction with surfaces or other chemical compounds, and, in case of IPS, show a wind speed dependency on damping factors. Similar to cospectral methods, IPS will also benefit from a well-defined footprint, equal canopy height, and sufficient turbulence. Satisfying these

aspects is quite difficult for a custom-built EC system, which is rather new and thus not all attenuation processes are identified, and designed for measuring a trace gas, which consists of several compounds with unknown contribution, complex reaction

570    pathways, and generally low fluctuations. Therefore, IPS is likely to be inappropriate for correcting flux measurement of trace gases with a high white and/or red noise level.

The number of good quality (co)spectra for $CO_2$ and $H_2O$ was at least one order of magnitude higher than for $\Sigma N_r$. Monthly averaged $\alpha$ for $CO_2$ and $H_2O$ by IPS were in the range of 0.95 and 0.90 which is quite reasonable for an open-path instrument and in agreement with studies dealing with the same instrument (Burba et al., 2010; Butterworth and Else, 2018).

575    **4.2    Assessment of cospectral approaches**

**4.2.1    THEO vs. (semi-)empirical approaches**

In general, $\alpha$ values determined by the (semi-) empirical cospectral methods (sICO, ICO and IOG) were considerably lower than the results of THEO. The difference indicates a strong additional damping effect whose impact on $\Sigma N_r$ fluxes is not detected by the fluid dynamics related transfer functions used in THEO. This additional damping must be caused by adsorption

580    processes at the inner surfaces of the inlet system, for example in the converter the sample lines or the CLD. Studies from Aubinet et al. (1999); Bernhofer et al. (2003); Ammann et al. (2006); Spank and Bernhofer (2008)  have also shown that the damping factor by the THEO approach is often too high. Beside disregarded damping processes, this could have also been caused by deviations of the site specific cospectra from theoretical cases. Therefore, it is advisable to apply empirical methods to measurements of gases with unknown properties or to setups and instrument devices with flux loss sources which

585    are difficult to quantify. Empirical methods take the sum of all potential flux losses into account and do not take care of an individual or specific flux loss. The difference between THEO and empirical methods in total flux losses at the two study sites can be explained by the different aerodynamic measurement height. With increasing measurement height, turbulence cospectra are shifted to lower frequencies (Fig. 1 and Fig. 3) and hence a weaker high-frequency damping is expected. Vertical sensor separation was not considered by the spectral transfer function in the THEO approach. However, the impact of vertical

590    sensor separation on the flux loss is very low if the gas analyzer is placed below the anemometer as in the present study. Kristensen et al. (1997) determined a flux loss of only 2% at the vertical separation of $20\,cm$ and measurement height of $1\,m$. This effect gets even smaller with increasing measurement height. Beside the measurement height, also the wind speed and stability are expected to have an influence on the position and shape of the cospectrum and thus on the damping factor. Yet, no clear systematic dependencies of (s)ICO and IOG results on these parameters were found. At BOG, The dependency on

595    wind speed  is only valid for medium and high wind speed classes.  $\alpha$ of IOG and ICO  appear to be invariant to changes in stability at BOG, whereas $\alpha$ of the empirical cospectral approaches at FOR are quite constant under unstable conditions at FOR. In contrast, sICO follows the expected drop at stable conditions as observed for  THEO at both sites. The reason for the difference between sICO and ICO is discussed in Sec. 4.2.2.

600

There could be other effects which superpose the wind speed and stability dependencies, for example, (chemical) damping processes occurring inside the TRANC-CLD system. Humidity and $\Sigma N_r$ could affect the aging of the tube and consequentially the adsorption at inner tube walls. However, we found no dependency of these parameters on damping factor and time response. Interactions with tube walls is probably less important, especially for the tube connecting the end of the TRANC to the CLD, because the main trace gas within the line is NO, which acts rather inert in the absence of ozone and $NO_2$. Because $NO_2$ and $O_3$ are converted in the TRANC, it can be assumed that the influence of interaction with tube walls on time response and high-frequency flux losses is mostly negligible compared to effects, which happen in the CLD and TRANC. The CLD contributes more to the total attenuation than the tubing, but supposedly not as much as the TRANC. Rummel et al. (2002) also used the CLD 780 TR as device for measuring NO fluxes. High-frequency flux losses were rather low and ranged between 21% (close to the ground) and 5% (11 m above ground). Also, Wang et al. (2020) observed low flux losses of NO by approximately 12% by measuring with a QCL (1.7 m above ground).

Consequently, the strongest contributor to the overall damping has to be the TRANC. $NH_3$ is, considering all possible convertible compounds, the most abundant in certain ecosystems, highly reactive, and rather "sticky". In absolute terms it has the highest influence on the damping of $\Sigma N_r$. QCL devices, which may be used for the detection of $NH_3$ (Ferrara et al., 2012; Zöll et al., 2016; Moravek et al., 2019), were equipped with a special designed, heated, and opaque inlet to avoid sticking of $NH_3$ at tube walls, water molecules, and preventing unwanted molecules entering the analyser cell. Thus, $NH_3$ has high flux loss factors ranging from 33 to 46% (Ferrara et al., 2012; Zöll et al., 2016; Moravek et al., 2019). These damping factors are closer to the damping factors of $\Sigma N_r$, in particular for BOG, at which high $NH_3$ concentrations were measured and most of $\Sigma N_r$ can be attributed to $NH_3$ (Hurkuck et al., 2014; Zöll et al., 2016). At FOR, flux losses were lower due to physical reasons and due to lower contribution of $NH_3$ to $\Sigma N_r$ at FOR. According to DELTA-Denuder measurements presented in Zöll et al. (2019), $NH_3$ concentrations were relatively low at FOR site (Beudert and Breit, 2010). 33% of $\Sigma N_r$ were $NH_3$ and 32% were attributed to $NO_2$. $NH_3$ is converted inside the TRANC at the platinum gauze after passing through the actively heated inlet and iron-nickel-chrome (FeNiCr) alloy tube. Since the main part of the pathway is heated and isolated against environmental impacts, the inlet of the TRANC and the distance to the sonic seem to be critical for the detection and attenuation of $NH_3$. Finally, we suppose that the response time and attenuation of our TRANC-CLD system is more similar to that of an $NH_3$ analyzer under a high ambient $NH_3$ load.

[revised manuscript text omitted]
 inclusion of the shift mismatch in Eq. (3) leads to a steeper slope in the empirical transfer function and variations around zero at higher frequencies (see Fig. 1) compared to a first-order function alone (not shown). If $\alpha$ is calculated without including phase-shift effect, we get an overestimation of the damping up to 10% for both sites. This could be expected and indicates that most of the damping is related to a time shift. Until now, there is no ideal transfer function which can capture all damping processes. The transfer function can differ depending on trace gas and site setup. Our empirical transfer function was chosen especially for reactive gases such as $\Sigma N_r$ or $NH_3$ measured with a closed-path instrument. The usage of Eq. (3) for other gases like $CO_2$ or $H_2O$ is not recommended without knowing any spectral characteristics. In case of $CO_2$ and $H_2O$ measured with Li-7500 at FOR and BOG, we have to modify Eq. (3). We would leave out the phase-shift mismatch since the Li-7500 has a faster response and consider using the sensor separation and/or path averaging transfer function (Moore, 1986).

**4.3 Recommendations for correcting high-frequency flux losses of $N_r$ compounds**

$\Sigma N_r$ is a complex trace gas signal, since it consists of many reactive N gases, which have various reaction pathways, and concentrations of the single compounds are unknown. We have shown that very low concentration differences of $\Sigma N_r$ are difficult to detect for the CLD. This has an influence on the variability of (co)spectra, strengthens their susceptibility to noise, and reduces the amount of high-quality (co)spectra. Since power spectra had a strong affinity to white noise and exhibited no spectral similarity to temperature spectrum due to red noise, we recommend using cospectra for estimating $\alpha$. We found that

flux loss is rather chemical driven, in particular determined by the dimensions of the inlet and ambient $NH_3$ load. It could lead to an invariance in wind speed and stability. As a consequence, common approaches, which are based on theoretical, physical assumptions or established dependencies on environmental dependencies, are not suitable for our EC system. Specifying the flux loss of the different compounds is rather difficult due to the measurement of the sum of individual $N_r$ compounds. Thus, we can only roughly estimate the contribution of individual species to the flux and its high frequency loss. At BOG, mostly $NH_3$ seems to influence the damping of $\Sigma N_r$. At FOR, $NH_3$ as well as $NO_2$ were the main $\Sigma N_r$ flux contributors, thereby holding an important role for the detected flux loss at the forest site (see Sec. 4.2.1). Due to the unknown physical and chemical characteristics of $\Sigma N_r$, an empirical approach seems to be the best solution for capturing attenuation processes of $\Sigma N_r$ and its complex compounds. Having carefully considered all pros and cons of the used approaches, our method of choice will be ICO.

A general or site-specific parameterization of the damping as a function of wind speed and stability was not possible for the entire wind speed and stability range. A parameterization would be possible only for certain wind speed and stability ranges. For example, a parameterization can be performed for unstable conditions and for wind speeds above $1.5\,\mathrm{m\,s^{-1}}$ at BOG. As mentioned in Sec. 3.2, other parameters such as global radiation showed no clear dependency on $\alpha$. No significant difference between day and nighttime $\alpha$ values was found. The exchange pattern of $\Sigma N_r$ is rather bi-directional during the entire day. The exchange pattern of inert gases like $CO_2$ is largely related to photosynthesis and respiration. During daytime $CO_2$ exhibits also bi-directional exchange characteristics. During nighttime the exchange of $CO_2$ is mostly unidirectional. Thus, we would expect a diurnal variation of the $CO_2$ attenuation. The influence of global radiation on the biosphere-atmosphere exchange of $\Sigma N_r$ and $CO_2$ was explicitly shown by (Zöll et al., 2019) for FOR. They also investigated drivers of $\Sigma N_r$. However, global radiation explained only 22% of the variability in $\Sigma N_r$ fluxes, whereas 66% of the variability in $CO_2$ fluxes were related to global radiation. $\Sigma N_r$ had the concentration as a second driver, which was approximately 24%. Consequently, there are additional factors controlling the biosphere-atmosphere exchange of total reactive nitrogen, which may be of chemical nature and challenging to quantify. Thus, a flux loss correction of $\Sigma N_r$ after meteorologically classified parameters is not provided.

For an aspired correction of the determined fluxes half-hourly estimated $\alpha$ of the empirical methods will not be used due to their variation with time and to the limited amount of high-quality $\Sigma N_r$ cospectra. Therefore, it is advisable to use averages over certain time periods. We decided to use monthly median values for correcting fluxes at BOG. A bimonthly classification was conducted for FOR, because the rejection rate was higher due to higher uncertainty of cospectra in the low-frequency range. For estimating $\alpha$, a reliable determination of $\tau_r$ is needed. Using a constant $\tau_r$ is possible but not recommended for our $\Sigma N_r$ setup, since $\tau_r$ varied with time and started to increase after a few month. It seems that the variation of $\alpha$ in time was mainly driven by the change of $\tau_r$. The increase of $\tau_r$ and the enhanced variation of $\tau_r$ after a few months could be related to instrumental performance problems caused by aging of the inlet, tubes, and filters, reducing pump performance, problems with the CO supply and TRANC temperature, or a sensitivity loss of the CLD. The variability in $\tau_r$ has also an influence on the meteorological classification of $\alpha$. Generally, it is not known how much the variability in $\tau_r$ contributes to the scattering in $\alpha$ for certain wind speeds or stability values. Thus, usage of $\tau_r$ and the corresponding $\alpha$ classified by meteorological parameters is only recommended for medium or high wind speeds at BOG or near-neutral and unstable atmospheric conditions at both

735  sites. Finally, it seems that the attenuation of the TRANC-CLD system is mainly driven by the performance of the EC setup and by changes in the composition of $\Sigma N_r$.

**5 Conclusions**

We investigated flux losses of total reactive nitrogen ($\Sigma N_r$) measured with a custom-built converter (TRANC) coupled to fast-response CLD above a mixed forest and a semi-natural peatland. We compared five different methods for the quantification and
740  correction of high-frequency attenuation: the first is adapted from Moore (1986) (THEO), the second uses measured cospectra of sensible heat and trace gas flux (ICO), the third uses response time calculated from measured cospectra and estimates damping with modified Kaimal cospectra (Ammann, 1999) (sICO), the fourth uses the measured ogives (IOG) and the fifth method is the power spectral method by Ibrom et al. (2007) (IPS). The flux losses by IPS for our closed-path eddy covariance setups were around 6% at the peatland site (BOG) and around 5% at the forest site (FOR). The attenuation after THEO was
745  about 12% at BOG and about 5% at FOR. The methods using measured cospectra or ogives (ICO, sICO and IOG) showed a flux loss of roughly 16-22% for the forest measurements and around 26-38% for the peatland measurements, with ICO showing the strongest damping at both sites. Flux losses of the empirical approaches are comparable to other EC studies on $\Sigma N_r$ and other reactive nitrogen compounds.

We found that $Ps(\Sigma N_r)$ were heavily affected by white and red noise. No robust estimation of the response time ($\tau_r$) by
750  using measured power spectra was possible. THEO could not capture strong damping processes of $\Sigma N_r$ fluxes, which are likely caused by adsorption processes occurring at inner surfaces of the inlet system or missing information about the contribution of specific gases to $\Sigma N_r$. Consequently, THEO and IPS are not recommended for estimating reliable flux losses of $\Sigma N_r$.

Differences in flux losses are related to measurement height and hence to the variable contribution of small and large-scale eddies to the flux. No systematic or only partly significant dependencies of the empirical methods (ICO, sICO, and IOG) on
755  parameters such as atmospheric stability and wind speed, which have an influence on the shape and position of cospectrum, were observed. In case of the empirical methods, we found a wind speed dependency on damping factors ($\alpha$), apparently a linear decrease in $\alpha$ with increasing wind speed at BOG. However, the trend is limited to wind speeds higher
760  than $1.5\,\mathrm{m\,s^{-1}}$. At FOR, $\alpha$ of IOG, sICO, and ICO seem to be invariant to changes in wind speed. For unstable cases $\alpha$ values are rather constant at FOR ($\sim 0.85$). At BOG, $\alpha$ of IOG and ICO were similar and vary between 0.60 and 0.80 at unstable conditions, whereas sICO values were higher by approximately 0.05-0.15. The expected decline of $\alpha$ with increasing stability was only observed in sICO at both sites, probably related to the usage of Kaimal cospectra. IOG and ICO showed no clear trend for stable cases. We suppose that other factors like varying atmospheric concentration, distribution and strength of sources and
765  sinks, enhanced chemical activity of $\Sigma N_r$ compared to $CO_2$ and $H_2O$, aging of the TRANC inlet, varying CLD performance and vegetation could influence $\alpha$ stronger and may superpose slight effects of wind speed and stability. Thus, a general or site-specific parameterization of the damping for the complete wind speed and stability range was not possible.

The empirical methods perform well at both sites and median $\alpha$ are in the range of former studies about reactive nitrogen compounds. However, we detected significant discrepancies to ICO which were related to site-specific problems or to using different frequency ranges of the cospectrum for the assessment. We discovered a bias between $\alpha$ computed with ICO and sICO for the BOG measurements. No significant bias for ICO and sICO was detected at the FOR site. We supposed that Kaimal cospectra may underestimate the attenuation of fluxes under certain site conditions (cf. Mamadou et al., 2016). Differences in $\alpha$ to IOG are induced by utilizing the low-frequency part of the cospectrum. The low-frequency part is more variable than the high-frequency part on half-hourly basis. Strong attenuation cases could be underestimated by IOG since damping already occurs in the fit range.

Our investigation of different spectral correction methods showed that ICO is most suitable for capturing damping processes of $\Sigma N_r$. However, not all damping processes of reactive gases are fully understood yet and current correction methods have to be improved with regard to quality selection of cospectra. Power spectral and purely theoretical methods which are established in flux calculation software worked well for inert gases, but are not suitable for reactive nitrogen compounds. Estimating damping of EC setups designed for highly reactive gases with an empirical method may be a considerable and reliable option. For further correction of fluxes, we will use monthly median $\alpha$, since half-hourly values will lead to significant uncertainties in fluxes. Using a constant $\tau_r$ is not recommended as we noticed variation of $\tau_r$ with time, which is caused by altering the inlet system. Correcting fluxes after meteorologically classified $\alpha$ is possible if dependencies are exhibited by the EC setup.

*Data availability.*

*Code and data availability.* All data are available upon request from the first author of this study (pascal.wintjen@thuenen.de). Also, Python 3.7 code for damping factor calculation as well as the data analysis code can be requested from the first author. All necessary equations for determining the damping factors are given in this manuscript.

**Appendix A**

**A1   Transfer functions of the $\Sigma N_r$ setup**

Transfer functions used for validation of $\alpha$ after THEO, ICO and sICO are listed in Table A1. A detailed description is given in the mentioned literature. Table 1 contains physical parameters of the setup which are necessary to estimate $\alpha$.

**Table A1.** Transfer functions used for evaluation of the $\Sigma N_r$ damping factors.

| Transfer Function | physical parameters |
|---|---|
| **first-order filter** $$TF_R(f) = \frac{1}{\sqrt{1+(2\pi\tau_r f)^2}}$$ | response time $\tau_r$; for THEO: analyser response time is used $\tau_{r,a}$ ((Moore, 1986; Moncrieff et al., 1997)) |
| **sensor separation** $$TF_s(f) = \exp\left(-9.9(fd_s/u)^{1.5}\right) \text{ with } d_s = d_{sa}|\sin(\alpha_d)|$$ | $u$ wind speed, effective lateral separation distance $d_s$, measured separation distance $d_{sa}$, $\alpha_d$ angle between the line joining the sensors and wind direction (Moore, 1986; Aubinet et al., 2012), |
| **path averaging anemometer** $$TF_w(f_p) = \frac{2}{\pi f_p}\left(1 + \frac{1}{2}\exp\left(-2\pi f_p\right) - 3\frac{1-\exp(-2\pi f_p)}{4\pi f_p}\right); f_p = \frac{fp_1}{u}$$ | $p_1$ sonic path length ((Moore, 1986; Moncrieff et al., 1997; Aubinet et al., 2012)) |
| **tube attenuation** $$TF_{t,lam}(f) = \exp\left(-0.82ReScf_t^2\right) \text{ with } f_t = f \cdot (0.5DL)^{0.5}/v_t$$ | $D$ Diameter of tube, $L$ length of tube, Sc Schmidt Number, Re Reynolds Number, $v_t$ flow speed inside the tube ((Ammann, 1999; Aubinet et al., 1999, 2012)) |
| **phase-shift mismatch** $$TF_{\Delta R}(f) \approx \cos\left[\arctan\left(2\pi f\tau_r\right) - 2\pi f\tau_r\right]$$ | $\tau_r$ response time ((Zeller et al., 1988; Ammann, 1999)) |

**A2 Kaimal cospectrum used in THEO and sICO**

795 The cospectrum for stable conditions after Ammann (1999) has the following form

$$\mathrm{Co}_{mod}(f,a,u) = \frac{f \cdot (a/u)}{0.284 \cdot (1+6.4 \cdot \zeta)^{0.75} + 9.345 \cdot (1+6.4 \cdot \zeta)^{-0.825} \cdot (f \cdot (a/u))^{2.1}} \tag{A1}$$

where $a$ is the aerodynamic measurement height and is given by the difference of measurement height $z$ and the zero-plane displacement height $d$ with $a = z - d$ (Spank and Bernhofer, 2008). $\zeta$ is the stability parameter and is defined by $\zeta = a/L$. $L$ is the Obukov-Length. The cospectrum for unstable conditions is determined by two parts

$$\quad \mathrm{Co}_{mod}(f,a,u) = \begin{cases} 12.92 \cdot f(a/u) \cdot (1 + 26.7 \cdot f(a/u))^{-1.375} & f(a/u) < 0.54 \\ 4.378 \cdot f(a/u) \cdot (1 + 3.8 \cdot f(a/u))^{-2.4} & f(a/u) \geq 0.54 \end{cases} \tag{A2}$$

**Appendix B**

**B1 Results of different damping correction methods**

**Table B1.** Result of the comparison between different damping determination methods at the two measurement sites. Bias ($\Delta$) is computed as averaged difference between $\alpha$. Precision is given as 1.96 standard deviation of the difference. $r$ is the correlation coefficient.

| method | Bavarian Forest | | | Bourtanger Moor | | |
|---|---|---|---|---|---|---|
| | $\Delta$ | $1.96\sigma$ | $r$ | $\Delta$ | $1.96\sigma$ | $r$ |
| ICO, IOG | -0.07 | 0.33 | 0.50 | -0.10 | 0.31 | 0.67 |
| ICO, sICO | 0.0 | 0.25 | 0.78 | -0.07 | 0.33 | 0.66 |
| ICO, THEO | -0.19 | 0.37 | 0.09 | -0.25 | 0.43 | -0.08 |
| ICO, IPS | -0.19 | 0.38 | -0.09 | -0.30 | 0.43 | -0.14 |
| | | | | | | |
| sICO, IOG | -0.07 | 0.36 | 0.36 | -0.03 | 0.36 | 0.42 |
| sICO, THEO | -0.20 | 0.33 | 0.22 | -0.18 | 0.37 | 0.36 |
| sICO, IPS | -0.20 | 0.37 | -0.05 | -0.23 | 0.38 | 0.38 |
| | | | | | | |
| IOG, THEO | -0.12 | 0.22 | 0.0 | -0.15 | 0.26 | 0.01 |
| IOG, IPS | -0.12 | 0.22 | -0.08 | -0.20 | 0.26 | -0.16 |
| | | | | | | |
| THEO,IPS | 0.0 | 0.05 | 0.47 | -0.05 | 0.07 | 0.70 |

**B2 Analysis of the response time estimated by ICO**

**Table B2.** Median $\tau_\mathrm{r}$ averaged over certain measurement periods at both sites.

| Site | time period | averaged $\tau_\mathrm{r}$ [s] | lower quartile [s] | upper quartile [s] |
|---|---|---|---|---|
| Bavarian Forest | Jun 2016 - Nov 2016 | 1.85 | 0.72 | 4.14 |
| | Dec 2016 - Jun 2018 | 3.51 | 1.43 | 7.15 |
| | whole period | 3.13 | 1.26 | 6.46 |
| Bourtanger Moor | Oct 2012 - Dec 2012 | 0.74 | 0.40 | 1.47 |
| | Jan 2013 - Jul 2013 | 1.63 | 0.78 | 3.47 |
| | whole period | 1.37 | 0.67 | 2.87 |

[Figure]

**Figure B1.** Dependency of the response time ($\tau_r$) on stability and wind speed classes as box plots without whiskers and outliers (box frame = 25 % to 75 % interquartile range (IQR), bold line = median). (a) and (b) refer to the BOG site and (c) and (d) to the FOR site.

*Author contributions.* PW wrote the manuscript, carried out the measurements at the forest site and performed data analysis and interpretation. CA gave scientific advice. FS helped with coding and evaluated meteorological measurements. CB conducted the measurements at the peatland site and gave scientific advice. All authors discussed the results and FS, CA and CB reviewed the manuscript.

*Competing interests.* The authors declare that they have no conflict of interest.

*Acknowledgements.* This research was funded by the German Enviroment Agency (UBA) (project FORESTFLUX, support code FKZ 3715512110) and by the German Federal Ministry of Education and Research (BMBF) within the framework of the Junior Research Group NITROSPHERE (support code FKZ 01LN1308A). We thank Undine Zöll for scientific and logistical help, Jeremy Rüffer and Jean-Pierre Delorme for excellent technical support, particularly during the field campaigns and the Bavarian Forest Nationalpark Administration, namely

Burkhard Beudert, Wilhelm Breit and Ludwig Höcker for excellent technical and logistical support at BF site. We further thank two anonymous reviewers for valuable comments that helped improve the quality of the manuscript.

[revised manuscript text omitted]

Zöll, U., Lucas-Moffat, A. M., Wintjen, P., Schrader, F., Beudert, B., and Brümmer, C.: Is the biosphere-atmosphere exchange of total reactive nitrogen above forest driven by the same factors as carbon dioxide? An analysis using artificial neural networks, Atmospheric Environment, 206, 108–118, https://doi.org/10.1016/j.atmosenv.2019.02.042, http://www.sciencedirect.com/science/article/pii/S1352231019301463, 2019.